# On the Robustness of Spectral Algorithms for Semirandom Stochastic Block Models

Aditya Bhaskara [*]     Agastya Vibhuti Jha [†]     Michael Kapralov [‡]     Naren Sarayu Manoj [§]

Davide Mazzali[¶]     Weronika Wrzos-Kaminska [‖]

## Abstract

In a graph bisection problem, we are given a graph $G$ with two equally-sized un-labeled communities, and the goal is to recover the vertices in these communities. A popular heuristic, known as spectral clustering, is to output an estimated com-munity assignment based on the eigenvector corresponding to the second smallest eigenvalue of the Laplacian of $G$. Spectral algorithms can be shown to provably recover the cluster structure for graphs generated from certain probabilistic mod-els, such as the Stochastic Block Model (SBM). However, spectral clustering is known to be non-robust to model mis-specification. Techniques based on semidef-inite programming have been shown to be more robust, but they incur significant computational overheads.

In this work, we study the robustness of spectral algorithms against semirandom adversaries. Informally, a semirandom adversary is allowed to "helpfully" change the specification of the model in a way that is consistent with the ground-truth so-lution. Our semirandom adversaries in particular are allowed to add edges inside clusters or increase the probability that an edge appears inside a cluster. Semiran-dom adversaries are a useful tool to determine the extent to which an algorithm has overfit to statistical assumptions on the input.

On the positive side, we identify classes of semirandom adversaries under which spectral bisection using the *unnormalized* Laplacian is strongly consistent, i.e., it exactly recovers the planted partitioning. On the negative side, we show that in these classes spectral bisection with the *normalized* Laplacian outputs a parti-tioning that makes a classification mistake on a constant fraction of the vertices. Finally, we demonstrate numerical experiments that complement our theoretical findings.

## 1 Introduction

Graph partitioning or clustering is a fundamental unsupervised learning primitive. In a graph par-titioning problem, one seeks to identify clusters of vertices that are highly internally connected and sparsely connected to the outside. This task is of particular significance when the given graph presents a latent community structure. In this setting, the goal is to recover the communities as accurately as possible. Various statistical models that attempt to capture this situation have been pro-posed and studied in the literature. Perhaps the most popular of these is the Symmetric Stochastic Block Model (SSBM) [HLL83].

---

[*]University of Utah. Email: bhaskaraaditya@gmail.com.

[†]University of Chicago. Email: agastyavjha.28@gmail.com.

[‡]École Polytechnique Fédérale de Lausanne. Email: michael.kapralov@epfl.ch.

[§]Toyota Technological Institute at Chicago. Email: nsm@ttic.edu.

[¶]École Polytechnique Fédérale de Lausanne. Email: davide.mazzali@epfl.ch.

[‖]École Polytechnique Fédérale de Lausanne. Email: weronika.wrzos-kaminska@epfl.ch.

38th Conference on Neural Information Processing Systems (NeurIPS 2024).

Following the notation of previous works [AFWZ20; DLS21], in this paper we describe an SSBM with specifications $n, P_1, P_2, p, q$, where $n$ is an even positive integer, $P_1$ and $P_2$ are a partitioning of the vertex set $V = \{1, \ldots, n\}$ into subsets of equal size, and $p$ and $q$ are probabilities. Without loss of generality, we may assume that the partitions $P_1$ and $P_2$ consist of vertices $1, \ldots, n/2$ and $n/2 + 1, \ldots, n$, respectively. Hence, with a mild abuse of notation, we write an SSBM with parameters $n, p, q$ only and write it as $\mathsf{SSBM}(n, p, q)$. Now, let $\mathsf{SSBM}(n, p, q)$ be a distribution over random undirected graphs $G = (V, E)$ where each edge $(v, w) \in P_1 \times P_1$ and $(v, w) \in P_2 \times P_2$ (which we refer to as "internal edges") appears independently with probability $p$, and each edge $(v, w) \in P_1 \times P_2$ (which we refer to as "crossing edges") appears independently with probability $q$. When $p \gg q$, there should be many more internal edges than crossing edges. Hence, we expect the community structure to become more evident as $p$ tends away from $q$.

In such scenarios, our general algorithmic goal is to efficiently identify $P_1$ and $P_2$ when given $G$ without any community labels. This task is hereafter referred to as the *graph bisection problem*. In this work, we will be interested in *exact recovery*, also known as *strong consistency*, in which we want an algorithm that, with probability at least $1 - 1/n$ over the randomness of the instance, exactly returns the partition $\{P_1, P_2\}$ for all $n$ sufficiently large. Other approximate notions of recovery (such as almost exact, partial, and weak recovery) are also well-studied but are beyond the scope of this work.

Although the $\mathsf{SSBM}(n, p, q)$ distribution over graphs is a useful starting point for algorithm design and has led to a deep theory about when recovery is possible and of what nature [Abb18], it may not be representative of all scenarios in which we should expect our algorithms to succeed. To remedy this, researchers have proposed several different random graph models that may be more reflective of properties satisfied by real-world networks. These include the geometric block model [GNW24], the Gaussian mixture block model [LS24], and others.

In this paper, we take a different perspective to graph generation by considering various *semirandom models*. At a high level, a semirandom model for a statistical problem interpolates between an average-case input (for example produced by a model such as the SSBM) and a worst-case input, in a way that still allows for a meaningful notion of ground-truth solution. In our context of graph bisection, this can be achieved by an adversary adding internal edges or by the distribution of internal edges itself being nonhomogeneous (i.e., every internal edge $(v, w)$ appears independently with probability $p_{vw} \geq p$, where the $p_{vw}$ may be chosen adversarially for each internal edge). Researchers have studied similar semirandom models for graph bisection [**beyondclusterin**; FK01; MMV12; MPW16; Moi21] and other statistical problems such as classification under Massart noise [MN06], detecting a planted clique in a random graph [FK01; CSV17; MMT20; BKS23], sparse recovery [KLLST23], and top-$K$ ranking [YCOM24].

These modeling modifications are not necessarily meant to capture a real-world data generation process. Rather, they are a useful testbed with which we can determine whether commonly used algorithms have overfit to statistical assumptions present in the model. In particular, observe that these changes in model specification are ostensibly helpful, in that increasing the number of internal edges should only enhance the community structure. Perhaps surprisingly, it is known that a number of natural algorithms that succeed in the SSBM setting no longer work under such helpful modifications [Moi21]. Therefore, it is natural to ask which algorithms for graph bisection are robust in semirandom models.

At this point, the performance of approaches based on convex programming is well-understood in various semirandom models [FK01; MMV12; MPW16; Moi21; CdM24]. However, in practice, it is impractical to run such an algorithm due to computational costs. Another class of algorithms, that we call *spectral algorithms*, is more widely used in practice. Loosely speaking, a spectral algorithm constructs a matrix $\mathbf{M}$ that is a function of the graph $G$ and outputs a clustering arising from the embedding of the vertices determined by the eigenvectors of $\mathbf{M}$. Popular choices of matrices include the unnormalized Laplacian $\mathbf{L}_G$ and the normalized Laplacian $\mathcal{L}_G$ (we will formally define and intuit these notions in the sequel) [Von07]. This is because structural properties of both $\mathbf{L}_G$ and $\mathcal{L}_G$ imply that the second smallest eigenvalue of each, denoted as $\lambda_2(\mathbf{L}_G)$ and $\lambda_2(\mathcal{L}_G)$, serves as a continuous proxy for connectivity, and the corresponding eigenvector, $\boldsymbol{u}_2(\mathbf{L}_G)$ and $\boldsymbol{u}_2(\mathcal{L}_G)$, has entries whose signs reveal a lot of information about the underlying community structure. This motivates Algorithm 1. It can be run, for example, with $\mathsf{Matrix}(G) := \mathbf{L}_G$ or $\mathsf{Matrix}(G) := \mathcal{L}_G$. Following this discussion, we arrive at the question we study in this paper.

---

**Algorithm 1** SpectralBisection: given $G = (V, E)$, outputs a bipartition of $V$

---

1: **procedure** SpectralBisection($G$)                                    ▷ $G = (V, E)$ is the input graph
2:     $\mathbf{M} \leftarrow \mathsf{Matrix}(G)$                           ▷ $\mathbf{M} \in \mathbb{R}^{V \times V}$ is a matrix with real eigenvalues
3:     $((\lambda_i, \boldsymbol{u}_i))_{i=1}^n \leftarrow$ eigenvalue-eigenvector pairs of $\mathbf{M}$ with $\lambda_1 \leq \cdots \leq \lambda_n$        ▷ $n = |V|$
4:     $S \leftarrow \{v \in V : \boldsymbol{u}_2[v] < 0\}$
5:     **return** $\{S, V \setminus S\}$

---

**Question 1.** *Under which semirandom models do the Laplacian-based spectral algorithms, using the second eigenvector of $\mathbf{L}_G$ or $\mathcal{L}_G$, exactly recover the ground-truth communities $P_1$ and $P_2$?*

**Main contributions.**     Our results show a surprising difference in the robustness of spectral bisection when considering the normalized versus the unnormalized Laplacian. We summarize our results below:

- Consider a nonhomogeneous symmetric stochastic block model with parameters $q < p < \overline{p}$, where every internal edge appears independently with probability $p_{uv} \in [p, \overline{p}]$ and every crossing edge appears independently with probability $q$. We show that under an appropriate spectral gap condition, the spectral algorithm with the unnormalized Laplacian exactly recovers the communities $P_1$ and $P_2$. Moreover, this holds even if an adversary plants $\ll np$ internal edges per vertex prior to the edge sampling phase.

- Consider a stronger semirandom model where the subgraphs on the two communities $P_1$ and $P_2$ are adversarially chosen and the crossing edges are sampled independently with probability $q$. We show that if the graph is sufficiently dense and satisfies a spectral gap condition, then the spectral algorithm with the unnormalized Laplacian exactly recovers the communities $P_1$ and $P_2$.

- We show that there is a family of instances from a nonhomogeneous symmetric stochastic block model in which the spectral algorithm achieves exact recovery with the unnormalized Laplacian, but incurs a constant error rate with the normalized Laplacian. This is surprising because it contradicts conventional wisdom that normalized spectral clustering should be favored over unnormalized spectral clustering [Von07].

We also numerically complement our findings via experiments on various parameter settings.

**Outline.**     The rest of this paper is organized as follows. In Section 2, we more formally define our semirandom models, the Laplacians $\mathbf{L}$ and $\mathcal{L}$, and formally state our results. In Section 3, we give sketches of the proofs of our results. In Section 4, we show results from numerical trials suggested by our theory. In Appendices A.1 and A.5 we prove important auxiliary lemmas we need for our results. In Appendix A.6, we prove our robustness results for the unnormalized Laplacian. In Appendix A.8, we prove our inconsistency result for the normalized Laplacian. In Appendix B, we give additional numerical trials and discussion.

## 2     Models and main results

In this paper, we study unnormalized and normalized spectral clustering in several semirandom SSBMs. These models permit a richer family of graphs than the SSBM alone.

**Matrices related to graphs.**     Throughout this paper, all graphs are to be interpreted as being undirected, and we assume that the vertices of an $n$-vertex graph coincide with the set $\{1, \ldots, n\}$. With this in mind, we begin with defining various matrices associated with graphs, building up to the unnormalized and normalized Laplacians, which are central to the family of algorithms we analyze (Algorithm 1).

**Definition 2.1** (Adjacency matrix)**.** *Let $G = (V, E)$ be a graph. The adjacency matrix $\mathbf{A}_G \in \mathbb{R}^{V \times V}$ of $G$ is the matrix with entries defined as $\mathbf{A}_G[v, w] = \mathbb{1}\{(v, w) \in E\}$.*

**Definition 2.2** (Degree matrix)**.** *Let $G = (V, E)$ be a graph. The degree matrix $\mathbf{D}_G \in \mathbb{R}^{V \times V}$ of $G$ is the diagonal matrix with entries defined as $\mathbf{D}_G[v, v] = \boldsymbol{d}_G[v]$, where $\boldsymbol{d}_G[v]$ is the degree of $v$.*

**Definition 2.3** (Unnormalized Laplacian)**.** *Let $G = (V, E)$ be a graph. The unnormalized Laplacian $\mathbf{L}_G \in \mathbb{R}^{V \times V}$ of $G$ is the matrix defined as $\mathbf{L}_G := \mathbf{D}_G - \mathbf{A}_G = \sum_{(v,w) \in E}(\boldsymbol{e}_v - \boldsymbol{e}_w)(\boldsymbol{e}_v - \boldsymbol{e}_w)^\top$, where $\boldsymbol{e}_i$ denotes the $i$-th standard basis vector.*

**Definition 2.4** (Normalized Laplacians)**.** *Let $G = (V, E)$ be a graph. The symmetric normalized Laplacian $\mathcal{L}_{G,\mathsf{sym}} \in \mathbb{R}^{V \times V}$ and the random walk Laplacian $\mathcal{L}_{G,\mathsf{rw}} \in \mathbb{R}^{V \times V}$ of $G$ are defined as*

$$\mathcal{L}_{G,\mathsf{sym}} := \mathbf{I} - \mathbf{D}_G^{-1/2}\mathbf{A}_G\mathbf{D}_G^{-1/2}\,, \qquad\qquad \mathcal{L}_{G,\mathsf{rw}} := \mathbf{I} - \mathbf{D}_G^{-1}\mathbf{A}_G.$$

For all notions above, when the graph $G$ is clear from context, we omit the subscript $G$. Furthermore, when we discuss normalized Laplacians, we intend its symmetric version $\mathcal{L}_{\mathsf{sym}}$ unless otherwise stated. So, we omit this subscript as well and simply write $\mathcal{L}$.

Next, we define the spectral bisection algorithms. We will discuss some intuition for why these algorithms are reasonable heuristics in Section 3.

**Definition 2.5** (Unnormalized and normalized spectral bisection)**.** *Let $G = (V, E)$ be a graph, and let its unnormalized and normalized Laplacians be $\mathbf{L}$ and $\mathcal{L}$, respectively. We refer to the algorithm resulting from running Algorithm 1 on $G$ with $\mathsf{Matrix}(G) := \mathbf{L}_G$ as* unnormalized spectral bisection. *We refer to the algorithm resulting from running Algorithm 1 on $G$ with $\mathsf{Matrix}(G) = \mathcal{L}_G$ as* normalized spectral bisection.

Our goal is to understand when the above algorithms, applied to a graph with a latent community structure, achieve *exact recovery* or *strong consistency*, defined as follows.

**Definition 2.6.** *Let $\{P_1, P_2\}$ be a partitioning of $V = \{1, \ldots, n\}$, and let $\mathcal{D} := \mathcal{D}(\{P_1, P_2\})$ be a distribution over $n$-vertex graphs $G = (V, E)$. We say that an algorithm is* strongly consistent *or achieves* exact recovery *on $\mathcal{D}$ if given a graph $G \sim \mathcal{D}$ it outputs the correct partitioning $\{P_1, P_2\}$ with probability at least $1 - 1/n$ over the randomness of $G$.*

### 2.1 Nonhomogeneous symmetric stochastic block model

Our first model is a family of nonhomogeneous symmetric stochastic block models, defined below.

**Model 1** (Nonhomogeneous symmetric stochastic block model)**.** *Let $n$ be an even positive integer, $V = \{1, \ldots, n\}$, $\{P_1, P_2\}$ be a partitioning of $V$ into two equally-sized subsets, and $q < p \leq \overline{p}$ be probabilities. Let $\mathcal{D}$ be any probability distribution over graphs $G = (V, E)$ such that for every $(v, w) \in P_1 \times P_1$ and $(v, w) \in P_2 \times P_2$, the edge $(v, w)$ appears in $E$ independently with some probability $p_{vw} \in [p, \overline{p}]$, and for every $(v, w) \in P_1 \times P_2$, the edge $(v, w)$ appears in $E$ independently with probability $q$. We call such $\mathcal{D}$ a nonhomogeneous symmetric stochastic block model (which we will abbreviate as NSSBM). We call the set of all such $\mathcal{D}$ the family of nonhomogeneous stochastic block models with parameters $p, \overline{p}, q$, written as $\mathsf{NSSBM}(n, p, \overline{p}, q)$.*

To visualize Model 1, consider the expected adjacency matrix of some NSSBM distribution. We then have the relations

$$\left[\begin{array}{c|c} p \cdot \mathbf{J}_{n/2} & q \cdot \mathbf{J}_{n/2} \\ \hline q \cdot \mathbf{J}_{n/2} & p \cdot \mathbf{J}_{n/2} \end{array}\right] \leq \left[\begin{array}{c|c} \mathbf{P}_{P_1} & q \cdot \mathbf{J}_{n/2} \\ \hline q \cdot \mathbf{J}_{n/2} & \mathbf{P}_{P_2} \end{array}\right] \leq \left[\begin{array}{c|c} \overline{p} \cdot \mathbf{J}_{n/2} & q \cdot \mathbf{J}_{n/2} \\ \hline q \cdot \mathbf{J}_{n/2} & \overline{p} \cdot \mathbf{J}_{n/2} \end{array}\right],$$

where the leftmost matrix denotes the expected adjacency matrix of $\mathsf{SSBM}(n, p, q)$, the rightmost matrix denotes the expected adjacency matrix of $\mathsf{SSBM}(n, \overline{p}, q)$, $\mathbf{J}_k$ denotes the $k \times k$ all-ones matrix, and $\mathbf{P}_{P_1}$ and $\mathbf{P}_{P_2}$ denote the edge probability matrices for edges internal to $P_1$ and $P_2$, respectively.

The above also shows that the rank of the expected adjacency matrix for $\mathsf{SSBM}(n, p, q)$ is 2. However, the rank for the expected adjacency matrix for some NSSBM distribution may be as large as $\Omega(n)$. Perhaps surprisingly, this will turn out to be unimportant for our entrywise eigenvector perturbation analysis. In particular, the tools we use were originally designed for low-rank signal matrices or spiked low-rank signal matrices [AFWZ20; DLS21; BV24], but we will see that they can be adapted to the signal matrices we consider.

The NSSBM family generalizes the symmetric stochastic block model described in the previous section – this is attained by setting $p_{vw} = p$ for all internal edges $(v, w)$. However, it can also encode biases for certain graph properties. For instance, a distribution from the NSSBM family may encode the idea that certain subsets of $P_1$ are expected to be denser than $P_1$ as a whole.

With this definition in hand, we are ready to formally state our first technical result in Theorem 1.

**Theorem 1.** *Let $p, \overline{p}, q$ be probabilities such that $q < p \leq \overline{p}$ and such that $\alpha := \overline{p}/(p - q)$ is an arbitrary constant. Let $\mathcal{D} \in \mathsf{NSSBM}(n, p, \overline{p}, q)$. Let $n \geq N(\alpha)$ where the function $N(\alpha)$ only depends on $\alpha$. There exists a universal constant $C > 0$ such that if*

$$n(p - q) \geq C \left( \sqrt{n\overline{p} \log n} + \log n \right), \qquad \text{(gap condition)}$$

*then unnormalized spectral bisection is strongly consistent on $\mathcal{D}$.*

We prove Theorem 1 in Appendix A.7.1. In fact, we show a somewhat stronger statement – in addition to the process described above, we also allow the adversary to, before sampling the graph, set a small number of the $p_{vw}$ to 1 (at most $n\overline{p}/\log\log n$ edges per vertex). We detail this further in Appendix A.7.1.

We now remark on the tightness of our gap condition in Theorem 1. A work of Abbe, Bandeira, and Hall [ABH16] identifies an exact information-theoretic threshold above which exact recovery with high probability is possible and below which no algorithm can be strongly consistent. In particular, the threshold states that for any $p$ and $q$ satisfying $\sqrt{p} - \sqrt{q} > \sqrt{2 \log n / n}$, exact recovery is possible, and when $p$ and $q$ do not satisfy this, exact recovery is information-theoretically impossible. Furthermore, Feige and Kilian [FK01] prove that the information-theoretic threshold does not change in a somewhat stronger semirandom model that includes the NSSBM family. Additionally, Deng, Ling, and Strohmer [DLS21] show that unnormalized spectral bisection is strongly consistent all the way to this threshold in the special case where the graph is drawn from $\mathsf{SSBM}(n, p, q)$. By contrast, our gap condition holds in the same critical degree regime as in the information-theoretic threshold (namely, $p = \Theta(\log n / n)$) but our constant is not optimal. We incur this constant loss because for the sake of presentation, we opt for a cleaner argument that can handle the nonhomogeneity and generalizes more readily across degree regimes. To our knowledge, none of these features are present in prior work analyzing spectral methods in an SSBM setting [AFWZ20; DLS21].

## 2.2 Deterministic clusters model

Given Theorem 1, it is natural to ask what happens if we allow the adversary full control over the structure of the graphs in $P_1$ and $P_2$ instead of simply allowing the adversary to perturb the edge probabilities. In this section, we answer this question. We first describe a more adversarial semirandom model than the NSSBM family. We call this model the *deterministic clusters* model, defined as follows.

**Model 2** (Deterministic clusters model). *Let $n$ be an even positive integer, $V = \{1, \ldots, n\}$, $\{P_1, P_2\}$ be a partitioning of $V$ into two equally-sized subsets, $q$ be a probability, and $d_{\mathsf{in}}$ be an integer degree lower bound. Consider a graph $G = (V, E)$ generated according to the following process.*

1. *The adversary chooses arbitrarily graphs $G[P_1]$ and $G[P_2]$ with minimum degree $d_{\mathsf{in}}$;*

2. *Nature samples every edge $(v, w) \in P_1 \times P_2$ to be in $E$ independently with probability $q$.*

3. *The adversary arbitrarily adds edges $(v, w) \in P_1 \times P_1$ and $(v, w) \in P_2 \times P_2$ to $E$ after observing the edges sampled by nature.*

*We call a distribution $\mathcal{D}$ of graphs generated according to the above process a deterministic clusters model (DCM). We call the set of all such $\mathcal{D}$ the family of deterministic clusters models with parameters $d_{\mathsf{in}}$ and $q$, written as $\mathsf{DCM}(n, d_{\mathsf{in}}, q)$.*

The DCM graph generation process is heavily motivated by the one studied by Makarychev, Makarychev, and Vijayaraghavan [MMV12]. This model is much more flexible than the SSBM and NSSBM settings in that the graphs the adversary draws on $P_1$ and $P_2$ are allowed to look very far from random graphs. This means the DCM is a particularly good benchmark for algorithms to ensure they are not implicitly using properties of random graphs that might not hold in the worst case.

Within the DCM setting, we have Theorem 2.

**Theorem 2.** *Let $q$ be a probability and $d_{\mathsf{in}}$ be an integer, and let $\mathcal{D} \in \mathsf{DCM}(n, d_{\mathsf{in}}, q)$. For $G \sim \mathcal{D}$, let $\widehat{\mathbf{L}}$ denote the expectation of $\mathbf{L}$ after step (2) but before step (3) in Model 2. There exists constants*

$C_1, C_2, C_3 > 0$ *such that for all $n$ sufficiently large, if*

$$d_{\mathsf{in}} \geq C_1 \cdot \left( \frac{nq}{2} + \sqrt{n} \right) \quad and \quad \lambda_3(\widehat{\mathbf{L}}) - \lambda_2(\widehat{\mathbf{L}}) \geq \sqrt{n} + C_2 nq + C_3 \left( \sqrt{nq \log n} + \log n \right) ,$$

*then unnormalized spectral bisection is strongly consistent on $\mathcal{D}$.*

We prove Theorem 2 in Appendix A.7.2. We remark that, as in Theorem 1, the constants that appear in Theorem 2 are somewhat arbitrary. They are chosen to make our proofs cleaner and can likely be optimized.

As a basic application of Theorem 2, note that in the SSBM, if $p = \omega(1/\sqrt{n})$ and $q = 1/\sqrt{n}$, then for $n$ sufficiently large, with high probability, the resulting graph satisfies the conditions needed to apply Theorem 2. For a more interesting example, let $P_1$ and $P_2$ be two $d$-regular spectral expanders with $d = \omega(\sqrt{n})$ and let $q \leq 1/\sqrt{n}$. On top of both of these two graph classes, one can further allow arbitrary edge insertions inside $P_1$ and $P_2$ while still being guaranteed exact recovery from unnormalized spectral bisection.

## 2.3 Inconsistency of normalized spectral clustering

Notice that in Theorem 1 and Theorem 2, we only address the strong consistency of the unnormalized Laplacian in our nonhomogeneous and semirandom models. But what happens when we run spectral bisection with the *normalized* Laplacian?

In Theorem 3, we prove that there is a subfamily of instances belonging to $\mathsf{NSSBM}(n, p, \overline{p}, q)$ with $\overline{p} = 6p, q = p/2$ on which unnormalized spectral bisection is strongly consistent (following from Theorem 1) but normalized spectral clustering is inconsistent in a rather strong sense. Thus, one cannot obtain results similar to Theorem 1 and Theorem 2 for normalized spectral bisection.

**Theorem 3.** *For all $n$ sufficiently large, there exists a nonhomogeneous stochastic block model such that unnormalized spectral bisection is strongly consistent whereas normalized spectral bisection (both symmetric and random-walk) incurs a misclassification rate of at least $24\%$ with probability $1 - 1/n$.*

We prove Theorem 3 in Appendix A.8. Furthermore, we expect that it is straightforward to adapt the example in Theorem 3 to prove an analogous result for our DCM setting.

The result of Theorem 3 may run counter to conventional wisdom, which suggests that normalized spectral clustering should be favored over the unnormalized variant [Von07]. Perhaps a more nuanced view in light of Theorem 1 and Theorem 2 is to acknowledge that the normalized Laplacian and its eigenvectors enjoy stronger concentration guarantees [SB15; DLS21], but the unnormalized Laplacian's second eigenvector is more robust to monotone adversarial changes.

## 2.4 Open problems

Perhaps the most natural follow-up question inspired by our results is to determine whether the restriction that every internal edge probability $p_{vw} \leq \overline{p}$ can be lifted entirely while still maintaining strong consistency of the unnormalized Laplacian (Theorem 2). Another exciting direction for future work is to lower the degree and/or spectral gap requirement present in our results in the DCM setting (Theorem 2). Finally, we only study insertion-only monotone adversaries, as crossing edge deletions change the second eigenvector of the expected Laplacian. It would be illuminating to understand the robustness of Laplacian-based spectral algorithms against a monotone adversary that is also allowed to delete crossing edges. We are optimistic that the answers to one or more of these questions will further improve our understanding of the robustness of spectral clustering to "helpful" model misspecification.

## 3 Analysis sketch

First, let us give some intuition as to why one may expect that unnormalized spectral bisection is robust against our monotone adversaries. Here and in the sequel, let $\boldsymbol{u}_2^{\star} = [\mathbb{1}_{n/2} \oplus -\mathbb{1}_{n/2}]/\sqrt{n}$, where $\mathbb{1}_k$ denotes the all-1s vector in $k$ dimensions and $\oplus$ denotes vector concatenation. Let $\mathbf{L}$ be the unnormalized Laplacian of the graph we want to partition, $\mathbf{L}^{\star} := \mathbb{E}[\mathbf{L}]$, $\mathbf{E} := \mathbf{L} - \mathbf{L}^{\star}$, and $\lambda_i^{\star} := \lambda_i(\mathbf{L}^{\star})$ for $1 \leq i \leq n$. For an edge $(v, w)$, let $\boldsymbol{e}_{vw} := \boldsymbol{e}_v - \boldsymbol{e}_w$, so that $\boldsymbol{e}_{vw}$ is an edge incidence vector corresponding to the edge $(v, w)$. Let $p_{vw}$ be the probability that the edge $(v, w)$

appears in $G$ and observe that $\mathbf{L}^\star$ can be written as

$$\mathbf{L}^\star = \sum_{(v,w) \in E_{\text{internal}}} p_{vw} \cdot \boldsymbol{e}_{vw} \boldsymbol{e}_{vw}^T + \sum_{(v,w) \in E_{\text{crossing}}} q \cdot \boldsymbol{e}_{vw} \boldsymbol{e}_{vw}^T,$$

where $E_{\text{internal}} = (P_1 \times P_1) \cup (P_2 \times P_2)$ and $E_{\text{crossing}} = P_1 \times P_2$. We can verify that $\boldsymbol{u}_2^\star$ is an eigenvector of $\mathbf{L}^\star$ – indeed, we do so in Lemma A.14. And, for now, assume that $\boldsymbol{u}_2^\star$ does correspond to the second smallest eigenvalue of $\mathbf{L}^\star$ (in our NSSBM family, this is easily ensured by enforcing $p > q$). Moreover, for every internal edge $(v,w) \in E_{\text{internal}}$, we have $\langle \boldsymbol{e}_{vw}, \boldsymbol{u}_2^\star \rangle = 0$. Hence, any changes in internal edges do not change the fact that $\boldsymbol{u}_2^\star$ is an eigenvector of the perturbed matrix. Thus, if the sampled $\mathbf{L}$ is close enough to $\mathbf{L}^\star$, then it is plausible that the second eigenvector of $\mathbf{L}$, denoted as $\boldsymbol{u}_2$, is pretty close to $\boldsymbol{u}_2^\star$. In fact, the following conceptually stronger statement holds. If the subgraph formed by selecting just the crossing edges of $G$ is regular, then $\boldsymbol{u}_2^\star$ is an eigenvector of $\mathbf{L}$. This follows from the fact that $\boldsymbol{u}_2^\star$ is an eigenvector of the unnormalized Laplacian of any regular bipartite graph where both sides have size $n/2$ and the previous observation that every internal edge is orthogonal to $\boldsymbol{u}_2^\star$.

To make this perturbation idea more formal, we recall the Davis-Kahan Theorem. Loosely, it states that $\|\boldsymbol{u}_2 - \boldsymbol{u}_2^\star\|_2 \lesssim \|(\mathbf{L} - \mathbf{L}^\star)\boldsymbol{u}_2^\star\|_2 / (\lambda_3^\star - \lambda_2^\star)$ (we give a more formal statement in Lemma A.15). Expanding the entrywise absolute value $|(\mathbf{L} - \mathbf{L}^\star)\boldsymbol{u}_2^\star|$ reveals that its entries can be expressed as $2 |\boldsymbol{d}_{\text{out}}[v] - \mathbb{E}[\boldsymbol{d}_{\text{out}}[v]]| / \sqrt{n}$, where $\boldsymbol{d}_{\text{out}}[v]$ denotes the number of edges incident to $v$ crossing to the opposite community as $v$. This is unaffected by any increase in the number of edges incident to $v$ that stay within the same community as $v$, denoted as $\boldsymbol{d}_{\text{in}}[v]$. Hence, regardless of how many internal edges we add before sampling or what substructures they encourage/create, if we have $\lambda_2^\star \ll \lambda_3^\star$, then we get $\|\boldsymbol{u}_2 - \boldsymbol{u}_2^\star\|_2 \leq o(1)$. This immediately implies that $\boldsymbol{u}_2$ is a correct classifier on all but an $o(1)$ fraction of the vertices.

**Entrywise analysis of $\boldsymbol{u}_2$ and NSSBM strong consistency.** In order to achieve strong consistency, we need that for all $n$ sufficiently large, $\boldsymbol{u}_2$ is a perfect classifier. Unfortunately, the above argument does not immediately give that. In particular, in the density and spectral gap regimes we consider, the bound of $o(1)$ yielded by the Davis-Kahan theorem is not sufficiently small to directly yield $\|\boldsymbol{u}_2^\star - \boldsymbol{u}_2\|_2 \ll 1/\sqrt{n}$. Instead, we carry out an entrywise analysis of $\boldsymbol{u}_2$. A general framework for doing so is given by Abbe, Fan, Wang, and Zhong [AFWZ20] and is adapted to the unnormalized and normalized Laplacians by Deng, Ling, and Strohmer [DLS21].

At a high level, we adapt the analysis of Deng, Ling, and Strohmer [DLS21] to our setting. We consider the intermediate estimator vector $(\mathbf{D} - \lambda_2 \mathbf{I})^{-1} \mathbf{A} \boldsymbol{u}_2^\star$. This is a natural choice because we can verify $(\mathbf{D} - \lambda_2 \mathbf{I})^{-1} \mathbf{A} \boldsymbol{u}_2 = \boldsymbol{u}_2$. We will see that it is enough to show that this intermediate estimator correctly classifies all the vertices while satisfying $|(\mathbf{D} - \lambda_2 \mathbf{I})^{-1} \mathbf{A} (\boldsymbol{u}_2^\star - \boldsymbol{u}_2)| \leq |(\mathbf{D} - \lambda_2 \mathbf{I})^{-1} \mathbf{A} \boldsymbol{u}_2^\star|$ (again, the absolute value is taken entrywise). With this in mind, taking some entry indexed by $v \in V$ and multiplying both sides by $\boldsymbol{d}[v] - \lambda_2$ (which we will show is positive with high probability), we see that it is enough to show

$$|\langle \boldsymbol{a}_v, \boldsymbol{u}_2^\star - \boldsymbol{u}_2 \rangle| \leq |\langle \boldsymbol{a}_v, \boldsymbol{u}_2^\star \rangle| = \frac{|\boldsymbol{d}_{\text{in}}[v] - \boldsymbol{d}_{\text{out}}[v]|}{\sqrt{n}}, \tag{1}$$

where $\boldsymbol{a}_v$ denotes the $v$-th row of $\mathbf{A}$. The advantage of this rewrite is that the right hand side can be uniformly bounded, so it is enough to control the left hand side.

To argue about the left hand side of (1), it may be tempting to use the fact that $\boldsymbol{a}_v$ is a Bernoulli random vector and use Bernstein's inequality to argue about the sum of rescalings of these Bernoulli random variables. Unfortunately, we cannot do this since $\boldsymbol{u}_2$ and $\boldsymbol{a}_v$ are dependent. To resolve this, we use a leave-one-out trick [AFWZ20; BV24]. We can think of this as leaving out the vertex $v$ corresponding to the entry we want to analyze and sampling the edges incident to the rest of the vertices. The second eigenvector of the resulting $\mathbf{L}^{(v)}$, denoted as $\boldsymbol{u}_2^{(v)}$, is a very good proxy for $\boldsymbol{u}_2$ and is independent from $\boldsymbol{a}_v$. Hence, we may complete the proof of Theorem 1.

One of our main observations is that although this style of analysis was originally built for low-rank signal matrices [AFWZ20; BV24], it can be adapted to handle the nonhomogeneity inside $P_1$ and $P_2$. In particular, the nonhomogeneity we permit in the NSSBM family may make $\mathbf{L}^\star$ look very far from a spiked low-rank signal matrix. Furthermore, our entrywise analysis of eigenvectors under perturbations is one of the first that we are aware of that moves beyond analyzing low-rank signal matrices or spiked low-rank signal matrices.

| | $L_1$ | $L_2$ | $R$ |
|---|---|---|---|
| $L_1$ | $Kp \cdot \mathbb{1}_{n/4 \times n/4}$ | $p \cdot \mathbb{1}_{n/4 \times n/4}$ | $q \cdot \mathbb{1}_{n/2 \times n/2}$ |
| $L_2$ | $p \cdot \mathbb{1}_{n/4 \times n/4}$ | $Kp \cdot \mathbb{1}_{n/4 \times n/4}$ | |
| $R$ | $q \cdot \mathbb{1}_{n/2 \times n/2}$ | | $p \cdot \mathbb{1}_{n/2 \times n/2}$ |

Table 1: $\mathbf{A}^\star$ for Theorem 3 is defined to have the above block structure.

**Extension to deterministic clusters.** To prove Theorem 2, we start again at (1). An alternate way to upper bound the left hand side is to use the Cauchy-Schwarz inequality. A variant of the Davis-Kahan theorem gives us control over $\|\boldsymbol{u}_2 - \boldsymbol{u}_2^\star\|_2$ while $\|\boldsymbol{a}_v\|_2 = \sqrt{\boldsymbol{d}[v]}$. The advantage of this is that we get a worst-case upper bound on the left hand side of (1) – it holds no matter what edges orthogonal to $\boldsymbol{u}_2^\star$ are inserted before or after nature samples the crossing edges (which are precisely the internal edges). Combining these and using the fact that the right hand side of (1) is increasing in $\boldsymbol{d}_{\mathsf{in}}[v]$ (and increases faster than $\|\boldsymbol{a}_v\|_2 = \sqrt{\boldsymbol{d}[v]}$) allows us to complete the proof of Theorem 2.

**Inconsistency of normalized spectral bisection.** Finally, we describe the family of hard instances we use to prove Theorem 3. To motivate this family of instances, recall that by the graph version of Cheeger's inequality, the second eigenvalue of $\mathcal{L}$ and the corresponding eigenvector can be used to find a sparse cut in $G$. Thus, if we create sparse cuts inside $P_1$ that are sparser than the cut formed by separating $P_1$ and $P_2$, then conceivably the normalized Laplacian's second eigenvector may return the new sparser cut.

To make this formal, consider the following graph structure. Let $n$ be a multiple of 4. Let $L_1$ consist of indices $1, \ldots, n/4$, $L_2$ consist of indices $n/4 + 1, \ldots, n/2$, and $R$ consist of indices $n/2 + 1, \ldots, n$. Consider the block structure induced by the matrix $\mathbf{A}^\star = \mathbb{E}[\mathbf{A}]$ shown in Table 1.

Intuitively, as $K$ gets larger, the cut separating $L_1$ from $V \setminus L_1$ becomes sparser. From Cheeger's inequality, this witnesses a small $\lambda_2(\mathcal{L})$ and therefore the corresponding $\boldsymbol{u}_2(\mathcal{L})$ may return the cut $L_1, V \setminus L_1$. We formally prove that this is indeed what happens when $K$ is a sufficiently large constant and then Theorem 3 follows.

## 4 Numerical trials

We programmatically generate synthetic graphs that help illustrate our theoretical findings using the libraries NetworkX 3.3 (BSD 3-Clause license), SciPy 1.13.0 (BSD 3-Clause License), and NumPy 1.26.4 (modified BSD license) [HSS08; VGO+20; HMvdW+20]. We ran all our experiments on a free Google Colab instance with the CPU runtime, and each experiment takes under one hour to run. In this section we focus on a setting that allows relating Theorem 1 and Theorem 3, and defer more experiments that investigate both NSSBM and DCM graphs to Appendix B.

To put Theorem 1 and Theorem 3 in perspective, we consider graphs generated following the process outlined in the proof of Theorem 3, which gives rise to the following benchmark distribution.

**Benchmark distribution.** Let $n$ be divisible by 4 and let $\{P_1, P-2\}$ be a partitioning of $V = [n]$ into two equally-sized subsets. Let $\{L_1, L_2\}$ be a bipartition of $P_1$ such that $|L_1| = |L_2| = n/4$ and call $L = P_1$, $R = P_2$ for convenience as in the proof of Theorem 3. Then, for some $p, \overline{p}, q \in [0, 1]$ such that $q \le p \le \overline{p}$, consider the distribution $\mathcal{D}_{p,\overline{p},q}$ over graphs $G = (V, E)$ obtained by sampling every edge $(u, v) \in (L_1 \times L_1) \cup (L_2 \times L_2)$ independently with probability $\overline{p}$, every edge $(u, v) \in (L_1 \times L_2) \cup (R \times R)$ independently with probability $p$, and every edge $(u, v) \in L \times R$ independently with probability $q$. One can see that $\mathcal{D}_{p,\overline{p},q}$ is in fact in the set $\mathsf{NSSBM}(n, p, \overline{p}, q)$.

**Setup.** Let us fix $n = 2000$, $p = 24 \log n/n$, $q = 8 \log n/n$. For varying values of $\overline{p}$ in the range $[p, 1]$, we sample $t = 10$ independent draws $G$ from $\mathcal{D}_{p,\overline{p},q}$. For each of them, we run spectral bisection (i.e. Algorithm 1) with matrices $\mathbf{L}, \mathcal{L}_{\mathsf{sym}}, \mathcal{L}_{\mathsf{rw}}, \mathbf{A}$. Then, we compute the *agreement* of the bipartition hence obtained (with respect to the planted bisection), that is the fraction of correctly classified vertices. We average the agreement across the $t$ independent draws. The results are shown in the top left plot of Fig. 1. Another natural way to get a bipartition of $V$ from the eigenvector is a *sweep cut*. In a sweep cut, we sort the entries of $\boldsymbol{u}_2$ and take the vertices corresponding to the smallest $n/2$ entries to be on one side of the bisection and put the remaining on the other side. The average agreement obtained in this other fashion is shown in the bottom left plot of Fig. 1.

**Theoretical framing.** As per Theorem 1, we expect unnormalized spectral bisection to achieve exact recovery (i.e. agreement equal to 1) whenever $\bar{p} \leq \bar{p}_{\mathsf{max}}$, where

$$\bar{p}_{\mathsf{max}} = \frac{(n(p-q) - \log n)^2}{n \log n} \tag{2}$$

is obtained by rearranging the precondition of Theorem 1, ignoring the constants and disregarding the fact that $\alpha$ should be $O(1)$. On the contrary, the proof of Theorem 3 shows that normalized spectral bisection misclassifies a constant fraction of vertices provided that $p/q \geq 2$ (which our choice of parameters satisfies) and $\bar{p} \geq \bar{p}_{\mathsf{thr}}$, where

$$\bar{p}_{\mathsf{thr}} = 3 \cdot p^2/q. \tag{3}$$

In Fig. 1, the solid vertical line corresponds to the value of $\bar{p}_{\mathsf{thr}}$ on the $x$-axis, and the dashed vertical line corresponds to the value of $\bar{p}_{\mathsf{max}}$ on the $x$-axis. In particular, observe that in our setting $\bar{p}_{\mathsf{thr}} < \bar{p}_{\mathsf{max}}$, so there is an interval of values for $\bar{p}$ where we expect Theorem 1 and Theorem 3 to apply simultaneously.

**Empirical evidence: consistency.** One can see from the top left plot in Fig. 1 that the agreement of unnormalized spectral bisection is $100\%$ for all values of $\bar{p}$, even beyond $\bar{p}_{\mathsf{thr}}$ and $\bar{p}_{\mathsf{max}}$. On the other hand, the agreement of the bipartition obtained from all other matrices (hence including normalized spectral bisection) drops below $70\%$ well before the threshold $\bar{p}_{\mathsf{thr}}$ predicted by Theorem 3. From the right plot in Fig. 1, we see that computing the bipartition by taking a sweep cut of $n/2$ vertices does not change the results – $\boldsymbol{u}_2$ of the unnormalized Laplacian continues to achieve $100\%$ agreement, while for all other matrices the corresponding $\boldsymbol{u}_2$ remains inconsistent.

**Empirical evidence: embedding variance.** From the setting of the experiment we just illustrated, observe that as we increase $\bar{p}$, we expect the subgraph $G[L]$ to have increasing volume. As illustrated in Fig. 1, this seems to correlate with a decrease in the "variance" of the second eigenvector $\boldsymbol{u}_2$ of the unnormalized Laplacian with respect to the ideal second eigenvector $\boldsymbol{u}_2^\star$. More precisely, we compute the average distance squared of the embedding of a vertex in $\boldsymbol{u}_2$ from its ideal embedding in $\boldsymbol{u}_2^\star$, i.e. the quantity

$$\min_{s \in \{\pm 1\}} \frac{1}{n} \| \boldsymbol{u}_2 - s \cdot \boldsymbol{u}_2^\star \|_2^2. \tag{4}$$

This suggests that not only does the second eigenvector of the unnormalized Laplacian remain robust to monotone adversaries, but it actually concentrates more strongly around the ideal embedding $\boldsymbol{u}_2^\star$.

**Empirical evidence: example embedding.** Let us fix the value $\bar{p} = \bar{p}_{\mathsf{thr}}$, for which we see in Fig. 3 that all matrices except the unnormalized Laplacian fail to recover the planted bisection. We generate a graph from $\mathcal{D}_{p,\bar{p},q}$, and plot how the vertices are embedded in the real line by the second eigenvector of all the matrices we consider. The result is shown in Fig. 1, where the three horizontal dashed lines, from top to bottom, respectively correspond to the value of $1/\sqrt{n}, 0, -1/\sqrt{n}$ on the $y$-axis.

## 4.1 Related work

**Community detection.** Community detection has garnered significant attention in theoretical computer science, statistics, and data science. For a general overview of recent progress and related literature, see the survey by Abbe [Abb18]. In what follows, we discuss the works we believe are most related to what we study in this paper.

As mentioned in the introduction, perhaps the most fundamental and well-studied model is the symmetric stochastic block model (SSBM), due to [HLL83]. The celebrated work of Abbe, Bandeira, and Hall [ABH16] gives sharp bounds on the threshold for exact recovery for the SSBM setting. They complement their result by showing that SDP based methods can achieve the information theoretic lower bound for the planted bisection problem, even with a monotone adversary [Moi21]. A line of work [AFWZ20; DLS21] demonstrates that natural spectral algorithms achieve exact recovery for the SSBM all the way to the information-theoretic threshold.

**Generalizations of the symmetric stochastic block model.** Since the introduction of SBMs [HLL83], numerous variants have been proposed that are designed to better reflect real-world graph properties. For instance, real-life social networks are likely to contain triangles. To address this, Sankararaman and Baccelli [SB17] introduced a spatial stochastic block model, sometimes known as the geometric stochastic block model (GSBM). Other variations were introduced in the works of

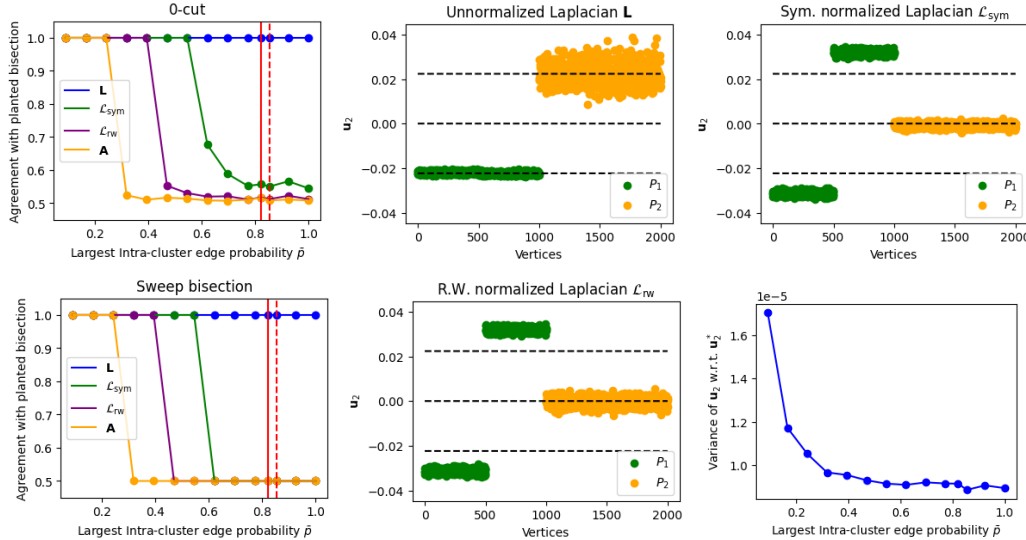

Figure 1: **Top left, bottom left**: Agreement with the planted bisection of the bipartition obtained from several matrices associated with an input graph generated from a distribution in NSSBM$(n, p, \overline{p}, q)$ for fixed values of $n, p, q$ and varying values of $\overline{p}$. In the top left plot, the bipartition is the 0-cut of the second eigenvector, as in Algorithm 1. In the bottom left plot, the bipartition is the sweep cut of the first $n/2$ vertices in the second eigenvector. The dashed vertical line corresponds to $\overline{p}_{\mathsf{max}} = \overline{p}_{\mathsf{max}}(n, p, q)$ (see (2)), and the solid vertical line corresponds to $\overline{p}_{\mathsf{thr}} = \overline{p}_{\mathsf{thr}}(n, p, q)$ (see (3)). **Top middle, top right, bottom middle**: Embedding of the vertices given by the second eigenvector $\boldsymbol{u}_2$ of several matrices associated with a graph sampled from $\mathcal{D}_{p, \overline{p}, q}$ with $\overline{p} = \overline{p}_{\mathsf{thr}}$. Horizontal dashed lines, from top to bottom, correspond to $1/\sqrt{n}, 0, -1/\sqrt{n}$ respectively.
**Bottom right**: Variance of the embedding in the second eigenvector $\boldsymbol{u}_2$ of the unnormalized Laplacian with respect to the ideal eigenvector $\boldsymbol{u}_2^\star$ (see (4)), for input graphs generated from a distribution in NSSBM$(n, p, \overline{p}, q)$ with fixed values of $n, p, q$ and varying values of $\overline{p}$.

[GPMS18; GMPS19]. Subsequent work studies the performance of spectral algorithms on certain Gaussian or Geometric Mixture block models [ABRS20; ABD21; LS24; GNW24].

Studying community detection with a semirandom model approaches this modeling question differently. Rather than implicitly encouraging a particular structure within the clusters like the models just mentioned, a semirandom adversary (including the ones we study in this paper) can more directly test the robustness of the algorithm to specially designed substructures.

**Semirandom and monotone adversaries.** As far as we are aware, Blum and Spencer [BS95] were the first to introduce a semirandom model. Within this model, they studied graph coloring problems. Feige and Kilian [FK01] demonstrated that semidefinite programming methods can accurately recover communities up to a certain threshold, even in the semi-random setting. Other problems, such as detecting a planted clique [Jer92; Ku95; BHKKMP19], have also been studied in the semi-random model of [FK01]. In the setting of planted clique, a natural spectral algorithm fails against monotone adversaries [MMT20; BKS23]. Monotone adversaries and semirandom models have also been extensively studied for other statistical and algorithmic problems [VA18; KLLST23; GC23; BGLMSY24]. Finally, [SL17] shows that a spectral heuristic due to Boppana [Bop87] is robust under a monotone adversary that is allowed to both insert internal edges and delete crossing edges. However, as far as we are aware, this algorithm does not fit in the framework of Algorithm 1.

We remark that the models we study in this paper are most closely related to models studied by [MN06] and [MMV12]. In particular, allowing increased internal edge probabilities is analogous to Massart noise in classification problems, and our model with adversarially chosen internal edges can be seen as the same model as that studied in [MMV12] (although without allowing crossing edge deletions). Finally, note that Cohen-Addad, d'Orsi, and Mousavifar [CdM24] give a near-linear time algorithm for graph clustering in the model of [MMV12], though they do not explicitly show their algorithm is strongly consistent on instances that are information-theoretically exactly recoverable.

**Acknowledgments.** AB was partially supported by the National Science Foundation under Grant Nos. CCF-2008688 and CCF-2047288. NSM was supported by a National Science Foundation Graduate Research Fellowship. We thank Avrim Blum and Yury Makarychev for helpful discussions. We thank Nirmit Joshi for pointing us to the reference [DLS21].

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

|  |  |
|---|---|
|  | July 2023. arXiv: 2203.04002 [cs.DS]. URL: https://proceedings.mlr.press/v195/kelner23a.html (cited on pages 2, 10). |
| [Ku95] | Ludk Kuera. Expected complexity of graph partitioning problems. *Discrete Applied Mathematics*, 57(2):193–212, 1995. ISSN: 0166-218X. DOI: https://doi.org/10.1016/0166-218X(94)00103-K. URL: https://www.sciencedirect.com/science/article/pii/0166218X9400103K. Combinatorial optimization 1992 (cited on page 10). |
| [LLV17] | Can M Le, Elizaveta Levina, and Roman Vershynin. Concentration and regularization of random graphs. *Random Structures & Algorithms*, 51(3):538–561, 2017. arXiv: 1506.00669 [math.PR] (cited on page 17). |
| [LS24] | Shuangping Li and Tselil Schramm. Spectral clustering in the gaussian mixture block model, 2024. arXiv: 2305.00979 [stat.ML] (cited on pages 2, 10). |
| [MMV12] | Konstantin Makarychev, Yury Makarychev, and Aravindan Vijayaraghavan. Approximation algorithms for semi-random partitioning problems. In *Proceedings of the Forty-Fourth Annual ACM Symposium on Theory of Computing*, STOC '12, pages 367–384, New York, New York, USA. Association for Computing Machinery, 2012. ISBN: 9781450312455. arXiv: 1205.2234 [cs.DS] (cited on pages 2, 5, 10). |
| [MN06] | Pascal Massart and Élodie Nédélec. Risk bounds for statistical learning. *The Annals of Statistics*, 34(5), October 2006. ISSN: 0090-5364. DOI: 10.1214/009053606000000786. URL: http://dx.doi.org/10.1214/009053606000000786 (cited on pages 2, 10). |
| [MMT20] | Theo McKenzie, Hermish Mehta, and Luca Trevisan. A new algorithm for the robust semi-random independent set problem. In *Proceedings of the Thirty-First Annual ACM-SIAM Symposium on Discrete Algorithms*, SODA '20, pages 738–746, Salt Lake City, Utah. Society for Industrial and Applied Mathematics, 2020. arXiv: 1808.03633 [cs.DS] (cited on pages 2, 10). |
| [Moi21] | Ankur Moitra. *Semirandom stochastic block models*. In *Beyond the Worst-Case Analysis of Algorithms*. Tim Roughgarden, editor. Cambridge University Press, 2021, pages 212–233. DOI: 10.1017/9781108637435.014 (cited on pages 2, 9). |
| [MPW16] | Ankur Moitra, William Perry, and Alexander S. Wein. How robust are reconstruction thresholds for community detection? In *Proceedings of the Forty-Eighth Annual ACM Symposium on Theory of Computing*, STOC '16, pages 828–841, Cambridge, MA, USA. Association for Computing Machinery, 2016. ISBN: 9781450341325. DOI: 10.1145/2897518.2897573. arXiv: 1511.01473 [cs.DS]. URL: https://doi.org/10.1145/2897518.2897573 (cited on page 2). |
| [SB17] | Abishek Sankararaman and François Baccelli. Community detection on euclidean random graphs. In *2017 55th Annual Allerton Conference on Communication, Control, and Computing (Allerton)*, pages 510–517, 2017. DOI: 10.1109/ALLERTON.2017.8262780 (cited on page 9). |
| [SB15] | Purnamrita Sarkar and Peter J. Bickel. Role of normalization in spectral clustering for stochastic blockmodels. *The Annals of Statistics*, 43(3), June 2015. ISSN: 0090-5364. DOI: 10.1214/14-aos1285. arXiv: 1310.1495 [stat.ML]. URL: http://dx.doi.org/10.1214/14-AOS1285 (cited on page 6). |
| [SL17] | Martin R. Schuster and Maciej Liskiewicz. New Abilities and Limitations of Spectral Graph Bisection. In Kirk Pruhs and Christian Sohler, editors, *25th Annual European Symposium on Algorithms (ESA 2017)*, volume 87 of *Leibniz International Proceedings in Informatics (LIPIcs)*, 66:1–66:15, Dagstuhl, Germany. Schloss Dagstuhl – Leibniz-Zentrum für Informatik, 2017. ISBN: 978-3-95977-049-1. DOI: 10.4230/LIPIcs.ESA.2017.66. URL: https://drops.dagstuhl.de/entities/document/10.4230/LIPIcs.ESA.2017.66 (cited on page 10). |
| [Ver18] | Roman Vershynin. *High-dimensional probability: An introduction with applications in data science*, volume 47. Cambridge university press, 2018 (cited on page 14). |

[VA18]     Aravindan Vijayaraghavan and Pranjal Awasthi. Clustering semi-random mixtures of Gaussians. In Jennifer Dy and Andreas Krause, editors, *Proceedings of the 35th International Conference on Machine Learning*, volume 80 of *Proceedings of Machine Learning Research*, pages 5055–5064. PMLR, July 2018. arXiv: 1711.08841 [cs.DS]. URL: https://proceedings.mlr.press/v80/vijayaraghavan18a.html (cited on page 10).

[VGO+20]   Pauli Virtanen, Ralf Gommers, Travis E. Oliphant, Matt Haberland, Tyler Reddy, David Cournapeau, Evgeni Burovski, Pearu Peterson, Warren Weckesser, Jonathan Bright, Stéfan J. van der Walt, Matthew Brett, Joshua Wilson, K. Jarrod Millman, Nikolay Mayorov, Andrew R. J. Nelson, Eric Jones, Robert Kern, Eric Larson, C J Carey, lhan Polat, Yu Feng, Eric W. Moore, Jake VanderPlas, Denis Laxalde, Josef Perktold, Robert Cimrman, Ian Henriksen, E. A. Quintero, Charles R. Harris, Anne M. Archibald, Antônio H. Ribeiro, Fabian Pedregosa, Paul van Mulbregt, and SciPy 1.0 Contributors. SciPy 1.0: Fundamental Algorithms for Scientific Computing in Python. *Nature Methods*, 17:261–272, 2020. DOI: 10.1038/s41592-019-0686-2 (cited on page 8).

[Von07]    Ulrike Von Luxburg. A tutorial on spectral clustering. *Statistics and computing*, 17:395–416, 2007. arXiv: 0711.0189 [cs.DS] (cited on pages 2, 3, 6).

[YCOM24]   Yuepeng Yang, Antares Chen, Lorenzo Orecchia, and Cong Ma. Top-$K$ ranking with a monotone adversary. In Shipra Agrawal and Aaron Roth, editors, *Proceedings of Thirty Seventh Conference on Learning Theory*, volume 247 of *Proceedings of Machine Learning Research*, pages 5123–5162. PMLR, July 2024. arXiv: 2402.07445 [stat.ML]. URL: https://proceedings.mlr.press/v247/yang24b.html (cited on page 2).

# A   Deferred proofs

In this section, we build the tools we need to prove Theorem 1, Theorem 2, andTheorem 3. Throughout, it will be helpful to refer to the overview (Section 3) for a proof roadmap.

**Notation in the proofs.**     In all proofs, we adopt the notation used in the technical overview (Section 3). Additionally, for a vertex $v \in V$, let $P(v)$ denote the community that $v$ belongs to.

## A.1   Concentration inequalities

Our proof strategy for Theorem 1 and Theorem 2 is to appeal to Lemma A.23, which guarantees strong consistency provided that $\boldsymbol{d}[v] - \lambda_2 > 0$, $\boldsymbol{d}_{\mathsf{in}}[v] > \boldsymbol{d}_{\mathsf{out}}[v]$, and $|\langle \boldsymbol{a}_v, \boldsymbol{u}_2^\star - \boldsymbol{u}_2 \rangle| \leq (\boldsymbol{d}_{\mathsf{in}}[v] - \boldsymbol{d}_{\mathsf{out}}[v])/\sqrt{n}$ for all vertices $v$. Proving that the first two conditions hold is relatively easy. In the setting of Theorem 1, it essentially follows from concentration of the degrees, which is proved in Appendix A.2. In the setting of Theorem 2, it follows from the assumptions of the Theorem. Proving that the third condition holds is the main technical challenge.

For all three parts, our proofs rely on several auxiliary concentration results. We prove these in Appendix A.3 and Appendix A.4.

We extensively use the following variants of Bernstein's Inequality, which can be derived from [Ver18, Theorem 2.8.4].

**Lemma A.1.** *Let* $X = \sum_{i=1}^m X_i$, *where* $X_i = 1$ *with probability* $p_i$ *and* $X_i = 0$ *with probability* $1 - p_i$ *and all the* $X_i$ *are independent. Let* $\mu = \mathbb{E}[X]$. *Then, for all* $t > 0$ *we have*

$$\Pr[|X - \mu| \geq t] \leq 2\exp\left(-\min\left\{\frac{t^2}{4\sum_{i=1}^m p_i(1 - p_i)}, \frac{3t}{4}\right\}\right).$$

From this, we get the following very useful corollary.

**Lemma A.2.** *Let* $X = \sum_{i=1}^m X_i$, *where* $X_i = 1$ *with probability* $p_i$ *and* $X_i = 0$ *with probability* $1 - p_i$ *and all the* $X_i$ *are independent. Let* $\mu = \mathbb{E}[X]$. *Then, for all* $t > 0$, *with probability at least* $1 - \delta$ *we have*

$$|X - \mu| \leq \sqrt{4\sum_{i=1}^m p_i(1 - p_i)\log(2/\delta)} + 4/3\log(2/\delta).$$

## A.2 Concentration of degrees

In this Section, we give concentration statements regarding the number of internal vertices incident to each vertex and the number of crossing edges incident to each vertex. We then compare these against $\lambda_2$.

**Lemma A.3.** *Suppose the crossing edges are sampled identically and independently with probability $q$. Then, for some universal constant $C > 0$, with probability at least $1 - \delta$ we have that*

$$\forall v \in V, \quad |\boldsymbol{d}_{\text{out}}[v] - \mathbb{E}\left[\boldsymbol{d}_{\text{out}}[v]\right]| \leq C\left(\sqrt{nq \log\left(n/\delta\right)} + \log\left(n/\delta\right)\right).$$

*Proof of Lemma A.3.* Choose some $v \in V$. Consider the random variable $\boldsymbol{d}_{\text{out}}[v]$. Using Lemma A.2, we have that there is a constant $C > 0$ such that with probability at least $1 - \delta/n$ one has

$$|\boldsymbol{d}_{\text{out}}[v] - \mathbb{E}\left[\boldsymbol{d}_{\text{out}}[v]\right]| \leq C\left(\sqrt{4nq/2 \log\left(2n/\delta\right)} + \log\left(2n/\delta\right)\right).$$

Taking a union bound over all $n$ vertices completes the proof of Lemma A.3. $\qquad\square$

Note that Lemma A.3 above applies in both the settings of Theorem 1 and Theorem 2.

**Lemma A.4.** *Suppose the internal edges are sampled independently with probabilities $p_{vw}$ such that $p \leq p_{vw} \leq \bar{p}$. Then, for some universal constant $C > 0$, with probability $\geq 1 - \delta$ we have that*

$$\forall v \in V, \quad |\boldsymbol{d}_{\text{in}}[v] - \mathbb{E}\left[\boldsymbol{d}_{\text{in}}[v]\right]| \leq C\left(\sqrt{\sum_{w \in P(v)\backslash\{v\}} p_{vw}(1 - p_{vw}) \log\left(n/\delta\right)} + \log\left(n/\delta\right)\right).$$

*Proof of Lemma A.4.* As before, choose some $v \in V$ and consider the random variable $\boldsymbol{d}_{\text{in}}[v]$. By Lemma A.2, we have that there is a constant $C > 0$ such that with probability at least $1 - \delta/n$ one has

$$|\boldsymbol{d}_{\text{in}}[v] - \mathbb{E}\left[\boldsymbol{d}_{\text{in}}[v]\right]| \leq C\left(\sqrt{4 \sum_{w \in P(v)\backslash\{v\}} p_{vw}(1 - p_{vw}) \log\left(2n/\delta\right)} + \log\left(2n/\delta\right)\right).$$

Taking a union bound over all $n$ vertices completes the proof of Lemma A.4. $\qquad\square$

Combining the above two lemmas, we obtain a lower-bound on $\boldsymbol{d}_{\text{in}}[v] - \boldsymbol{d}_{\text{out}}[v]$. In particular, the following lemma implies that in the setting of Theorem 1, we have $\boldsymbol{d}_{\text{in}}[v] > \boldsymbol{d}_{\text{out}}[v]$. This will be required for applying Lemma A.23.

**Lemma A.5.** *There exists a universal constant $C > 0$ such that with probability $\geq 1 - \delta$, in the same settings as Lemma A.3 and Lemma A.4 and assuming the gap condition in Theorem 1, if $p \geq q$, then for all $v \in V$ we have*

$$\boldsymbol{d}_{\text{in}}[v] - \boldsymbol{d}_{\text{out}}[v] \geq \frac{n(p - q)}{2} - C\left(\sqrt{np \log\left(n/\delta\right)} + \log\left(n/\delta\right)\right).$$

*Proof of Lemma A.5.* Let $v \in V$. First, we call Lemma A.3 with a failure probability of $\delta/(2n)$ to conclude that

$$\boldsymbol{d}_{\text{out}}[v] \leq \frac{nq}{2} + C_{A.3}\left(\sqrt{\frac{nq}{2} \log\left(2n^2/\delta\right)} + \log\left(2n^2/\delta\right)\right).$$

Next, we call Lemma A.4 with a failure probability of $\delta/(2n)$ to conclude that

$$
\boldsymbol{d}_{\text{in}}[v] \geq \sum_{w \in P(v) \setminus \{v\}} p_{vw} - C_{A.4} \left( \sqrt{\sum_{w \in P(v) \setminus \{v\}} p_{vw}(1 - p_{vw}) \log \left(2n^2/\delta\right)} + \log \left(2n^2/\delta\right) \right)
$$

$$
\geq \sum_{w \in P(v) \setminus \{v\}} p_{vw} - C_{A.4} \left( \sqrt{\sum_{w \in P(v) \setminus \{v\}} p_{vw} \log \left(2n^2/\delta\right)} + \log \left(2n^2/\delta\right) \right)
$$

$$
\geq \frac{np}{2} - 2C_{A.4} \left( \sqrt{\frac{np}{2} \log \left(n^2/\delta\right)} + \log \left(2n^2/\delta\right) \right).
$$

where the last line uses the fact that $x - c\sqrt{x}$ is increasing in $x$ whenever $x \geq c^2/4$ and $c > 0$. We subtract and conclude the proof of Lemma A.5 by a union bound. $\square$

The following lemma will be useful for lower-bounding $\boldsymbol{d}[v] - \lambda_2$ in Theorem 1.

**Lemma A.6.** *Suppose every crossing edge appears independently with probability $q$. Then, with probability $\geq 1 - \delta$, for all $v \in V$ we have*

$$
\lambda_2 \leq 2\boldsymbol{d}_{\text{out}}[v] + C \left( \sqrt{nq \log (n/\delta)} + \log (n/\delta) \right).
$$

*Proof of Lemma A.6.* Observe that with probability at least $1 - \delta$, $\boldsymbol{d}_{\text{out}}[w] - \mathbb{E}[\boldsymbol{d}_{\text{out}}[v]] \leq \sqrt{2nq \log(2n/\delta)} + 2\log(2n/\delta)$ for all $w \in V$ by Lemma A.2. Then, for every $v \in V$ we have

$$
\frac{2}{n} \sum_{w \in P(v)} \boldsymbol{d}_{\text{out}}[w] - \boldsymbol{d}_{\text{out}}[v] = \left( \frac{2}{n} \sum_{w \in P(v)} \boldsymbol{d}_{\text{out}}[w] - \mathbb{E}[\boldsymbol{d}_{\text{out}}[v]] \right) + (\mathbb{E}[\boldsymbol{d}_{\text{out}}[v]] - \boldsymbol{d}_{\text{out}}[v])
$$

$$
\leq \left| \frac{2}{n} \sum_{w \in P(v)} \boldsymbol{d}_{\text{out}}[w] - \mathbb{E}[\boldsymbol{d}_{\text{out}}[v]] \right| + |\mathbb{E}[\boldsymbol{d}_{\text{out}}[v]] - \boldsymbol{d}_{\text{out}}[v]|
$$

$$
\leq \sqrt{2nq \log (2n/\delta)} + \sqrt{2nq \log (2n/\delta)} + 4\log (2n/\delta)
$$

$$
\leq 3\sqrt{nq \log (n/\delta)} + 10\log (n/\delta).
$$

Next, by the min-max principle, we have

$$
\lambda_2 \leq \sum_{(w,w') \in E} (\boldsymbol{u}_2^{\star}[w] - \boldsymbol{u}_2^{\star}[w'])^2 = \frac{4}{n} \sum_{w \in P(v)} \boldsymbol{d}_{\text{out}}[w].
$$

Combining everything, we get

$$
\lambda_2 \leq 2 \left( \frac{2}{n} \sum_{w \in P(v)} \boldsymbol{d}_{\text{out}}[w] \right) \leq 2 \left( \boldsymbol{d}_{\text{out}}[v] + 3\sqrt{nq \log (n/\delta)} + 10\log (n/\delta) \right),
$$

completing the proof of Lemma A.6. $\square$

We can now lower-bound $\boldsymbol{d}[v] - \lambda_2$. Note that the following lower bound implies that $\boldsymbol{d}[v] > \lambda_2$, as required by Lemma A.23.

**Lemma A.7.** *In the setting of Theorem 1, with probability $\geq 1 - \delta$, for all $v \in V$, we have $\boldsymbol{d}[v] - \lambda_2 > n(p - q)/4$.*

*Proof of Lemma A.7.* Recall that the gap condition in Theorem 1 tells us that $p$ and $q$ are such that for a universal constant $C$,

$$
n(p - q) \geq C \left( \sqrt{np \log (n/\delta)} + \log (n/\delta) \right).
$$

We have for all $n$ sufficiently large (specifically, $n \geq N(\alpha, \delta)$ for some $N$ that is a function only of the constant $\alpha$, and we take $\delta \geq 1/n^{O(1)}$) that with probability at least $1 - \delta$,

$$
\begin{aligned}
\boldsymbol{d}[v] - \lambda_2 &= \boldsymbol{d}_{\text{in}}[v] - \boldsymbol{d}_{\text{out}}[v] + (2\boldsymbol{d}_{\text{out}}[v] - \lambda_2) \\
&\geq \boldsymbol{d}_{\text{in}}[v] - \boldsymbol{d}_{\text{out}}[v] - C_{A.6} \left( \sqrt{nq \log\left(n/\delta\right)} + \log\left(n/\delta\right) \right) \\
&\geq \frac{n(p-q)}{2} - (C_{A.5} + C_{A.6}) \left( \sqrt{np \log\left(n/\delta\right)} + \log\left(n/\delta\right) \right),
\end{aligned}
$$

so insisting

$$
\frac{n(p-q)}{4} \geq (C_{A.5} + C_{A.6}) \left( \sqrt{np \log\left(n/\delta\right)} + \log\left(n/\delta\right) \right) + 1
$$

gives the condition required to complete the proof of Lemma A.7. $\qquad \square$

The following technical lemma will be useful for upper-bounding $\|\boldsymbol{u}_2\|_\infty$ in Lemma A.22.

**Lemma A.8.** *In the setting of Theorem 1, there exists a universal constant $C$ such that with probability $\geq 1 - \delta$, for all $v \in V$ we have*

$$
\frac{n\overline{p} + \log\left(n/\delta\right)}{\boldsymbol{d}[v] - \lambda_2} \leq 4\alpha + C.
$$

*Proof of Lemma A.8.* By Lemma A.7, we have with probability $\geq 1 - \delta$ that for all $v \in V$,

$$
\boldsymbol{d}[v] - \lambda_2 \geq \frac{n(p-q)}{4}.
$$

This gives

$$
\frac{n\overline{p} + \log\left(n/\delta\right)}{\boldsymbol{d}[v] - \lambda_2} \leq \frac{4(n\overline{p} + \log\left(n/\delta\right))}{n(p-q)} = \frac{4\overline{p}}{p-q} + \frac{4\log\left(n/\delta\right)}{n(p-q)} \leq 4\alpha + C.
$$

This completes the proof of Lemma A.8. $\qquad \square$

### A.3 Concentration of Laplacian and eigenvalue perturbations

For the matrix concentration lemmas, we need a result due to Le, Levina, and Vershynin [LLV17]. We reproduce it below.

**Lemma A.9** ([LLV17, Theorem 2.1]). *Consider a random graph from the model $G(n, \{p_{ij}\})$. Let $d = \max_{ij} np_{ij}$. For any $r \geq 1$, the following holds with probability at least $1 - n^{-r}$ for a universal constant $C$. Consider any subset consisting of $10n/d$ vertices, and reduce the weights of the edges incident to those vertices in an arbitrary way. Let $d'$ be the maximal degree of the resulting graph. Then, the adjacency matrix $\mathbf{A}'$ of the new weighted graph satisfies*

$$
\|\mathbf{A}' - \mathbb{E}[\mathbf{A}]\|_{\text{op}} \leq Cr^{3/2} \left( \sqrt{d} + \sqrt{d'} \right).
$$

*Moreover, the same holds for $d'$ being the maximal $\ell_2$ norm of the rows of $\mathbf{A}'$.*

**Lemma A.10.** *Let $\mathbf{L}$ be a Laplacian sampled from the nonhomogeneous Erdős-Rényi model where each edge $(i, j)$ is present independently with probability $p_{ij}$. Then, there exists a universal constant $C$ such that for all $n$ sufficiently large, with probability $\geq 1 - \delta$ for any $\delta \geq n^{-10}$,*

$$
\|\mathbf{L} - \mathbb{E}[\mathbf{L}]\|_{\text{op}} \leq C \left( \sqrt{n \max_{(i,j)\,:\, p_{ij}\neq 1} p_{ij} \log\left(n/\delta\right)} + \log\left(n/\delta\right) \right).
$$

*Proof of Lemma A.10.* Without loss of generality, for all $p_{ij}$ that are 1, reset their probabilities to 0. To see that this is valid, let $\mathbf{L}'$ be a Laplacian sampled from this modified distribution and notice that $\mathbf{L}' - \mathbb{E}[\mathbf{L}'] = \mathbf{L} - \mathbb{E}[\mathbf{L}]$.

By Lemma A.9 and Lemma A.2, we have with probability $\geq 1 - \delta/2$ that

$$\|\mathbf{A} - \mathbb{E}\left[\mathbf{A}\right]\|_{\mathrm{op}} \leq 200 C_{A.9} \sqrt{2n \max_{ij} p_{ij} + C_{A.2}\left(\sqrt{n \max_{ij} p_{ij} \log(8n/\delta)} + \log(8n/\delta)\right)}$$

$$\leq 400 C_{A.9} C_{A.2} \sqrt{n \max_{ij} p_{ij} + \log(8n/\delta)}$$

$$\leq 400 C_{A.9} C_{A.2} \left(\sqrt{n \max_{ij} p_{ij} \log(8n/\delta)} + \log(8n/\delta)\right)$$

and by Lemma A.3 and Lemma A.4, we have with probability $1 - \delta/2$ that

$$\|\mathbf{D} - \mathbb{E}\left[\mathbf{D}\right]\|_{\mathrm{op}} \leq \max_{v \in V} |\boldsymbol{d}_{\mathsf{out}}[v] - \mathbb{E}\left[\boldsymbol{d}_{\mathsf{out}}[v]\right]| + \max_{v \in V} |\boldsymbol{d}_{\mathsf{in}}[v] - \mathbb{E}\left[\boldsymbol{d}_{\mathsf{in}}[v]\right]|$$

$$\leq 2 \max\{C_{A.3}, C_{A.4}\}\left(\sqrt{n \max_{ij} p_{ij} \log\left(2n/\delta\right)} + \log\left(2n/\delta\right)\right)$$

Now, observe that with probability $\geq 1 - \delta$ (following from a union bound),

$$\|\mathbf{L} - \mathbb{E}\left[\mathbf{L}\right]\|_{\mathrm{op}} = \|\mathbf{D} - \mathbb{E}\left[\mathbf{D}\right] - (\mathbf{A} - \mathbb{E}\left[\mathbf{A}\right])\|_{\mathrm{op}} \leq \|\mathbf{D} - \mathbb{E}\left[\mathbf{D}\right]\|_{\mathrm{op}} + \|\mathbf{A} - \mathbb{E}\left[\mathbf{A}\right]\|_{\mathrm{op}}$$

$$\leq 800 C_{A.9} C_{A.2} \max\{C_{A.3}, C_{A.4}\}\left(\sqrt{n \max_{ij} p_{ij} \log\left(8n/\delta\right)} + \log\left(8n/\delta\right)\right),$$

completing the proof of Lemma A.10. $\qquad\square$

By applying the above lemma, we can show that there is a gap between $\lambda_3$ and $\lambda_2^\star$, which will allow us to apply Davis-Kahan style bounds. More concretely, Lemma A.11 and Lemma A.12, together with Lemma A.16, show that $\|u_2 - u_2^\star\|_2$ is small. This will be useful for proving that in the context for Theorem 1, the condition $|\langle \boldsymbol{a}_v, \boldsymbol{u}_2^\star - \boldsymbol{u}_2\rangle| \leq (\boldsymbol{d}_{\mathsf{in}}[v] - \boldsymbol{d}_{\mathsf{out}}[v])/\sqrt{n}$ in Lemma A.23 is satisfied.

**Lemma A.11.** *In the setting of Theorem 1, there exists a universal constant $C$ such that the following holds.*

*Let $p$ and $q$ be such that we have*

$$n(p - q) \geq C\left(\sqrt{n\overline{p}\log\left(n/\delta\right)} + \log(n/\delta)\right).$$

*Then, for any $\delta \geq n^{-10}$, with probability $\geq 1 - \delta$, we have $\lambda_3 - \lambda_2^\star \geq n(p - q)/4$.*

*Proof of Lemma A.11.* By Weyl's inequality and Lemma A.10, we have with probability $\geq 1 - \delta$ that

$$\lambda_3 - \lambda_2^\star \geq \lambda_3^\star - \lambda_2^\star - \|\mathbf{L} - \mathbf{L}^\star\|_{\mathrm{op}} \geq \frac{n(p - q)}{2} - C_{A.10}\left(\sqrt{n\overline{p}\log\left(n/\delta\right)} + \log(n/\delta)\right).$$

Let $C \geq 4C_{A.10}$. Then,

$$\frac{n(p - q)}{4} \geq C_{A.10}\left(\sqrt{n\overline{p}\log\left(n/\delta\right)} + \log(n/\delta)\right).$$

Subtracting completes the proof of Lemma A.11. $\qquad\square$

Next, we bound $\|\mathbf{E}\boldsymbol{u}_2^\star\|_2$, which we will need in order to apply our Davis-Kahan style bound in Lemma A.16. We remark that Lemma A.12 below holds both in the setting of Theorem 1 and of Theorem 2.

**Lemma A.12.** *Suppose each crossing edge in our graph appears independently with probability $q$. There exists a universal constant $C$ such that for all $n$ sufficiently large, with probability $\geq 1 - \delta$, we have*

$$\|\mathbf{E}\boldsymbol{u}_2^\star\|_2 \leq C\left(\frac{\log\left(1/\delta\right)}{\log n}\right)^{3/2}\left(\sqrt{nq} + (nq\log\left(n/\delta\right))^{1/4} + \sqrt{\log\left(n/\delta\right)}\right).$$

*Proof of Lemma A.12.* Observe that $|\mathbf{E}\boldsymbol{u}_2^\star| = 2\left|\boldsymbol{d}_{\mathsf{out}} - \mathbb{E}\left[\boldsymbol{d}_{\mathsf{out}}\right]\right|/\sqrt{n}$. By Lemma A.3, for all $v \in V$, with probability $\geq 1 - \delta/2$, we have $\boldsymbol{d}_{\mathsf{out}}[v] \leq nq/2 + C_{A.3}\left(\sqrt{nq \cdot \log\left(2n/\delta\right)} + \log\left(2n/\delta\right)\right)$.

So, if we let $\mathbf{A}_{\mathsf{out}}$ and $\mathbf{A}_{\mathsf{out}}^\star$ denote the adjacency matrices consisting only of the crossing edges and the expected value of that, respectively, then invoking Lemma A.9, with probability $\geq 1 - \delta$, we have

$$\left\|\mathbf{E}\boldsymbol{u}_2^\star\right\|_2 = \frac{2\left\|\boldsymbol{d}_{\mathsf{out}} - \mathbb{E}\left[\boldsymbol{d}_{\mathsf{out}}\right]\right\|_2}{\sqrt{n}} = \frac{2\left\|\left(\mathbf{A}_{\mathsf{out}} - \mathbf{A}_{\mathsf{out}}^\star\right)\mathbb{1}\right\|_2}{\sqrt{n}} \leq 2\left\|\mathbf{A}_{\mathsf{out}} - \mathbf{A}_{\mathsf{out}}^\star\right\|_{\mathsf{op}}$$

$$\leq 2C_{A.9}\left(\frac{\log\left(2/\delta\right)}{\log n}\right)^{3/2}\left(\sqrt{\frac{nq}{2}} + \sqrt{C_{A.3}}\sqrt{nq + \sqrt{nq\log\left(2n/\delta\right)} + \log\left(2n/\delta\right)}\right),$$

completing the proof of Lemma A.12. $\qquad\square$

Finally, we apply Lemma A.9 in order to bound bound $\left\|\boldsymbol{a}_v - \boldsymbol{a}_v^\star\right\|_2$.

**Lemma A.13.** *In the setting of Theorem 1, with probability $\geq 1 - \delta$, we have*

$$\left\|\boldsymbol{a}_v - \boldsymbol{a}_v^\star\right\|_2 \leq C\left(\frac{\log\left(1/\delta\right)}{\log n}\right)^{3/2}\left(\sqrt{n\overline{p}} + \left(n\overline{p}\log\left(n/\delta\right)\right)^{1/4} + \sqrt{\log\left(n/\delta\right)}\right).$$

*Proof of Lemma A.13.* We use a similar proof to that of Lemma A.12. Indeed, invoke Lemma A.9 (observe that we can set $p_{ij}$ for the deterministic internal edges to $0$ as they do not affect $\mathbf{A} - \mathbb{E}\left[\mathbf{A}\right]$) and notice that

$$\left\|\boldsymbol{a}_v - \boldsymbol{a}_v^\star\right\|_2 \leq \left\|\mathbf{A} - \mathbf{A}^\star\right\|_{\mathsf{op}} \leq C_{A.9}\left(\frac{\log\left(2/\delta\right)}{\log n}\right)^{3/2}\left(\sqrt{n\overline{p}} + \left(n\overline{p}\log\left(2n/\delta\right)\right)^{1/4} + \sqrt{\log\left(2n/\delta\right)}\right),$$

where we used $d' \leq n(\overline{p} + q)/2 + 2\max\left\{C_{A.3}, C_{A.4}\right\}\left(\sqrt{n\overline{p}\log\left(2n/\delta\right)} + \log\left(2n/\delta\right)\right)$ from combining Lemma A.3 and Lemma A.4. This completes the proof of Lemma A.13. $\qquad\square$

### A.4 Eigenvector perturbations

In this Appendix, we give our Euclidean norm eigenvector perturbation bounds.

First, we verify that $\boldsymbol{u}_2^\star$ is indeed the second eigenvector of $\mathbf{L}^\star$.

**Lemma A.14.** *In the setting of Theorem 1, we have $\mathbf{L}^\star\boldsymbol{u}_2^\star = \lambda_2(\mathbf{L}^\star)\boldsymbol{u}_2^\star = nq\boldsymbol{u}_2^\star$, where $\mathbf{L}^\star = \mathbb{E}\left[\mathbf{L}\right]$.*

*In the setting of Theorem 2, we have $\mathbf{L}^\star\boldsymbol{u}_2^\star = \lambda_2(\mathbf{L}^\star)\boldsymbol{u}_2^\star = nq\boldsymbol{u}_2^\star$, where $\mathbf{L}^\star$ denotes the Laplacian matrix that agrees with $\mathbf{L}$ on all internal edges and agrees with $\mathbb{E}\left[\mathbf{L}\right]$ on all crossing edges.*

*Proof of Lemma A.14.* In both cases, one can check that $\boldsymbol{u}_2^\star$ is an eigenvector of $\mathbf{L}^\star$ with eigenvalue $nq$: for any $v \in P_2$ (i.e. $\boldsymbol{u}_2^\star[v] = -1/\sqrt{n}$ without loss of generality), one has

$$\left(\mathbf{L}^\star\boldsymbol{u}_2^\star\right)_v = \frac{1}{\sqrt{n}}\left(-(\boldsymbol{d}_{\mathsf{in}}[v] + nq/2) - \sum_{w \in P_1 : \{v,w\} \in E}(-1) + \sum_{w \in P_2}(-q)\right) = -\frac{nq}{\sqrt{n}} = nq \cdot \boldsymbol{u}_2^\star[v].$$

By virtue of the above observations, it suffices to argue that $nq < \lambda_3(\mathbf{L}^\star) \leq \cdots \leq \lambda_n(\mathbf{L}^\star)$.

In the setting of Theorem 1, we claim $\lambda_3^\star \geq \frac{n(p+q)}{2} > nq$. This is because because $p_{vw} \geq p$, which implies that if we consider $\mathbf{L}_1^\star$ to be the expected Laplacian for $\mathsf{SSBM}(n, p, q)$ and $\mathbf{L}_2^\star$ to be the expected Laplacian for $\mathsf{NSSBM}(n, p, \overline{p}, q)$, then $\mathbf{L}_2^\star \succeq \mathbf{L}_1^\star$..

In the setting of Theorem 2, we have $\lambda_3(\widehat{\mathbf{L}}) - \lambda_2(\widehat{\mathbf{L}}) > nq$, by the theorem assumption. Since $\mathbf{L}^\star$ is obtained from $\widehat{\mathbf{L}}$ by adding the adversarial edges, we have $\lambda_i(\mathbf{L}^\star) \geq \lambda_i(\widehat{\mathbf{L}})$ for all $i$. In particular, we have $\lambda_3(\mathbf{L}^\star) \geq \lambda_3(\widehat{\mathbf{L}}) = \lambda_2(\widehat{\mathbf{L}}) + (\lambda_3(\widehat{\mathbf{L}}) - \lambda_2(\widehat{\mathbf{L}})) > nq$, where the last inequality is using the fact $\lambda_2(\widehat{\mathbf{L}}) \geq 0$. Therefore, $nq$ must be the second eigenvalue of $\mathbf{L}^\star$, completing the proof of Lemma A.14. $\qquad\square$

Next, we prove a general Davis-Kahan style bound.

**Lemma A.15.** *Let* $\mathbf{L}$ *and* $\widehat{\mathbf{L}}$ *be two weighted Laplacian matrices. Let* $\boldsymbol{u}_2$ *and* $\widehat{\boldsymbol{u}_2}$ *be the second eigenvectors of* $\mathbf{L}$ *and* $\widehat{\mathbf{L}}$, *respectively. Then,*

$$\|\boldsymbol{u}_2 - \widehat{\boldsymbol{u}_2}\|_2 \le \sqrt{2} \cdot \min \left\{ \frac{\left\|(\widehat{\mathbf{L}} - \mathbf{L})\boldsymbol{u}_2\right\|_2}{\left|\lambda_3(\widehat{\mathbf{L}}) - \lambda_2(\mathbf{L})\right|}, \frac{\left\|(\widehat{\mathbf{L}} - \mathbf{L})\widehat{\boldsymbol{u}_2}\right\|_2}{\left|\lambda_3(\mathbf{L}) - \lambda_2(\widehat{\mathbf{L}})\right|} \right\}$$

*Proof of Lemma A.15.* One can get this sort of guarantee from variants of the Davis-Kahan theorem, but it is more illuminating to write an eigenvalue decomposition and observe it from there. Without loss of generality, assume that $\langle \widehat{\boldsymbol{u}_2}, \boldsymbol{u}_2 \rangle \ge 0$ (indeed, otherwise we can always negate $\widehat{\boldsymbol{u}_2}$ if this is not the case). Notice that

$$\left\|(\widehat{\mathbf{L}} - \mathbf{L})\boldsymbol{u}_2\right\|_2^2 = \left\|\left(\widehat{\mathbf{L}} - \lambda_2(\mathbf{L})\mathbf{I}\right)\boldsymbol{u}_2\right\|_2^2$$

$$= (\lambda_2(\widehat{\mathbf{L}}) - \lambda_2(\mathbf{L}))^2 \langle \widehat{\boldsymbol{u}_2}, \boldsymbol{u}_2 \rangle^2 + \sum_{i=3}^{n} \left(\lambda_i(\widehat{\mathbf{L}}) - \lambda_2(\mathbf{L})\right)^2 \langle \widehat{\boldsymbol{u}_i}, \boldsymbol{u}_2 \rangle^2$$

$$\ge \sum_{i=3}^{n} \left(\lambda_3(\widehat{\mathbf{L}}) - \lambda_2(\mathbf{L})\right)^2 \langle \widehat{\boldsymbol{u}_i}, \boldsymbol{u}_2 \rangle^2 = \left(\lambda_3(\widehat{\mathbf{L}}) - \lambda_2(\mathbf{L})\right)^2 \left(1 - \langle \widehat{\boldsymbol{u}_2}, \boldsymbol{u}_2 \rangle^2\right),$$

which rearranges to

$$\langle \widehat{\boldsymbol{u}_2}, \boldsymbol{u}_2 \rangle^2 \ge 1 - \left(\frac{\left\|(\widehat{\mathbf{L}} - \mathbf{L})\boldsymbol{u}_2\right\|_2}{\lambda_3(\widehat{\mathbf{L}}) - \lambda_2(\mathbf{L})}\right)^2.$$

Now, if $\left\|(\widehat{\mathbf{L}} - \mathbf{L})\boldsymbol{u}_2\right\|_2 \ge |\lambda_3(\widehat{\mathbf{L}}) - \lambda_2(\mathbf{L})|$, then the condition $\|\boldsymbol{u}_2 - \widehat{\boldsymbol{u}_2}\|_2 \le \sqrt{2} \cdot \frac{\left\|(\widehat{\mathbf{L}} - \mathbf{L})\boldsymbol{u}_2\right\|_2}{|\lambda_3(\widehat{\mathbf{L}}) - \lambda_2(\mathbf{L})|}$ is trivially satisfied, since $\|\boldsymbol{u}_2 - \widehat{\boldsymbol{u}_2}\|_2 \le \sqrt{2 - 2 \langle \widehat{\boldsymbol{u}_2}, \boldsymbol{u}_2 \rangle} \le \sqrt{2}$. Otherwise, taking the square roots of both sides, we obtain

$$\langle \widehat{\boldsymbol{u}_2}, \boldsymbol{u}_2 \rangle \ge \sqrt{1 - \left(\frac{\left\|(\widehat{\mathbf{L}} - \mathbf{L})\boldsymbol{u}_2\right\|_2}{\lambda_3(\widehat{\mathbf{L}}) - \lambda_2(\mathbf{L})}\right)^2},$$

which gives

$$\|\widehat{\boldsymbol{u}_2} - \boldsymbol{u}_2\|_2^2 = 2 - 2 \langle \widehat{\boldsymbol{u}_2}, \boldsymbol{u}_2 \rangle \le 2 - 2\sqrt{1 - \left(\frac{\left\|(\widehat{\mathbf{L}} - \mathbf{L})\boldsymbol{u}_2\right\|_2}{\lambda_3(\widehat{\mathbf{L}}) - \lambda_2(\mathbf{L})}\right)^2} \le 2 \cdot \left(\frac{\left\|(\widehat{\mathbf{L}} - \mathbf{L})\boldsymbol{u}_2\right\|_2}{\lambda_3(\widehat{\mathbf{L}}) - \lambda_2(\mathbf{L})}\right)^2.$$

Taking the square root of both sides and repeating this argument by exchanging the roles of $\mathbf{L}$ and $\widehat{\mathbf{L}}$ yields the statement of Lemma A.15. $\square$

This immediately implies the following upper-bound on $\|\boldsymbol{u}_2 - \boldsymbol{u}_2^\star\|_2$. We will use it repeatedly, both in Theorem 1 and Theorem 2.

**Lemma A.16.** *We have*

$$\|\boldsymbol{u}_2 - \boldsymbol{u}_2^\star\|_2 \le \sqrt{2} \cdot \frac{\|\mathbf{E}\boldsymbol{u}_2^\star\|_2}{|\lambda_3 - \lambda_2^\star|}.$$

*Proof.* Lemma A.16 immediately follows from Lemma A.15 by letting $\widehat{\mathbf{L}} = \mathbf{L}^\star$. $\square$

Combining with Lemma A.11 and Lemma A.12, we can now upper-bound $\|\boldsymbol{u}_2 - \boldsymbol{u}_2^\star\|_2$ in the setting of Theorem 1.

**Lemma A.17.** *In the setting of Theorem 1, there exists a universal constant $C$ such that, for $\delta \geq 3n^{-10}$, with probability $\geq 1 - \delta$, we have*

$$\|\boldsymbol{u}_2 - \boldsymbol{u}_2^\star\|_2 \leq \frac{C}{\sqrt{\log(n/\delta)}}.$$

*Proof of Lemma A.17.* Using Lemma A.16, Lemma A.11 and Lemma A.12, we have

$$\|\boldsymbol{u}_2 - \boldsymbol{u}_2^\star\|_2 \leq \frac{400\sqrt{2}C_{A.12}\left(\sqrt{nq} + (nq\log(3n/\delta))^{1/4} + \sqrt{\log(3n/\delta)}\right)}{n(p-q)}.$$

At this point, it is enough to show that there exists a universal constant $C$ such that

$$Cn(p-q) \geq 400\sqrt{2}C_{A.12}\left(\sqrt{nq\log(n/\delta)} + (nq)^{1/4}\left(\log(n/\delta)\right)^{3/4} + \log(n/\delta)\right).$$

To see this, note that for any two nonnegative real numbers we have $2a^{1/4}b^{1/4} \leq \sqrt{b} + \sqrt{a}$, which implies $2a^{1/4}b^{3/4} \leq b + \sqrt{ab}$. Let $a = nq$ and $b = \log(3n/\delta)$, and we get

$$400\sqrt{2}C_{A.12}\left(\sqrt{nq\log(3n/\delta)} + (nq)^{1/4}\left(\log(n/\delta)\right)^{3/4} + \log(3n/\delta)\right)$$

$$\leq 800\sqrt{2}C_{A.12}\left(\sqrt{nq\log(3n/\delta)} + \log(3n/\delta)\right)$$

$$\leq 800\sqrt{2}C_{A.12}\left(\sqrt{n\overline{p}\log(3n/\delta)} + \log(3n/\delta)\right) \leq Cn(p-q),$$

where the last inequality follows from the assumption we gave in Theorem 1. We therefore conclude the proof of Lemma A.17. $\qquad\square$

Next, we prove $\ell_1$ norm concentration for the rows of $\mathbf{A}$ and for the rows of $\mathbf{L}$ in the setting of Theorem 1. We will use this in Lemma A.19, where we will bound $\left\|\boldsymbol{u}_2^{(v)} - \boldsymbol{u}_2\right\|_2$. Here $\boldsymbol{u}_2^{(v)}$ denotes the second eigenvector of the leave-one-out Laplacian $\mathbf{L}^{(v)}$.

**Lemma A.18.** *In the setting of Theorem 1, there exists a universal constant $C$ such that with probability $\geq 1 - \delta$, for all $v \in V$, we have*

$$\|\boldsymbol{a}_v - \boldsymbol{a}_v^\star\|_1 \leq C\left(n\overline{p} + \sqrt{n\overline{p}\log(n/\delta)} + \log(n/\delta)\right)$$

$$\|\boldsymbol{l}_v - \mathbb{E}\left[\boldsymbol{l}_v\right]\|_1 \leq C\left(n\overline{p} + \sqrt{n\overline{p}\log(n/\delta)} + \log(n/\delta)\right).$$

*Proof of Lemma A.18.* It is easy to see that
$$\|\boldsymbol{l}_v - \mathbb{E}\left[\boldsymbol{l}_v\right]\|_1 = |\boldsymbol{d}[v] - \mathbb{E}\left[\boldsymbol{d}[v]\right]| + \|\boldsymbol{a}_v - \boldsymbol{a}_v^\star\|_1.$$
Let us consider the second term above. By Lemma A.4 and Lemma A.3, we have with probability $\geq 1 - \delta/2$ that for all $v \in V$

$$\|\boldsymbol{a}_v - \boldsymbol{a}_v^\star\|_1 \leq \|\boldsymbol{a}_v\|_1 + \|\boldsymbol{a}_v^\star\|_1$$

$$\leq 2\left(\frac{n\overline{p}}{2} + \max\{C_{A.3}, C_{A.4}\}\left(\sqrt{n\overline{p}\log(4n/\delta)} + \log(4n/\delta)\right)\right) + n\overline{p}$$

$$= 2n\overline{p} + 2\max\{C_{A.3}, C_{A.4}\}\left(\sqrt{n\overline{p}\log(4n/\delta)} + \log(4n/\delta)\right).$$

Finally, by Lemma A.3 and Lemma A.4, we have with probability $1 - \delta/2$ that for all $v \in V$,
$$|\boldsymbol{d}[v] - \mathbb{E}\left[\boldsymbol{d}[v]\right]| \leq \max_{v \in V}|\boldsymbol{d}_{\mathsf{out}}[v] - \mathbb{E}\left[\boldsymbol{d}_{\mathsf{out}}[v]\right]| + \max_{v \in V}|\boldsymbol{d}_{\mathsf{in}}[v] - \mathbb{E}\left[\boldsymbol{d}_{\mathsf{in}}[v]\right]|$$

$$\leq 2\max\{C_{A.3}, C_{A.4}\}\left(\sqrt{n\overline{p}\log(4n/\delta)} + \log(4n/\delta)\right)$$

Adding everything up means that with probability $\geq 1 - \delta$, for all $v \in V$, we have

$$\|\boldsymbol{l}_v - \mathbb{E}\left[\boldsymbol{l}_v\right]\|_1 \leq 2n\overline{p} + 4\max\{C_{A.4}, C_{A.3}\}\left(\sqrt{n\overline{p}\log(4n/\delta)} + \log(4n/\delta)\right),$$

which completes the proof of Lemma A.18. $\qquad\square$

Having established Lemma A.18, we can now upper-bound $\left\|\boldsymbol{u}_2^{(v)} - \boldsymbol{u}_2\right\|_2$.

**Lemma A.19.** *In the setting of Theorem 1, for $\delta \geq 2n^{-9}$ with probability $\geq 1 - \delta$, for all $v \in V$, we have*

$$\left\|\boldsymbol{u}_2^{(v)} - \boldsymbol{u}_2\right\|_2 \leq \|\boldsymbol{u}_2\|_\infty \cdot \frac{C\left(\overline{p} + \sqrt{\overline{p}\log\left(n/\delta\right)/n} + \log\left(n/\delta\right)/n\right)}{p - q}$$

*Proof of Lemma A.19.* Recall that the gap condition in Theorem 1 means that $p$ and $q$ are such that for a universal constant $C$,

$$n(p - q) \geq C\left(\sqrt{n\overline{p}\log\left(n/\delta\right)} + \log\left(n/\delta\right)\right).$$

To appeal to Lemma A.15, we need to understand the entries of the matrix $\mathbf{L} - \mathbf{L}^{(v)}$. It is easy to see that this matrix only has nonzero entries on the diagonal and in the $v$th row and column. There, the $v$th row and column of $\mathbf{L} - \mathbf{L}^{(v)}$ are exactly equal to those of $\mathbf{L} - \mathbf{L}^\star$. Moreover, the $w \neq v$th diagonal entry of $\mathbf{L} - \mathbf{L}^{(v)}$ is exactly $\mathbb{1}\left\{(v, w) \in E\right\} - p_{vw}$.

Hence, we have

$$\left\|\left(\mathbf{L} - \mathbf{L}^{(v)}\right)\boldsymbol{u}_2\right\|_2$$

$$= \left(\sum_{w=1}^n \left\langle\left(\mathbf{L} - \mathbf{L}^{(v)}\right)_w, \boldsymbol{u}_2\right\rangle^2\right)^{1/2}$$

$$= \left(\left\langle(\mathbf{L} - \mathbf{L}^\star)_v, \boldsymbol{u}_2\right\rangle^2 + \sum_{w \neq v}\left((\boldsymbol{a}_v[w] - p_{vw})\boldsymbol{u}_2[w] - (\boldsymbol{a}_v[w] - p_{vw})\boldsymbol{u}_2[v]\right)^2\right)^{1/2}$$

$$\leq \left|\left\langle(\mathbf{L} - \mathbf{L}^\star)_v, \boldsymbol{u}_2\right\rangle\right| + \left(\sum_{w \neq v}\left((\boldsymbol{a}_v[w] - p_{vw})\boldsymbol{u}_2[w] - (\boldsymbol{a}_v[w] - p_{vw})\boldsymbol{u}_2[v]\right)^2\right)^{1/2}$$

$$\leq \left(\|\boldsymbol{l}_v - \mathbb{E}\left[\boldsymbol{l}_v\right]\|_1 + 2\|\boldsymbol{a}_v - \boldsymbol{a}_v^\star\|_2\right) \cdot \|\boldsymbol{u}_2\|_\infty$$

$$\leq \left(\|\boldsymbol{l}_v - \mathbb{E}\left[\boldsymbol{l}_v\right]\|_1 + 2\|\boldsymbol{a}_v - \boldsymbol{a}_v^\star\|_1\right) \cdot \|\boldsymbol{u}_2\|_\infty$$

$$\leq \|\boldsymbol{u}_2\|_\infty \cdot 3C_{A.18}\left(n\overline{p} + \sqrt{n\overline{p}\log\left(2n^2/\delta\right)} + \log\left(2n^2/\delta\right)\right).$$

Now, let $C \geq 8C_{A.10}$. Using Lemma A.10 to understand the concentration of sampling the graph except edges incident to $v$, along with Weyl's inequality, we have with probability $\geq 1 - \delta$ that for all $v \in V$ and for all $n$ sufficiently large,

$$\left|\lambda_3^{(v)} - \lambda_2\right| \geq \left(\lambda_3^{(v)} - \lambda_3^\star\right) - (\lambda_2 - \lambda_2^\star) + (\lambda_3^\star - \lambda_2^\star)$$

$$\geq -2\left(C_{A.10}\sqrt{n\overline{p}\log\left(2n^2/\delta\right)} + \log\left(2n^2/\delta\right)\right) + \frac{n(p - q)}{2} \geq \frac{n(p - q)}{4}.$$

Now, using Lemma A.15, we get

$$\left\|\boldsymbol{u}_2^{(v)} - \boldsymbol{u}_2\right\|_2 \leq \frac{\left\|\left(\mathbf{L} - \mathbf{L}^{(v)}\right)\boldsymbol{u}_2\right\|_2}{\left|\lambda_3^{(v)} - \lambda_2\right|} \leq \|\boldsymbol{u}_2\|_\infty \cdot \frac{12C_{A.18}\left(n\overline{p} + \sqrt{n\overline{p}\log\left(n/\delta\right)} + \log\left(n/\delta\right)\right)}{n(p - q)}$$

$$\leq \|\boldsymbol{u}_2\|_\infty \cdot \frac{12C_{A.18}\left(\overline{p} + \sqrt{\overline{p}\log\left(2n^2/\delta\right)/n} + \log\left(2n^2/\delta\right)/n\right)}{p - q},$$

completing the proof of Lemma A.19. $\qquad\square$

## A.5 Leave-one-out and bootstrap

The main goal of this section is to establish an upper-bound on $|\langle \boldsymbol{a}_v - \boldsymbol{a}_v^\star, \boldsymbol{u}_2 - \boldsymbol{u}_2^\star \rangle|$ in the setting of Theorem 1. To this end, we will need the following concentration inequality from [AFWZ20].

**Lemma A.20** (Lemma 7 from [AFWZ20]). *Let $\boldsymbol{w} \in \mathbb{R}^n$ and $X_i \sim \mathsf{Ber}(p_i)$. Let $p \geq p_i$ for all $i \in [n]$. Let $X \in \mathbb{R}^n$ be the vector formed by stacking the $X_i$. Then,*

$$
\Pr\left[ |\langle \boldsymbol{w}, X - \mathbb{E}\left[X\right] \rangle| \geq \frac{(2+a)pn}{\max\left(1, \log\left(\frac{\sqrt{n}\|\boldsymbol{w}\|_\infty}{\|\boldsymbol{w}\|_2}\right)\right)} \cdot \|\boldsymbol{w}\|_\infty \right] \leq 2\exp\left(-anp\right).
$$

**Lemma A.21.** *In the setting of Theorem 1, suppose $\boldsymbol{a}_v$ is such that $\boldsymbol{a}_v[w] \sim \mathsf{Bernoulli}(p_{vw})$ and let $\overline{p} \geq \max_{w:\, p_{vw} \neq 1} p_{vw}$. With probability $\geq 1 - \delta$ for $\delta \geq 1/n^2$, for all $v \in V$, we have*

$$
|\langle \boldsymbol{a}_v - \boldsymbol{a}_v^\star, \boldsymbol{u}_2 - \boldsymbol{u}_2^\star \rangle| \leq C\left(n\overline{p} + \log\left(n/\delta\right)\right)\left(\frac{\|\boldsymbol{u}_2\|_\infty}{\log\log n} + \frac{1}{\sqrt{n}\log\log n}\right).
$$

*Proof of Lemma A.21.* Ideally, one would treat $\boldsymbol{u}_2 - \boldsymbol{u}_2^\star$ as fixed and then apply Bernstein's inequality to argue that the sum of centered Bernoulli random variables as written above concentrates well. Unfortunately, since $\boldsymbol{u}_2$ depends on $\boldsymbol{a}_v - \boldsymbol{a}_v^\star$, we cannot express this inner product as the sum of independent random variables.

To resolve this, we use the leave-one-out method. Let $\boldsymbol{u}_2^{(v)}$ be the second eigenvector of the leave-one-out Laplacian $\mathbf{L}^{(v)}$ of $\mathbf{A}^{(v)}$, where $\mathbf{A}^{(v)}$ is chosen to agree with $\mathbf{A}$ everywhere except for the $v$th row and $v$th column. The $v$th row and $v$th column of $\mathbf{A}^{(v)}$ are replaced with those of $\mathbf{A}^\star$. Now, $\boldsymbol{a}_v$ does not depend on $\mathbf{L}^{(v)}$ and therefore $\boldsymbol{u}_2^{(v)}$.

We therefore write

$$
\begin{aligned}
|\langle \boldsymbol{a}_v - \boldsymbol{a}_v^\star, \boldsymbol{u}_2 - \boldsymbol{u}_2^\star \rangle| &\leq \left|\left\langle \boldsymbol{a}_v - \boldsymbol{a}_v^\star, \boldsymbol{u}_2 - \boldsymbol{u}_2^{(v)} \right\rangle\right| + \left|\left\langle \boldsymbol{a}_v - \boldsymbol{a}_v^\star, \boldsymbol{u}_2^{(v)} - \boldsymbol{u}_2^\star \right\rangle\right| \\
&\leq \|\boldsymbol{a}_v - \boldsymbol{a}_v^\star\|_2 \cdot \left\|\boldsymbol{u}_2^{(v)} - \boldsymbol{u}_2\right\|_2 + \left|\left\langle \boldsymbol{a}_v - \boldsymbol{a}_v^\star, \boldsymbol{u}_2^{(v)} - \boldsymbol{u}_2^\star \right\rangle\right| \\
&\leq \|\boldsymbol{a}_v - \boldsymbol{a}_v^\star\|_2 \cdot \frac{C_{A.19}\overline{p}}{p-q}\|\boldsymbol{u}_2\|_\infty + \left|\left\langle \boldsymbol{a}_v - \boldsymbol{a}_v^\star, \boldsymbol{u}_2^{(v)} - \boldsymbol{u}_2^\star \right\rangle\right|.
\end{aligned}
$$

To bound the rightmost term of the RHS, we use Lemma 7 of [AFWZ20], reproduced in Lemma A.20. In that, let $\boldsymbol{w} := \boldsymbol{u}_2^{(v)} - \boldsymbol{u}_2^\star$. Let $a = \frac{1}{n\overline{p}}\log\left(20n/\delta\right)$ so that $2\exp(-2an\overline{p}) \leq \delta/(10n)$. Note that for the deterministic entries, we have $\boldsymbol{a}_v - \boldsymbol{a}_v^\star = 1 - 1 = 0$, so in Lemma A.20, we can set $X_w \sim \mathsf{Ber}(0)$ for these entries. Now, by Lemma A.20, with probability $\geq 1 - \delta/n$, we have

$$
\left|\left\langle \boldsymbol{u}_2^{(v)} - \boldsymbol{u}_2^\star, \boldsymbol{a}_v - \boldsymbol{a}_v^\star \right\rangle\right| \leq \frac{2n\overline{p} + \log\left(\frac{20n}{\delta}\right)}{\max\left(1, \log\left(\frac{\sqrt{n}\|\boldsymbol{w}\|_\infty}{\|\boldsymbol{w}\|_2}\right)\right)} \cdot \|\boldsymbol{w}\|_\infty. \tag{5}
$$

Let us first bound $\|\boldsymbol{w}\|_\infty = \left\|\boldsymbol{u}_2^{(v)} - \boldsymbol{u}_2^\star\right\|_\infty$. We write

$$
\left\|\boldsymbol{u}_2^{(v)} - \boldsymbol{u}_2^\star\right\|_\infty \leq \left\|\boldsymbol{u}_2^{(v)} - \boldsymbol{u}_2\right\|_\infty + \|\boldsymbol{u}_2 - \boldsymbol{u}_2^\star\|_\infty \tag{6}
$$

$$
\leq \left\|\boldsymbol{u}_2^{(v)} - \boldsymbol{u}_2\right\|_2 + \|\boldsymbol{u}_2\|_\infty + \|\boldsymbol{u}_2^\star\|_\infty \tag{7}
$$

$$
\leq 2\max\left\{C_{A.19}(\alpha, \delta)\|\boldsymbol{u}_2\|_\infty, \frac{1}{\sqrt{n}}\right\}. \tag{8}
$$

In what follows, we omit the arguments $\alpha$ and $\delta$ in mentions of $C_{A.19}$. Next, using Lemma A.17, the triangle inequality, and $\delta \geq 1/n^3$, we have

$$
\|\boldsymbol{w}\|_2 = \left\|\boldsymbol{u}_2^{(v)} - \boldsymbol{u}_2^\star\right\|_2 \leq C_{A.19}\|\boldsymbol{u}_2\|_\infty + \frac{4C_{A.17}}{\sqrt{\log n}}.
$$

We now have two cases based on the value of $\sqrt{n} \cdot \frac{\left\|\boldsymbol{u}_2^{(v)} - \boldsymbol{u}_2^\star\right\|_\infty}{\left\|\boldsymbol{u}_2^{(v)} - \boldsymbol{u}_2^\star\right\|_2}$.

**Case 1 – $w$ is not too "flat."**    Let us first handle the case where

$$\frac{\sqrt{n} \cdot \left\|\boldsymbol{u}_2^{(v)} - \boldsymbol{u}_2^\star\right\|_\infty}{\left\|\boldsymbol{u}_2^{(v)} - \boldsymbol{u}_2^\star\right\|_2} \geq \sqrt{\log n}.$$

We plug this into (5) and get

$$\left|\left\langle \boldsymbol{u}_2^{(v)} - \boldsymbol{u}_2^\star, \boldsymbol{a}_v - \boldsymbol{a}_v^\star \right\rangle\right| \leq \frac{2n\overline{p} + \log\left(\frac{20n}{\delta}\right)}{\max\left(1, \log\left(\frac{\sqrt{n}\|\boldsymbol{w}\|_\infty}{\|\boldsymbol{w}\|_2}\right)\right)} \cdot \|\boldsymbol{w}\|_\infty$$

$$\leq 4 \cdot \frac{n\overline{p} + \log\left(20n/\delta\right)}{\log\log n} \left(C_{A.19} \|\boldsymbol{u}_2\|_\infty + \frac{1}{\sqrt{n}}\right),$$

where the last inequality follows from (8).

**Case 2 – $w$ is "flat."**    We now assume

$$\frac{\sqrt{n} \cdot \left\|\boldsymbol{u}_2^{(v)} - \boldsymbol{u}_2^\star\right\|_\infty}{\left\|\boldsymbol{u}_2^{(v)} - \boldsymbol{u}_2^\star\right\|_2} \leq \sqrt{\log n}.$$

We can easily check that the function

$$\frac{x}{\max\left(1, \log x\right)}$$

is increasing, so its maximum will be attained at the largest value of $x$ in the domain. Let $x = \sqrt{n} \|\boldsymbol{w}\|_\infty / \|\boldsymbol{w}\|_2$ and write

$$\frac{2n\overline{p} + \log\left(\frac{20n}{\delta}\right)}{\max\left(1, \log\left(\frac{\sqrt{n}\|\boldsymbol{w}\|_\infty}{\|\boldsymbol{w}\|_2}\right)\right)} \cdot \|\boldsymbol{w}\|_\infty$$

$$= \frac{2n\overline{p} + \log\left(\frac{20n}{\delta}\right)}{\max\left(1, \log\left(\frac{\sqrt{n}\|\boldsymbol{w}\|_\infty}{\|\boldsymbol{w}\|_2}\right)\right)} \cdot \frac{\sqrt{n}\|\boldsymbol{w}\|_\infty}{\|\boldsymbol{w}\|_2} \cdot \frac{\|\boldsymbol{w}\|_2}{\sqrt{n}}$$

$$\leq \frac{2n\overline{p} + \log\left(\frac{20n}{\delta}\right)}{\log\log n} \cdot \sqrt{\frac{\log n}{n}} \cdot \|\boldsymbol{w}\|_2$$

$$\leq \frac{2n\overline{p} + \log\left(\frac{20n}{\delta}\right)}{\log\log n} \cdot \sqrt{\frac{\log n}{n}} \cdot C_{A.19} \left(\|\boldsymbol{u}_2\|_\infty + \frac{1}{\sqrt{\log\left(20n^2/\delta\right)}}\right)$$

$$= C_{A.19} \left(\frac{2n\overline{p} + \log\left(\frac{20n}{\delta}\right)}{\log\log n} \cdot \sqrt{\frac{\log n}{n}} \|\boldsymbol{u}_2\|_\infty + \frac{2n\overline{p} + \log\left(\frac{20n}{\delta}\right)}{\sqrt{n} \cdot \log\log n}\right).$$

All of this tells us that

$$\left|\left\langle \boldsymbol{a}_v - \boldsymbol{a}_v^\star, \boldsymbol{u}_2^{(v)} - \boldsymbol{u}_2^\star \right\rangle\right| \leq 4C_{A.19} \cdot \left(n\overline{p} + \log\left(20n/\delta\right)\right) \left(\frac{\|\boldsymbol{u}_2\|_\infty}{\log\log n} + \frac{1}{\sqrt{n}\log\log n}\right).$$

It remains to handle the term

$$\|\boldsymbol{a}_v - \boldsymbol{a}_v^\star\|_2 \cdot \|\boldsymbol{u}_2\|_\infty.$$

Indeed, using Lemma A.13, we have with probability $\geq 1 - \delta$ that

$$\|\boldsymbol{a}_v - \boldsymbol{a}_v^\star\|_2 \cdot \|\boldsymbol{u}_2\|_\infty \leq C_{A.13} \left(\frac{\log\left(20n/\delta\right)}{\log n}\right)^{3/2} \sqrt{n\overline{p}} \cdot \|\boldsymbol{u}_2\|_\infty.$$

Combining everything tells us that

$$|\langle \boldsymbol{a}_v - \boldsymbol{a}_v^\star, \boldsymbol{u}_2 - \boldsymbol{u}_2^\star \rangle| \leq 30 C_{A.13} \left( \frac{\log (20n/\delta)}{\log n} \right)^{3/2} \sqrt{n\overline{p}} \cdot \|\boldsymbol{u}_2\|_\infty$$

$$+ 4C_{A.19} \cdot (n\overline{p} + \log (20n/\delta)) \left( \frac{\|\boldsymbol{u}_2\|_\infty}{\log \log n} + \frac{1}{\sqrt{n} \log \log n} \right)$$

$$\leq C (n\overline{p} + \log (20n/\delta)) \left( \frac{\|\boldsymbol{u}_2\|_\infty}{\log \log n} + \frac{1}{\sqrt{n} \log \log n} \right).$$

Taking a union bound over all $v \in V$ concludes the proof of Lemma A.21. $\square$

Finally, we establish an upper-bound on $\|\boldsymbol{u}_2\|_\infty$. This will be used repeatedly in the proof of Theorem 1.

**Lemma A.22.** *In the same setting as Theorem 1, with probability $\geq 1 - \delta$ for $\delta \geq 10n^2$, we have for some constant $C(\alpha, \delta)$ that*

$$\|\boldsymbol{u}_2\|_\infty \leq \frac{C(\alpha, \delta)}{\sqrt{n}}.$$

*Proof of Lemma A.22.* First, observe that

$$(\mathbf{D} - \mathbf{A})\boldsymbol{u}_2 = \lambda_2 \boldsymbol{u}_2,$$

which means that

$$(\mathbf{D} - \lambda_2 \mathbf{I})^{-1} \mathbf{A}\boldsymbol{u}_2 = \boldsymbol{u}_2.$$

By Lemma A.7, with probability $\geq 1 - \delta$, for all $v \in V$ we have

$$\boldsymbol{d}[v] - \lambda_2 \geq \frac{n(p - q)}{4}.$$

Combining with Lemma A.6, we have

$$\frac{\boldsymbol{d}_{\text{in}}[v] - \boldsymbol{d}_{\text{out}}[v]}{\boldsymbol{d}_{\text{in}}[v] - \boldsymbol{d}_{\text{out}}[v] + (2\boldsymbol{d}_{\text{out}}[v] - \lambda_2)} = 1 - \frac{2\boldsymbol{d}_{\text{out}}[v] - \lambda_2}{\boldsymbol{d}_{\text{in}}[v] - \boldsymbol{d}_{\text{out}}[v] + (2\boldsymbol{d}_{\text{out}}[v] - \lambda_2)}$$

$$\leq 1 + \frac{C_{A.6} \left( \sqrt{nq \log (10n/\delta)} + \log (10n/\delta) \right)}{\boldsymbol{d}_{\text{in}}[v] - \boldsymbol{d}_{\text{out}}[v] + (2\boldsymbol{d}_{\text{out}}[v] - \lambda_2)}$$

$$\leq 1 + \frac{4C_{A.6} \left( \sqrt{nq \log (10n/\delta)} + \log (10n/\delta) \right)}{n(p - q)} \leq C',$$

for some constant $C' > 0$, where the penultimate line follows from Lemma A.7 and the last line follows from the gap assumption in Theorem 1. Furthermore, by Lemma A.8 and Lemma A.17, we have with probability $\geq 1 - \delta$ that for all $v \in V$,

$$\frac{|\langle \boldsymbol{a}_v^\star, \boldsymbol{u}_2^\star - \boldsymbol{u}_2 \rangle|}{\boldsymbol{d}[v] - \lambda_2} \leq \frac{\overline{p}\sqrt{n}}{\boldsymbol{d}[v] - \lambda_2} \cdot \frac{C_{A.17}}{\sqrt{\log (10n/\delta)}} \leq \frac{C_{A.8}(\alpha) \cdot C_{A.17}}{\sqrt{n \log (10n/\delta)}}.$$

Now, using Lemma A.8 (and using Lemma A.7 to ensure that $\boldsymbol{d}[v] - \lambda_2 > 0$ for all $v \in V$), we have

$$
\begin{aligned}
\|\boldsymbol{u}_2\|_\infty &= \left\|(\mathbf{D} - \lambda_2\mathbf{I})^{-1}\mathbf{A}\boldsymbol{u}_2\right\|_\infty \\
&= \left\|(\mathbf{D} - \lambda_2\mathbf{I})^{-1}\mathbf{A}\boldsymbol{u}_2 - (\mathbf{D} - \lambda_2\mathbf{I})^{-1}\mathbf{A}\boldsymbol{u}_2^\star + (\mathbf{D} - \lambda_2\mathbf{I})^{-1}\mathbf{A}\boldsymbol{u}_2^\star\right\|_\infty \\
&\leq \left\|(\mathbf{D} - \lambda_2\mathbf{I})^{-1}\mathbf{A}\boldsymbol{u}_2^\star\right\|_\infty + \left\|(\mathbf{D} - \lambda_2\mathbf{I})^{-1}\mathbf{A}(\boldsymbol{u}_2^\star - \boldsymbol{u}_2)\right\|_\infty \\
&= \max_{1 \leq v \leq n} \frac{|\langle \boldsymbol{a}_v, \boldsymbol{u}_2^\star \rangle|}{\boldsymbol{d}[v] - \lambda_2} + \max_{1 \leq v \leq n} \frac{|\langle \boldsymbol{a}_v, \boldsymbol{u}_2^\star - \boldsymbol{u}_2 \rangle|}{\boldsymbol{d}[v] - \lambda_2} \\
&= \frac{1}{\sqrt{n}}\left(\max_{1 \leq v \leq n} \frac{|\boldsymbol{d}_{\mathsf{in}}[v] - \boldsymbol{d}_{\mathsf{out}}[v]|}{\boldsymbol{d}[v] - \lambda_2}\right) + \max_{1 \leq v \leq n} \frac{|\langle \boldsymbol{a}_v, \boldsymbol{u}_2^\star - \boldsymbol{u}_2 \rangle|}{\boldsymbol{d}[v] - \lambda_2} \\
&\leq \frac{C}{\sqrt{n}} + \max_{1 \leq v \leq n} \frac{|\langle \boldsymbol{a}_v - \boldsymbol{a}_v^\star, \boldsymbol{u}_2^\star - \boldsymbol{u}_2 \rangle|}{\boldsymbol{d}[v] - \lambda_2} + \max_{1 \leq v \leq n} \frac{|\langle \boldsymbol{a}_v^\star, \boldsymbol{u}_2^\star - \boldsymbol{u}_2 \rangle|}{\boldsymbol{d}[v] - \lambda_2} \\
&\leq \frac{C}{\sqrt{n}} + \frac{C_{A.21}\left(n\bar{p} + \log\left(\frac{10n}{\delta}\right)\right)}{\boldsymbol{d}[v] - \lambda_2} \cdot \left(\frac{1}{\sqrt{n}\log\log n} + \frac{\|\boldsymbol{u}_2\|_\infty}{\log\log n}\right) + \frac{C_{A.8}(\alpha) \cdot C_{A.17}}{\sqrt{n}\log\left(\frac{10n}{\delta}\right)} \\
&\leq \frac{C}{\sqrt{n}} + C_{A.21} \cdot C_{A.8}(\alpha) \cdot \left(\frac{1}{\sqrt{n}\log\log n} + \frac{\|\boldsymbol{u}_2\|_\infty}{\log\log n}\right) + \frac{C_{A.8}(\alpha) \cdot C_{A.17}}{\sqrt{n}\log\left(\frac{10n}{\delta}\right)}.
\end{aligned}
$$

Note that any $n$ large enough

$$
\frac{C_{A.21} \cdot C_{A.8}(\alpha) \cdot \|\boldsymbol{u}_2\|_\infty}{\log\log n} \leq \frac{\|\boldsymbol{u}_2\|_\infty}{2}.
$$

Thus, rearranging and solving for $\|\boldsymbol{u}_2\|_\infty$ yields

$$
\|\boldsymbol{u}_2\|_\infty \leq 2\left(\frac{C}{\sqrt{n}} + C_{A.21} \cdot C_{A.8}(\alpha) \cdot \left(\frac{1}{\sqrt{n}\log\log n}\right) + \frac{C_{A.8}(\alpha) \cdot C_{A.17}}{\sqrt{n}\log\left(\frac{10n}{\delta}\right)}\right),
$$

completing the proof of Lemma A.22. □

## A.6 Strong consistency of unnormalized spectral bisection

In this section, we prove our main positive results Theorem 1 and Theorem 2. It will be helpful to recall the proof sketches given in Section 3 while reading this section.

At a high level, the proof plan is as follows.

1. We first establish a sufficient condition for a particular vertex to be classified correctly. We can think of this as simultaneously showing that the intermediate estimator $(\mathbf{D} - \lambda_2\mathbf{I})^{-1}\mathbf{A}\boldsymbol{u}_2^\star$ is strongly consistent and that the corresponding "noise" term $(\mathbf{D} - \lambda_2\mathbf{I})^{-1}\mathbf{A}(\boldsymbol{u}_2^\star - \boldsymbol{u}_2)$ is a lower-order term in comparison to this. For a more formal way to see this, see Lemma A.23.

2. For the proof of Theorem 1, the main technical challenge in showing that the noise term above is small amounts to analyzing the random quantity $|\langle \boldsymbol{a}_v, \boldsymbol{u}_2^\star - \boldsymbol{u}_2 \rangle|$. This is where we will have to use the leave-one-out method to decouple the dependence between $\boldsymbol{a}_v$ and $\boldsymbol{u}_2$. The relevant lemmas for the leave-one-out analysis are Lemma A.21 and Lemma A.22.

3. Finally, for the proof of Theorem 2, we again appeal to Lemma A.23 but use a different approach to show that the noise term is small.

### A.6.1 A sufficient condition for exact recovery and proof

The main result of this subsection is Lemma A.23, which gives a general condition under which a particular vertex will be classified correctly. The proofs of Theorem 1 and Theorem 2 will follow by invoking Lemma A.23. We remark that the point of this lemma is mostly conceptual; the crux of the analysis lies in establishing that these conditions are satisfied our models.

**Lemma A.23.** *Let $v \in V$ be some vertex. If $\boldsymbol{d}[w] - \lambda_2 > 0$ for all $w \in V$, $\boldsymbol{d}_{\mathsf{in}}[v] > \boldsymbol{d}_{\mathsf{out}}[v]$, and $|\langle \boldsymbol{a}_v, \boldsymbol{u}_2^\star - \boldsymbol{u}_2 \rangle| \leq (\boldsymbol{d}_{\mathsf{in}}[v] - \boldsymbol{d}_{\mathsf{out}}[v])/\sqrt{n}$, then $\mathsf{sign}\left(\boldsymbol{u}_2[v]\right) = \mathsf{sign}\left(\boldsymbol{u}_2^\star[v]\right)$, i.e., $\boldsymbol{u}_2$ correctly classifies vertex $v$.*

The goal of the rest of this section is to prove Lemma A.23.

Our approach is to study the intermediate estimator

$$(\mathbf{D} - \lambda_2 \mathbf{I})^{-1} \mathbf{A} \boldsymbol{u}_2^\star.$$

At a high level, our goal is to show that this correctly classifies all the vertices with high probability and also is very close to $\boldsymbol{u}_2$ in $\ell_\infty$ norm with high probability. Deng, Ling, and Strohmer [DLS21] used this intermediate estimator to prove the strong consistency of unnormalized spectral bisection for $\mathsf{SBM}(n, p, q)$ instances.

Next, we show that this estimator is consistent and prove Lemma A.23.

*Proof of Lemma A.23.* Observe that

$$\boldsymbol{u}_2 = (\mathbf{D} - \lambda_2 \mathbf{I})^{-1} \mathbf{A} \boldsymbol{u}_2^\star - (\mathbf{D} - \lambda_2 \mathbf{I})^{-1} \mathbf{A} (\boldsymbol{u}_2^\star - \boldsymbol{u}_2).$$

Without loss of generality, suppose $v \in P_1$. In particular, this means that $\boldsymbol{u}_2^\star[v] = 1/\sqrt{n}$. Our goal is to show that $\boldsymbol{u}_2[v] > 0$. And, as per the above, this means that it is enough to show that

$$\left( (\mathbf{D} - \lambda_2 \mathbf{I})^{-1} \mathbf{A} \boldsymbol{u}_2^\star \right)[v] \geq \left( (\mathbf{D} - \lambda_2 \mathbf{I})^{-1} \mathbf{A} (\boldsymbol{u}_2^\star - \boldsymbol{u}_2) \right)[v],$$

or equivalently, using the fact that $\boldsymbol{d}[v] - \lambda_2 > 0$,

$$\langle \boldsymbol{a}_v, \boldsymbol{u}_2^\star \rangle \geq \langle \boldsymbol{a}_v, \boldsymbol{u}_2^\star - \boldsymbol{u}_2 \rangle,$$

where $\boldsymbol{a}_v$ denotes the $v$-th row of $A$. To see that the above holds, use the fact that we know that $\boldsymbol{d}_{\mathsf{in}}[v] - \boldsymbol{d}_{\mathsf{out}}[v] > 0$, which gives

$$\langle \boldsymbol{a}_v, \boldsymbol{u}_2^\star \rangle = \frac{\boldsymbol{d}_{\mathsf{in}}[v] - \boldsymbol{d}_{\mathsf{out}}[v]}{\sqrt{n}} \geq |\langle \boldsymbol{a}_v, \boldsymbol{u}_2^\star - \boldsymbol{u}_2 \rangle| \geq \langle \boldsymbol{a}_v, \boldsymbol{u}_2^\star - \boldsymbol{u}_2 \rangle.$$

This is exactly what we needed, and we conclude the proof of Lemma A.23. □

## A.7 Proofs of main results

At this point, we are ready to prove our main results.

### A.7.1 Nonhomogeneous symmetric stochastic block model (Proof of Theorem 1)

We are finally ready to prove Theorem 1. For convenience, we reproduce its statement here.

**Theorem 1.** *Let $p, \overline{p}, q$ be probabilities such that $q < p \leq \overline{p}$ and such that $\alpha := \overline{p}/(p - q)$ is an arbitrary constant. Let $\mathcal{D} \in \mathsf{NSSBM}(n, p, \overline{p}, q)$. Let $n \geq N(\alpha)$ where the function $N(\alpha)$ only depends on $\alpha$. There exists a universal constant $C > 0$ such that if*

$$n(p - q) \geq C \left( \sqrt{n\overline{p} \log n} + \log n \right), \qquad \text{(gap condition)}$$

*then unnormalized spectral bisection is strongly consistent on $\mathcal{D}$.*

*Proof of Theorem 1.* As mentioned in Section 2, we actually prove a slightly stronger statement – we will allow the adversary to set at most $n\overline{p}/\log \log n$ of the $p_{vw}$ to 1 per vertex $v$ (in other words, the adversary can commit to at most $n\overline{p}/\log \log n$ edges per vertex that are guaranteed to appear in the final graph).

Our plan is to apply Lemma A.23. In order to do so, we start with showing that for all $v$, we have $\boldsymbol{d}_{\mathsf{in}}[v] > \boldsymbol{d}_{\mathsf{out}}[v]$. By Lemma A.5, with probability $\geq 1 - \delta$, we have for all $v \in V$ that

$$\boldsymbol{d}_{\mathsf{in}}[v] - \boldsymbol{d}_{\mathsf{out}}[v] \geq \frac{n(p - q)}{2} - C_{A.5} \left( \sqrt{np \log (n/\delta)} + \log (n/\delta) \right) > 0.$$

Additionally, by Lemma A.7, we have for all $v$ that $\boldsymbol{d}[v] > \lambda_2$.

The final item we need is to show that for all $v \in V$, we have $|\langle \boldsymbol{a}_v, \boldsymbol{u}_2^\star - \boldsymbol{u}_2 \rangle| \leq |\langle \boldsymbol{a}_v, \boldsymbol{u}_2^\star \rangle|$. Observe that

$$|\langle \boldsymbol{a}_v, \boldsymbol{u}_2^\star - \boldsymbol{u}_2 \rangle| \leq |\langle \boldsymbol{a}_v^\star, \boldsymbol{u}_2^\star - \boldsymbol{u}_2 \rangle| + |\langle \boldsymbol{a}_v - \boldsymbol{a}_v^\star, \boldsymbol{u}_2^\star - \boldsymbol{u}_2 \rangle|,$$

where $a_v^\star$ denotes the $v$-th row of $\mathbb{E}[\mathbf{A}]$. We handle the terms one at a time. First, note that by Lemma A.11, with probability $\geq 1 - \delta$, we have

$$\lambda_3 - \lambda_2^\star \geq \frac{n(p-q)}{4}.$$

Now, let $\mathbf{E} := \mathbf{L} - \mathbb{E}[\mathbf{L}]$, and let $a_v^\star[\mathrm{rand}] \in \mathbb{R}^V$ correspond to the vector that entrywise agrees with $a_v^\star$ wherever $a_v^\star$ is not 1 and is zero elsewhere. This corresponds to the edges incident to $v$ that will be sampled randomly from the distribution over graphs. This means that for all $n \geq N(\delta)$ and choosing $\delta \geq 1/(10n)$, we have

$$
\begin{aligned}
|\langle a_v^\star[\mathrm{rand}], u_2^\star - u_2\rangle| &\leq \|a_v^\star[\mathrm{rand}]\|_2 \cdot \frac{\sqrt{2}\,\|\mathbf{E}u_2^\star\|_2}{|\lambda_3 - \lambda_2^\star|} && (\text{ Lemma A.16}) \\
&\leq \overline{p}\sqrt{n} \cdot \frac{40\sqrt{2}C_{A.12}\left(\sqrt{nq} + (nq\log n)^{1/4} + \sqrt{\log n}\right)}{n(p-q)} && (\text{Lemmas A.11 and A.12}) \\
&\leq \frac{1000 C_{A.12} n\overline{p}}{\sqrt{n}\log\log n} && (\text{gap in Theorem 1})
\end{aligned}
$$

To handle the oblivious insertions, let $d_{\mathrm{det}} \in \mathbb{R}^V$ denote the degree vector that counts the number of deterministic edges inserted incident to $v$, for all $v \in V$. Under this notation, we have

$$|\langle a_v^\star - a_v^\star[\mathrm{rand}], u_2^\star - u_2\rangle| \leq d_{\mathrm{det}}[v] \cdot \|u_2 - u_2^\star\|_\infty \leq \frac{n\overline{p}}{\sqrt{n}\log\log n} + \frac{n\overline{p}\,\|u_2\|_\infty}{\log\log n}.$$

where the last inequality follows from using $\|u_2 - u_2^\star\|_\infty \leq \|u_2\|_\infty + \|u_2^\star\|_\infty$. Combining yields

$$|\langle a_v^\star, u_2^\star - u_2\rangle| \leq C'\frac{n\overline{p}}{\sqrt{n}\log\log n} + \frac{n\overline{p}\,\|u_2\|_\infty}{\log\log n},$$

for some constant $C' > 0$. Now, notice that for all $n$ sufficiently large,

$$
\begin{aligned}
&|\langle a_v - a_v^\star, u_2^\star - u_2\rangle| \\
&\leq C_{A.21}\left(n\overline{p} + \log(n/\delta)\right)\left(\frac{\|u_2\|_\infty}{\log\log n} + \frac{1}{\sqrt{n}\log\log n}\right) && (\text{Lemma A.21}) \\
&\leq C_{A.21}\left(n\overline{p} + \log(n/\delta)\right)\left(\frac{\frac{C_{A.22}(\alpha,\delta)}{\sqrt{n}}}{\log\log n} + \frac{1}{\sqrt{n}\log\log n}\right) && (\text{Lemma A.22}) \\
&\leq \frac{C_1(\alpha,\delta)\cdot(n\overline{p} + \log(n/\delta))}{\sqrt{n}\log\log n}.
\end{aligned}
$$

Adding yields for $n \geq N(\alpha,\delta)$,

$$
\begin{aligned}
|\langle a_v, u_2^\star - u_2\rangle| &\leq |\langle a_v^\star, u_2^\star - u_2\rangle| + |\langle a_v - a_v^\star, u_2^\star - u_2\rangle| \\
&\leq \frac{C_2(\alpha,\delta)\cdot(n\overline{p} + \log(n/\delta))}{\sqrt{n}\log\log n} \\
&\leq \frac{1}{\sqrt{n}} \cdot \left(\frac{n(p-q)}{2} - C_{A.5}\left(\sqrt{np\log(n/\delta)} + \log(n/\delta)\right)\right) && (\text{gap condition}) \\
&\leq \frac{d_{\mathrm{in}}[v] - d_{\mathrm{out}}[v]}{\sqrt{n}} = |\langle a_v, u_2^\star\rangle|,
\end{aligned}
$$

which means we satisfy the conditions required by Lemma A.23. Taking a union bound over all our (constantly many) probabilistic statements, setting $\delta = \Theta(1/n)$, and rescaling completes the proof of Theorem 1. $\qquad\square$

### A.7.2 Deterministic clusters model

For convenience, we reproduce the statement of Theorem 2 here.

**Theorem 2.** *Let $q$ be a probability and $d_{in}$ be an integer, and let $\mathcal{D} \in \mathsf{DCM}(n, d_{in}, q)$. For $G \sim \mathcal{D}$, let $\widehat{\mathbf{L}}$ denote the expectation of $\mathbf{L}$ after step (2) but before step (3) in Model 2. There exists constants $C_1, C_2, C_3 > 0$ such that for all $n$ sufficiently large, if*

$$d_{in} \geq C_1 \cdot \left( \frac{nq}{2} + \sqrt{n} \right) \quad and \quad \lambda_3(\widehat{\mathbf{L}}) - \lambda_2(\widehat{\mathbf{L}}) \geq \sqrt{n} + C_2 nq + C_3 \left( \sqrt{nq \log n} + \log n \right),$$

*then unnormalized spectral bisection is strongly consistent on $\mathcal{D}$.*

*Proof of Theorem 2.* In this proof, let $\mathbf{L}^\star$ be the Laplacian matrix that agrees with $\mathbf{L}$ on all internal edges and agrees with $\mathbb{E}[\mathbf{L}]$ on all crossing edges. Let $\mathbf{L}^{(cross)}$ denote the Laplacian matrix corresponding to the cross edges, so we can write $\mathbf{L}^\star = \mathbf{L} - \mathbf{L}^{(cross)} + \mathbb{E}[\mathbf{L}^{(cross)}]$. Although $\mathbf{L}^\star \neq \mathbb{E}[\mathbf{L}]$ due to the adaptive adversary, by Lemma A.14, we still have $\mathbf{L}^\star \boldsymbol{u}_2^\star = \lambda_2^\star \boldsymbol{u}_2^\star = nq \boldsymbol{u}_2^\star$. Moreover, $(\mathbf{L} - \mathbf{L}^\star) \boldsymbol{u}_2^\star$ is the vector whose entries are of the form $2(\boldsymbol{d}_{out}[v] - \mathbb{E}[\boldsymbol{d}_{out}[v]])/\sqrt{n}$. Thus, we will be able to apply Lemma A.16 and Lemma A.12 later on. Finally, observe that $\lambda_i(\mathbf{L}^\star) \geq \lambda_i(\widehat{\mathbf{L}})$ for all $i \geq 3$ and $\lambda_2(\widehat{\mathbf{L}}) = \lambda_2(\mathbf{L}^\star) = nq$. Thus, one can use the spectral gap $\lambda_3(\widehat{\mathbf{L}}) - \lambda_2(\widehat{\mathbf{L}})$ to reason about $\lambda_3^\star - \lambda_2^\star$.

Let $\delta \geq 1/(10n)$. We will apply Lemma A.23 to get strong consistency. First, let us verify that $\boldsymbol{d}[v] > \lambda_2$ for all $v$. Applying Lemma A.10 to the matrix $\mathbf{L}^{(cross)}$ gives

$$\left\| \mathbf{L} - \mathbf{L}^\star \right\|_{op} = \left\| \mathbf{L}^{(cross)} - \mathbb{E}\left[\mathbf{L}^{(cross)}\right] \right\|_{op} \leq C_{A.10}\left( \sqrt{nq \log (n/\delta)} + \log(n/\delta) \right).$$

Thus, using Weyl's inequality, for $n > N(\delta)$, we have

$$\boldsymbol{d}[v] - \lambda_2 \geq \boldsymbol{d}_{in}[v] - \lambda_2^\star - \left\| \mathbf{L} - \mathbf{L}^\star \right\|_{op}$$
$$\geq C_1 \frac{nq}{2} + C_1 \sqrt{n} - nq - C_{A.10}\left( \sqrt{nq \log (n/\delta)} + \log(n/\delta) \right) > 0.$$

Next, we verify that $\boldsymbol{d}_{in}[v] > \boldsymbol{d}_{out}[v]$ for all $v$. By Lemma A.3, with probability $\geq 1 - \delta$, for all $v \in V$, we have

$$\left| \boldsymbol{d}_{out}[v] - \frac{nq}{2} \right| \leq C_{A.3}\left( \sqrt{nq \log (n/\delta)} + \log (n/\delta) \right).$$

So for $n > N(\delta)$, we obtain

$$\boldsymbol{d}_{in}[v] - \boldsymbol{d}_{out}[v] \geq C_1 \frac{nq}{2} + C_1 \sqrt{n} - \frac{nq}{2} - C_{A.3}\left( \sqrt{nq \log (n/\delta)} + \log (n/\delta) \right) > 0.$$

Here, in the last inequality we used the fact that $\sqrt{nq \log (n/\delta)} \leq \max\{nq, \log(n/\delta)\}$.

Finally, we need to show that for all $v \in V$,

$$|\langle \boldsymbol{a}_v, \boldsymbol{u}_2^\star - \boldsymbol{u}_2 \rangle| \leq |\langle \boldsymbol{a}_v, \boldsymbol{u}_2^\star \rangle| = \frac{\boldsymbol{d}_{in}[v] - \boldsymbol{d}_{out}[v]}{\sqrt{n}}.$$

By Cauchy-Schwarz, we have

$$|\langle \boldsymbol{a}_v, \boldsymbol{u}_2^\star - \boldsymbol{u}_2 \rangle| \leq \|\boldsymbol{a}_v\|_2 \cdot \|\boldsymbol{u}_2^\star - \boldsymbol{u}_2\|_2 = \sqrt{\boldsymbol{d}_{in}[v] + \boldsymbol{d}_{out}[v]} \cdot \|\boldsymbol{u}_2^\star - \boldsymbol{u}_2\|_2.$$

Thus, it is enough to show that for all $v \in V$ we get

$$\sqrt{n} \|\boldsymbol{u}_2^\star - \boldsymbol{u}_2\|_2 \leq \frac{\boldsymbol{d}_{in}[v] - \boldsymbol{d}_{out}[v]}{\sqrt{\boldsymbol{d}_{in}[v] + \boldsymbol{d}_{out}[v]}}.$$

Observe that the RHS above is a decreasing function in $\boldsymbol{d}_{out}[v]$ and an increasing function in $\boldsymbol{d}_{in}[v]$.

Now, by Lemma A.16 and Lemma A.12, we have

$$\sqrt{n} \|\boldsymbol{u}_2^\star - \boldsymbol{u}_2\|_2 \leq \frac{\sqrt{n} \|\mathbf{E}\boldsymbol{u}_2^\star\|_2}{|\lambda_3 - \lambda_2^\star|} \leq \frac{6 C_{A.12} \sqrt{n} \left( \sqrt{nq} + (nq \log (n/\delta))^{1/4} + \sqrt{\log (n/\delta)} \right)}{|\lambda_3 - \lambda_2^\star|}. \tag{9}$$

We now do casework on the value of $q$.

**Case 1:** $q \leq \log\left(n/\delta\right)/n$.  Carrying on from (9) and applying Lemma A.10 (we can set $p_{ij}$ for the deterministic internal edges to 0 as they do not affect $\mathbf{L} - \mathbb{E}\left[\mathbf{L}\right]$) along with Weyl's inequality, for all $n \geq N(\delta)$ we have

$$\sqrt{n}\left\|\boldsymbol{u}_2^\star - \boldsymbol{u}_2\right\|_2 \leq \frac{18C_{A.12}\sqrt{n\log\left(n/\delta\right)}}{\left|\lambda_3 - \lambda_2^\star\right|} \leq \frac{18C_{A.12}\sqrt{n\log\left(n/\delta\right)}}{\sqrt{n} - 3C_{A.10}\log\left(n/\delta\right)}$$

$$\leq C\sqrt{\log\left(n/\delta\right)} \ll \frac{\boldsymbol{d}_{\mathsf{in}}[v] - \boldsymbol{d}_{\mathsf{out}}[v]}{\sqrt{\boldsymbol{d}_{\mathsf{in}}[v] + \boldsymbol{d}_{\mathsf{out}}[v]}},$$

as required.  Here the last inequality follows using the fact that $\boldsymbol{d}_{\mathsf{in}}[v] \geq C_1\left(\frac{nq}{2} + \sqrt{n}\right)$ and $\boldsymbol{d}_{\mathsf{out}}[v] \leq \frac{nq}{2} + 2C_{A.3}\log(n/\delta)$.

**Case 2:** $\log\left(n/\delta\right)/n \leq q$.  Similar to the previous case, we get

$$\sqrt{n}\left\|\boldsymbol{u}_2^\star - \boldsymbol{u}_2\right\|_2 \leq \frac{18C_{A.12}\sqrt{n}\cdot\sqrt{nq}}{\left|\lambda_3 - \lambda_2^\star\right|} \leq \frac{18C_{A.12}\sqrt{n}\cdot\sqrt{nq}}{\sqrt{n} + (C_2 - 2C_{A.10})nq} \tag{10}$$

$$\leq 18C_{A.12}\cdot\max\left\{\sqrt{nq}, \frac{1}{(C_2 - 2C_{A.10})\sqrt{q}}\right\}. \tag{11}$$

Additionally, we can use the conclusion of Lemma A.3 to write with probability $\geq 1 - \delta$ for all $v \in V$ and $n \geq N(\delta)$ that

$$\frac{\boldsymbol{d}_{\mathsf{in}}[v] - \boldsymbol{d}_{\mathsf{out}}[v]}{\sqrt{\boldsymbol{d}_{\mathsf{in}}[v] + \boldsymbol{d}_{\mathsf{out}}[v]}} \geq \frac{(C_1/2 - 2C_{A.3} - 1/2)nq + C_1\sqrt{n}}{\sqrt{(C_1/2 + 2C_{A.3} + 1/2)nq}} \tag{12}$$

$$\geq \frac{C_1/2 - 2C_{A.3} - 1/2}{\sqrt{C_1/2 + 2C_{A.3} + 1/2}}\max\left\{\sqrt{nq}, \sqrt{\frac{1}{q}}\right\}. \tag{13}$$

From this, it is clear that one can choose constants $C_1$ and $C_2$ such that (11) is at most (13). Taking a union bound over all our (constantly many) probabilistic statements, setting $\delta = \Theta(1/n)$, and rescaling completes the proof of Theorem 2.  $\square$

### A.8  Inconsistency of normalized spectral bisection

In this section, we design a family of problem instances on which unnormalized spectral bisection is strongly consistent whereas normalized spectral bisection is inconsistent. Specifically, our goal is to prove Theorem 3.

**Theorem 3.** *For all $n$ sufficiently large, there exists a nonhomogeneous stochastic block model such that unnormalized spectral bisection is strongly consistent whereas normalized spectral bisection (both symmetric and random-walk) incurs a misclassification rate of at least $24\%$ with probability $1 - 1/n$.*

#### A.8.1  The nested block example

We first state the family of instances on which we will prove our inconsistency results. Let $n$ be a multiple of 4. Let $L_1$ consist of indices $1, \ldots, n/4$, $L_2$ consist of indices $n/4 + 1, \ldots, n/2$, and $R$ consist of indices $n/2 + 1, \ldots, n$.

As mentioned in Section 3, consider the following block structure determined by the $\mathbf{A}^\star$ written below, where $q < p$ and $K \geq 3p/q$.

|       | $L_1$                           | $L_2$                           | $R$                           |
|-------|---------------------------------|---------------------------------|-------------------------------|
| $L_1$ | $Kp\cdot\mathbb{1}_{n/4\times n/4}$ | $p\cdot\mathbb{1}_{n/4\times n/4}$ | $q\cdot\mathbb{1}_{n/2\times n/2}$ |
| $L_2$ | $p\cdot\mathbb{1}_{n/4\times n/4}$ | $Kp\cdot\mathbb{1}_{n/4\times n/4}$ |                               |
| $R$   | $q\cdot\mathbb{1}_{n/2\times n/2}$ |                                 | $p\cdot\mathbb{1}_{n/2\times n/2}$ |

Table 2: $\mathbf{A}^\star$ is defined to have the above block structure.

We will draw our instances from the nonhomogeneous stochastic block model according to the probabilities prescribed above. Note that within the two clusters $L \coloneqq L_1 \cup L_2$ and $R$, each edge

appears with probability at least $p$. Moreover, each edge in $L \times R$ appears with probability exactly $q$. However, there are also two subcommunities $L_1$ and $L_2$ that appear within $L$. Furthermore, observe that unnormalized spectral bisection is consistent on this family of examples with probability $\geq 1 - 1/n$ by Theorem 1.

### A.8.2 Technical lemmas

We next show some technical statements that we will need later in the proof of Theorem 3.

**Lemma A.24.** *Let* $\mathbf{M} \in \mathbb{R}^{k \times k}$. *Then,*

$$\|\mathbf{M}\|_{\mathrm{op}} \leq \max_{i \leq k} |\mathbf{M}[i][i]| + k \max_{i \neq j} |\mathbf{M}[i][j]| .$$

*Proof of Lemma A.24.* For a matrix $\mathbf{N} \in \mathbb{R}^{k \times k}$, it is easy to check that

$$\|\mathbf{N}\|_{\mathrm{op}} \leq \|\mathbf{N}\|_F \leq k \max_{i,j \leq k} |\mathbf{N}[i][j]| .$$

Next, let $\mathrm{diag}\,(\mathbf{M})$ denote the matrix that agrees with $\mathbf{M}$ on the diagonal and is $0$ elsewhere. Notice that

$$\|\mathbf{M}\|_{\mathrm{op}} \leq \|\mathrm{diag}\,(\mathbf{M})\|_{\mathrm{op}} + \|\mathbf{M} - \mathrm{diag}\,(\mathbf{M})\|_{\mathrm{op}} \leq \max_{i \leq k} |\mathbf{M}[i][i]| + k \max_{i \neq j} |\mathbf{M}[i][j]| ,$$

completing the proof of Lemma A.24. $\qquad\square$

**Lemma A.25.** *Let* $\varepsilon_x$ *be a constant where* $0 \leq \varepsilon_x < x$. *Let* $\varepsilon_y$ *be defined similarly. The function* $f(x,y)$ *defined as*

$$f(x,y) := \frac{1}{\sqrt{x - \varepsilon_x}\sqrt{y - \varepsilon_y}} - \frac{1}{\sqrt{x}\sqrt{y}}$$

*is decreasing in* $x$ *and* $y$.

*Proof of Lemma A.25.* It is enough to just check the inequality for $x$. We take the derivative of $f(x,y)$ with respect to $x$ and get

$$\frac{1}{2}\left( -\frac{1}{(x - \varepsilon_x)^{3/2}(y - \varepsilon_y)^{1/2}} + \frac{1}{x^{3/2}y^{1/2}} \right) < 0,$$

where the inequality follows from observing $0 < x - \varepsilon_x \leq x$ and similarly for $y$. This completes the proof of Lemma A.25. $\qquad\square$

### A.8.3 Proof of Theorem 3

First, we construct $\mathcal{L}^\star$.

**Lemma A.26.** *Let* $\mathcal{L}^\star := \mathbf{I} - (\mathbf{D}^\star)^{-1/2} \mathbf{A}^\star (\mathbf{D}^\star)^{-1/2}$. *Then,* $\mathbf{I} - \mathcal{L}^\star$ *has the following block structure.*

|       | $L_1$ | $L_2$ | $R$ |
|-------|-------|-------|-----|
| $L_1$ | $\frac{Kp}{\frac{n}{2}\cdot\left(p\cdot\frac{K+1}{2}+q\right)} \cdot \mathbb{1}_{n/4 \times n/4}$ | $\frac{p}{\frac{n}{2}\cdot\left(p\cdot\frac{K+1}{2}+q\right)} \cdot \mathbb{1}_{n/4 \times n/4}$ | $\frac{q}{\sqrt{\frac{n}{2}\cdot\left(p\cdot\frac{K+1}{2}+q\right)\cdot\frac{n}{2}\cdot(p+q)}} \cdot \mathbb{1}_{n/2 \times n/2}$ |
| $L_2$ | $\frac{p}{\frac{n}{2}\cdot\left(p\cdot\frac{K+1}{2}+q\right)} \cdot \mathbb{1}_{n/4 \times n/4}$ | $\frac{Kp}{\frac{n}{2}\cdot\left(p\cdot\frac{K+1}{2}+q\right)} \cdot \mathbb{1}_{n/4 \times n/4}$ | $\frac{q}{\sqrt{\frac{n}{2}\cdot\left(p\cdot\frac{K+1}{2}+q\right)\cdot\frac{n}{2}\cdot(p+q)}} \cdot \mathbb{1}_{n/2 \times n/2}$ |
| $R$   | $\frac{q}{\sqrt{\frac{n}{2}\cdot\left(p\cdot\frac{K+1}{2}+q\right)\cdot\frac{n}{2}\cdot(p+q)}} \cdot \mathbb{1}_{n/2 \times n/2}$ | | $\frac{p}{\frac{n}{2}\cdot(p+q)} \cdot \mathbb{1}_{n/2 \times n/2}$ |

*Proof of Lemma A.26.* It is easy to see that for any $v \in L$, we have $\boldsymbol{d}^\star[v] = \frac{n}{2} \cdot \left(p \cdot \frac{K+1}{2} + q\right)$ and for any $v \in R$, we have $\boldsymbol{d}^\star[v] = \frac{n}{2} \cdot (p + q)$. Lemma A.26 follows by noting that every element of $\mathbf{I} - \mathcal{L}^\star$ is of the form $\boldsymbol{a}_i^\star[j]/\sqrt{\boldsymbol{d}^\star[i]\boldsymbol{d}^\star[j]}$. $\qquad\square$

Next, we analyze the eigenvalues and eigenvectors of $\mathcal{L}^\star$.

**Lemma A.27.** *Up to normalization and sign, the eigenvector-eigenvalue pairs of* $\mathbf{I} - \mathcal{L}^\star$ *corresponding to the nonzero eigenvalues of* $\mathbf{I} - \mathcal{L}^\star$ *are*

$$(\lambda_1^\star, \boldsymbol{u}_1^\star) = \left(1, \left[\mathbb{1}_{n/4} \oplus \mathbb{1}_{n/4} \oplus y_+ \cdot \mathbb{1}_{n/4} \oplus y_+ \cdot \mathbb{1}_{n/4}\right]\right)$$

$$(\lambda_2^\star, \boldsymbol{u}_2^\star) = \left(\frac{(K-1)p}{2\left(p \cdot \frac{K+1}{2} + q\right)}, \left[\mathbb{1}_{n/4} \oplus -\mathbb{1}_{n/4} \oplus 0_{n/4} \oplus 0_{n/4}\right]\right)$$

$$(\lambda_3^\star, \boldsymbol{u}_3^\star) = \left(-1 + p\left(\frac{1}{p+q} + \frac{K+1}{p(K+1)+2q}\right), \left[\mathbb{1}_{n/4} \oplus \mathbb{1}_{n/4} \oplus y_- \cdot \mathbb{1}_{n/4} \oplus y_- \cdot \mathbb{1}_{n/4}\right]\right)$$

*where* $y_+$ *and* $y_-$ *are chosen according to the formulas*

$$y_+ = \sqrt{\frac{2(p+q)}{p(K+1)+2q}} \qquad\qquad y_- = -\sqrt{\frac{p(K+1)+2q}{2(p+q)}}.$$

*Moreover, we have* $\lambda_1^\star > \lambda_2^\star > \lambda_3^\star > 0$ *and*

$$\lambda_2^\star - \lambda_3^\star \geq 1 - \frac{p^2(K+3) + 4pq}{p^2(K+3) + 4pq + 2q^2}.$$

*Proof of Lemma A.27.* As we can see from Lemma A.26, $\mathbf{I} - \mathcal{L}^\star$ is a matrix whose rank is at most 3, since it can be constructed by carefully repeating 3 distinct column vectors. Thus, it can have at most 3 nonzero eigenvalues. In what follows, we consider the case where $K > 1$ so that there are exactly 3 nonzero eigenvalues.

The next step is to confirm that the stated eigenvalue-eigenvector pairs are in fact valid. We begin with $\boldsymbol{u}_1^\star$. Every entry in the first $n/2$ entries of $(\mathbf{I} - \mathcal{L}^\star)\boldsymbol{u}_1^\star$ can be expressed as

$$\frac{n}{4} \cdot \frac{Kp}{\frac{n}{2} \cdot \left(p \cdot \frac{K+1}{2} + q\right)} + \frac{n}{4} \cdot \frac{p}{\frac{n}{2} \cdot \left(p \cdot \frac{K+1}{2} + q\right)} + \frac{n}{2} \cdot \left(\frac{q \cdot \sqrt{\frac{2(p+q)}{p(K+1)+2q}}}{\sqrt{\frac{n}{2} \cdot \left(p \cdot \frac{K+1}{2} + q\right) \cdot \frac{n}{2} \cdot (p+q)}}\right)$$

$$= \frac{(K+1)p}{(K+1)p + 2q} + \frac{q \cdot \sqrt{\frac{2(p+q)}{p(K+1)+2q}}}{\sqrt{\left(p \cdot \frac{K+1}{2} + q\right)(p+q)}} = \frac{(K+1)p}{(K+1)p + 2q} + \frac{q \cdot \sqrt{\frac{2}{p(K+1)+2q}}}{\sqrt{\left(p \cdot \frac{K+1}{2} + q\right)}}$$

$$= \frac{(K+1)p}{(K+1)p + 2q} + \frac{2q}{(K+1)p + 2q} = 1,$$

and every entry in the second $n/2$ entries of $(\mathbf{I} - \mathcal{L}^\star)\boldsymbol{u}_1^\star$ can be expressed as

$$\frac{n}{2} \cdot \frac{q}{\sqrt{\frac{n}{2} \cdot \left(p \cdot \frac{K+1}{2} + q\right) \cdot \frac{n}{2} \cdot (p+q)}} + \frac{n}{2} \cdot \frac{p}{\frac{n}{2} \cdot (p+q)} \cdot \sqrt{\frac{2(p+q)}{p(K+1)+2q}}$$

$$= \frac{q}{\sqrt{\left(p \cdot \frac{K+1}{2} + q\right)(p+q)}} + \frac{p}{(p+q)} \cdot \sqrt{\frac{2(p+q)}{p(K+1)+2q}}$$

$$= \frac{q}{\sqrt{\left(p \cdot \frac{K+1}{2} + q\right)(p+q)}} + p \cdot \sqrt{\frac{1}{\left(p \cdot \frac{K+1}{2} + q\right)(p+q)}}$$

$$= \frac{\sqrt{p+q}}{\sqrt{p \cdot \frac{K+1}{2} + q}} = \sqrt{\frac{2(p+q)}{p(K+1)+2q}} = y_+.$$

For $\boldsymbol{u}_2^\star$, we can use the block structure and easily verify

$$(\mathbf{I} - \mathcal{L}^\star)\boldsymbol{u}_2^\star = \frac{n}{4} \cdot \frac{(K-1)p}{\frac{n}{2} \cdot \left(p \cdot \frac{K+1}{2} + q\right)} \left[\mathbb{1}_{n/4} \oplus -\mathbb{1}_{n/4} \oplus 0_{n/4} \oplus 0_{n/4}\right] = \lambda_2^\star \boldsymbol{u}_2^\star.$$

We now address $\boldsymbol{u}_3^\star$. The first $n/2$ entries of $(\mathbf{I} - \mathcal{L}^\star)\boldsymbol{u}_3^\star$ are

$$\frac{n}{4} \cdot \frac{Kp}{\frac{n}{2} \cdot \left(p \cdot \frac{K+1}{2} + q\right)} + \frac{n}{4} \cdot \frac{p}{\frac{n}{2} \cdot \left(p \cdot \frac{K+1}{2} + q\right)} + \frac{n}{2} \cdot \left(\frac{q \cdot -\sqrt{\frac{p(K+1)+2q}{2(p+q)}}}{\sqrt{\frac{n}{2} \cdot \left(p \cdot \frac{K+1}{2} + q\right) \cdot \frac{n}{2} \cdot (p+q)}}\right)$$

$$= \frac{(K+1)p}{(K+1)p + 2q} + \left(\frac{q \cdot -\sqrt{\frac{1}{p+q}}}{\sqrt{p+q}}\right) = \frac{(K+1)p}{(K+1)p + 2q} - \frac{q}{p+q} = \lambda_3^\star,$$

and the second $n/2$ entries of $(\mathbf{I} - \mathcal{L}^\star)\boldsymbol{u}_3^\star$ are

$$\frac{n}{2} \cdot \frac{q}{\sqrt{\frac{n}{2} \cdot \left(p \cdot \frac{K+1}{2} + q\right) \cdot \frac{n}{2} \cdot (p+q)}} + \frac{n}{2} \cdot \frac{p}{\frac{n}{2} \cdot (p+q)} \cdot -\sqrt{\frac{p(K+1)+2q}{2(p+q)}}$$

$$= \frac{q}{\sqrt{\left(p \cdot \frac{K+1}{2} + q\right)(p+q)}} - \frac{p}{(p+q)} \cdot \sqrt{\frac{p(K+1)+2q}{2(p+q)}}$$

$$= -\sqrt{\frac{p(K+1)+2q}{2(p+q)}} \left(\frac{-2q}{p(K+1)+2q} + \frac{p}{p+q}\right) = y_- \cdot \lambda_3^\star.$$

Finally, it remains to check that $1 > \lambda_2^\star > \lambda_3^\star > 0$. The fact that $\lambda_2^\star < 1$ easily follows from using $p + q > 0$. To prepare to bound $\lambda_2^\star - \lambda_3^\star$, we first use $p \geq q$ to establish

$$p^2 - pq + 2q^2 = p(p - q) + 2q^2 \geq 2q^2.$$

This implies

$$pq(K - 1) + 2q^2 \geq 3p^2 - pq + 2q^2 = 2p^2 + (p^2 - pq + 2q^2) \geq 2p^2 + 2q^2,$$

which rearranges to

$$p^2(K + 1) + pq(K + 3) + 2q^2 \geq p^2(K + 3) + 4pq + 2q^2.$$

Next, we write

$$\lambda_2^\star - \lambda_3^\star = \left(\frac{(K - 1)p}{2\left(p \cdot \frac{K+1}{2} + q\right)}\right) - \left(-1 + p\left(\frac{1}{p+q} + \frac{K+1}{p(K+1)+2q}\right)\right)$$

$$= 1 - \frac{p}{p+q} - \frac{2p}{p(K+1)+2q} = 1 - \left(\frac{p^2(K+1) + 2pq + 2p^2 + 2pq}{(p+q)(p(K+1)+2q)}\right)$$

$$= 1 - \frac{p^2(K+3) + 4pq}{p^2(K+1) + pq(K+3) + 2q^2} \geq 1 - \frac{p^2(K+3) + 4pq}{p^2(K+3) + 4pq + 2q^2} > 0.$$

Finally, to show $\lambda_3^\star > 0$, we write

$$\lambda_3^\star + 1 = \frac{p}{p+q} + \frac{p(K+1)}{p(K+1)+2q} > \frac{2p}{p+q} > 1,$$

which allows us to complete the proof of Lemma A.27. $\qquad\square$

Next, we argue that studying $\mathcal{L}^\star$, which is formed by taking into account the weighted self-loops, gives us an understanding that is not too far from that of $\mathcal{L}_{\mathsf{nl}}^\star$, which is formed by setting $p_{vv} = 0$ for all $v \in V$.

**Lemma A.28.** *Let $\mathbf{P}$ be the diagonal matrix where $\mathbf{P}[v, v] = p_{vv}$. Let $\mathcal{L}_{\mathsf{nl}}^\star$ be the normalized Laplacian of the graph formed by $\mathbf{A}^\star - \mathbf{P}$. Then, we have*

$$\|\mathcal{L}^\star - \mathcal{L}_{\mathsf{nl}}^\star\|_{\mathsf{op}} \leq \frac{6K}{n - 2}.$$

*Proof of Lemma A.28.* Recall $\mathbf{L}^\star := \mathbf{D}^\star - \mathbf{A}^\star$. Let $\mathbf{D}^\star_{\mathsf{nl}}$ be defined analogously to $\mathcal{L}^\star_{\mathsf{nl}}$. Observe that we have

$$\mathcal{L}^\star = (\mathbf{D}^\star)^{-1/2}\,\mathbf{L}^\star\,(\mathbf{D}^\star)^{-1/2}$$

$$\mathcal{L}^\star_{\mathsf{nl}} = (\mathbf{D}^\star_{\mathsf{nl}})^{-1/2}\,\mathbf{L}^\star\,(\mathbf{D}^\star_{\mathsf{nl}})^{-1/2}\,.$$

From this, we see that writing down the $v, w$th entry of the difference gives

$$(\mathcal{L}^\star_{\mathsf{nl}} - \mathcal{L}^\star)\,[v, w] = \mathbf{L}^\star[v,w]\left(\frac{1}{\sqrt{(\boldsymbol{d}^\star[v] - p_{vv})(\boldsymbol{d}^\star[w] - p_{ww})}} - \frac{1}{\sqrt{\boldsymbol{d}^\star[v]\boldsymbol{d}^\star[w]}}\right).$$

This resolves to different forms based on whether $v = w$. When $v = w$, evaluating the formula gives

$$(\mathcal{L}^\star_{\mathsf{nl}} - \mathcal{L}^\star)\,[v, v] = \frac{\boldsymbol{d}^\star[v] - p_{vv}}{\boldsymbol{d}^\star[v] - p_{vv}} - \frac{\boldsymbol{d}^\star[v] - p_{vv}}{\boldsymbol{d}^\star[v]} = \frac{p_{vv}}{\boldsymbol{d}^\star[v]}.$$

When $v \neq w$, we apply Lemma A.25 and get

$$|(\mathcal{L}^\star_{\mathsf{nl}} - \mathcal{L}^\star)\,[v, w]| = p_{vw}\left(\frac{1}{\sqrt{(\boldsymbol{d}^\star[v] - p_{vv})(\boldsymbol{d}^\star[w] - p_{ww})}} - \frac{1}{\sqrt{\boldsymbol{d}^\star[v]\boldsymbol{d}^\star[w]}}\right)$$

$$\leq Kp\left(\frac{1}{np/2 - p} - \frac{1}{np/2}\right) = \frac{4K}{n^2 - 2n}.$$

Using this analysis and applying Lemma A.24 gives

$$\|\mathcal{L}^\star_{\mathsf{nl}} - \mathcal{L}^\star\|_{\mathrm{op}} \leq \max_{v \in V}\frac{p_{vv}}{\boldsymbol{d}^\star[v]} + n\max_{v \neq w}p_{vw}\left(\frac{1}{\sqrt{(\boldsymbol{d}^\star[v] - p_{vv})(\boldsymbol{d}^\star[w] - p_{ww})}} - \frac{1}{\sqrt{\boldsymbol{d}^\star[v]\boldsymbol{d}^\star[w]}}\right)$$

$$\leq \frac{2K}{n} + n\max_{v \neq w}p_{vw}\left(\frac{1}{\sqrt{(\boldsymbol{d}^\star[v] - p_{vv})(\boldsymbol{d}^\star[w] - p_{ww})}} - \frac{1}{\sqrt{\boldsymbol{d}^\star[v]\boldsymbol{d}^\star[w]}}\right)$$

$$\leq \frac{2K}{n} + \frac{4K}{n - 2} \leq \frac{6K}{n - 2},$$

completing the proof of Lemma A.28. $\qquad\square$

This gives Lemma A.29, which means we can use $\boldsymbol{u}^\star_2$ as a suitable proxy for $\mathsf{sign}\,(\boldsymbol{u}_2(\mathcal{L}^\star_{\mathsf{nl}}))$.

**Lemma A.29.** *There exists a constant $C(\alpha, K)$ depending on $\alpha$ and $K$ such that we have*

$$\|\boldsymbol{u}_2(\mathcal{L}^\star_{\mathsf{nl}}) - \boldsymbol{u}^\star_2\|_\infty \leq \frac{C(\alpha, K)}{n}.$$

*This implies that for all $n$ sufficiently large, we have $\mathsf{sign}\,(\boldsymbol{u}_2(\mathcal{L}^\star_{\mathsf{nl}})) = \mathsf{sign}\,(\boldsymbol{u}^\star_2)$.*

*Proof of Lemma A.29.* By Lemma A.27, Weyl's inequality, and Lemma A.28, we know that for all $n$ sufficiently large,

$$\lambda^\star_2 - \lambda_3(\mathcal{L}^\star_{\mathsf{nl}}) = (\lambda^\star_2 - \lambda^\star_3) + (\lambda^\star_3 - \lambda_3(\mathcal{L}^\star_{\mathsf{nl}}))$$

$$\geq \left(1 - \frac{p^2(K + 3) + 4pq}{p^2(K + 3) + 4pq + 2q^2}\right) - \frac{C_{A.28}}{n} \geq C_1(\alpha, K).$$

Combining this with Lemma A.28 again, the Davis-Kahan inequality tells us that

$$\|\boldsymbol{u}_2(\mathcal{L}^\star_{\mathsf{nl}}) - \boldsymbol{u}^\star_2\|_\infty \leq \|\boldsymbol{u}_2(\mathcal{L}^\star_{\mathsf{nl}}) - \boldsymbol{u}^\star_2\|_2 \leq \frac{C_2(\alpha, K)}{n},$$

and then using the fact that $\|\boldsymbol{u}^\star_2\|_\infty = 1/\sqrt{n}$ (arising from Lemma A.27) completes the proof of Lemma A.29. $\qquad\square$

We are now ready to prove the inconsistency of normalized spectral bisection on the nested block examples.

*Proof of Theorem 3.* Let $G$ be a graph drawn from the nested block example. We choose $p$ and $q$ such that $p \gtrsim \log n/n$ and $p/q = \alpha \geq 2$ where $\alpha$ is some constant and such that $p$ and $q$ both satisfy the conditions of Theorem 1. Let $K \geq 3\alpha$. Observe that the true communities are $L$ and $R$. We will show that bisection based on $\boldsymbol{u}_2$ of $\mathbf{I} - \mathcal{L}$ (corresponding to the eigenvector associated with the second smallest eigenvalue of $\mathcal{L}$) will attain a large misclassification rate. In particular, based on our calculation in Lemma A.27, we expect that $\boldsymbol{u}_2$ will output a bisection that places $L_1$ and $L_2$ into separate clusters. On the other hand, by Theorem 1, for all $n$ large enough, the unnormalized spectral bisection algorithm will be strongly consistent.

First, observe that it is enough to prove the inconsistency result just for the symmetric normalized Laplacian. Indeed, observe that if $\boldsymbol{u}_2$ is an eigenvector of $\mathbf{I} - \mathcal{L} = \mathbf{D}^{-1/2}\mathbf{A}\mathbf{D}^{-1/2}$, then we have
$$\lambda_2 \mathbf{D}^{-1/2}\boldsymbol{u}_2 = \mathbf{D}^{-1}\mathbf{A}\mathbf{D}^{-1/2}\boldsymbol{u}_2 = \mathbf{D}^{-1}\mathbf{A}(\mathbf{D}^{-1/2}\boldsymbol{u}_2),$$
which shows that $\mathbf{D}^{-1/2}\boldsymbol{u}_2$ must be the eigenvector of the random-walk normalized Laplacian $\mathbf{I} - \mathbf{D}^{-1}\mathbf{A}$ corresponding to eigenvalue $\lambda_2$. Since $\mathbf{D}$ is a positive diagonal matrix, it does not change the signs of $\boldsymbol{u}_2$ and therefore the output of the normalized spectral bisection algorithm is the same.

Our general approach to prove the inconsistency is to use the Davis-Kahan Theorem, a bound on $\|\mathcal{L} - \mathcal{L}_{\mathsf{nl}}^\star\|_{\mathrm{op}}$, and a bound on the gap $\lambda_2^\star - \lambda_3$. Let $\boldsymbol{d}_{\min}$ be the minimum degree of the graph given by adjacency matrix $\mathbf{A}$ and let $\boldsymbol{d}_{\min}^\star$ be the minimum weighted degree of the graph given by the adjacency matrix $\mathbf{A}^\star$. First, using [DLS21, Theorem 3.1], we have with probability $1 - n^{-r}$ for some constant $r \geq 1$ and constants $C(r)$ and $C$ (the latter of which does not depend on $r$), for all $n$ sufficiently large,

$$\|\mathcal{L} - \mathcal{L}_{\mathsf{nl}}^\star\|_{\mathrm{op}} \leq \frac{C(r)\left(n \max_{(i,j)} p_{ij}\right)^{5/2}}{\min\left\{\boldsymbol{d}_{\min}, \boldsymbol{d}_{\min}^\star\right\}^3}$$

$$\leq \frac{C(r)\left(n \cdot Kp\right)^{5/2}}{\min\left\{n(p+q)/3, n(p+q)/3 - C\sqrt{n(p+q)\log n}\right\}^3}$$

$$\leq \frac{C_1(r,\alpha)K^{5/2}(np)^{5/2}}{(np)^3} = \frac{C_1(r,\alpha)K^{5/2}}{\sqrt{np}}.$$

Next, we invoke Lemma A.28 to write
$$\lambda_2(\mathcal{L}_{\mathsf{nl}}^\star) - \lambda_3 = (\lambda_2^\star - \lambda_3^\star) + (\lambda_3^\star - \lambda_3) + (\lambda_2(\mathcal{L}_{\mathsf{nl}}^\star) - \lambda_2^\star)$$

$$\geq \left(1 - \frac{p^2(K+3) + 4pq}{p^2(K+3) + 4pq + 2q^2}\right) - \frac{C_2(r,\alpha)K^{5/2}}{\sqrt{np}} - \frac{C_{A.28}}{n} \geq C_g(\alpha, K),$$

where the last line denotes a positive constant depending on $q$ and $K$ (this constant will always be positive for sufficiently large $n$, as we showed that $\lambda_2^\star - \lambda_3^\star > 0$ in Lemma A.27).

Putting everything together, we get by the Davis-Kahan theorem that some signing of $\boldsymbol{u}_2$ satisfies
$$\|\boldsymbol{u}_2 - \boldsymbol{u}_2(\mathcal{L}_{\mathsf{nl}}^\star)\|_2 \leq \frac{\|\mathcal{L} - \mathcal{L}_{\mathsf{nl}}^\star\|_{\mathrm{op}}}{\min\left\{|\lambda_2(\mathcal{L}_{\mathsf{nl}}^\star) - \lambda_3|, 1 - \lambda_2(\mathcal{L}_{\mathsf{nl}}^\star)\right\}} \leq \frac{C_3(r)K^{5/2}}{C_g'(\alpha, K)\sqrt{np}} \leq \frac{C_4(r,\alpha,K)}{\sqrt{np}}.$$
Now, consider the subset of coordinates of $\boldsymbol{u}_2$ belonging to $L_1$. Suppose $m$ of these coordinates do not agree in sign with $\boldsymbol{u}_2^\star$. To maximize $m$, each of these coordinates in $\boldsymbol{u}_2$ should be 0, so using this reasoning and applying Lemma A.29 means the total $\ell_2$ error can be bounded (using Lemma A.28) as

$$m\left(\frac{1}{\sqrt{n/2}} - \frac{C_{A.28}}{n}\right)^2 \leq \|\boldsymbol{u}_2 - \boldsymbol{u}_2(\mathcal{L}_{\mathsf{nl}}^\star)\|_2^2 \leq \frac{C_4(r,\alpha,K)^2}{np}.$$

This means the number of coordinates $m$ on which $\boldsymbol{u}_2$ and $\boldsymbol{u}_2^\star$ disagree on is at most
$$\frac{n \cdot C_5(r,\alpha,K)^2}{2np},$$
and therefore the misclassification rate of $\boldsymbol{u}_2$ with respect to the true labeling induced by $L$ and $R$ must be at least

$$\frac{\frac{n}{4} - \frac{n \cdot C_5(r,\alpha,K)^2}{2np}}{n} = \frac{1}{4} - \frac{C_5(r,\alpha,K)^2}{2np}.$$

Since $p \gtrsim \log n/n$, this completes the proof of Theorem 3. $\qquad\square$

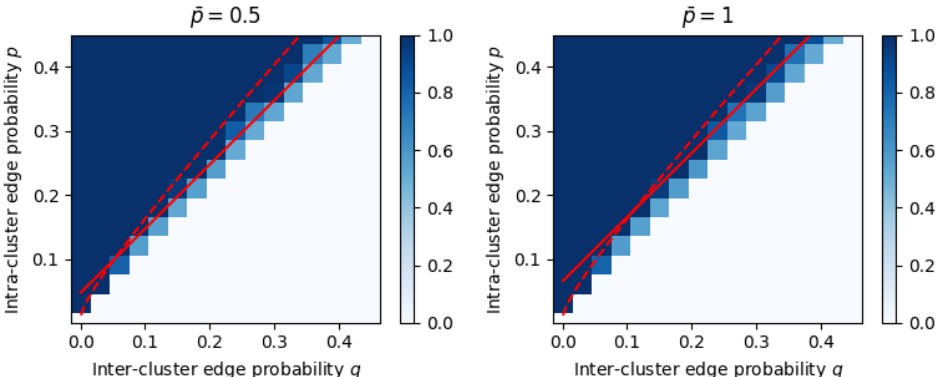

Figure 2: Agreement with the planted bisection of the bipartition obtained from unnormalized spectral bisection, for graphs generated from a distribution in $\mathrm{NSSBM}(n, p, \overline{p}, q)$ for fixed values of $n, \overline{p}$ and varying values of $p > q$. The left plot uses $\overline{p} = 1/2$, the right plot uses $\overline{p} = 1$. The solid red curves plot the function $p_{\mathsf{thr}}(q)$ (see (14)), and the dashed red curves plot the function $p_{\mathsf{info}}(q)$ (see (15)).

## B    Additional experiments

In this section, we show more numerical trials that complement those discussed in Section 4.

### B.1    Varying edge probabilities in an NSSBM

In Section 4, we investigated the behavior of an NSSBM model by fixing the values of $p, q$ and varying the largest edge probability $\overline{p}$. Here, we take an alternative approach, and instead fix $\overline{p}$ and vary the values of $p$ and $q$.

**Setup.**    Let us fix $n = 2000$, $\overline{p} \in \{1/2, 1\}$. For varying $p, q$ in the range $[1/n, 9/20]$ such that $p > q$, we sample $t = 3$ independent draws $G$ from the same benchmark distribution $\mathcal{D}_{p, \overline{p}, q}$ used in Section 4. For each of them, we compute the agreement of the bipartition obtained by unnormalized spectral bisection with respect to the planted bisection. For each $(p, q)$, we plot the average agreement across the $t$ independent draws. The results are shown in Fig. 2, where in the left and right plot we ran the experiments with $\overline{p} = 1/2$ and $\overline{p} = 1$ respectively. The lower diagonal of these plots, where $p \le q$, is artificially set to 0.

**Theoretical framing.**    According to Theorem 1, fixing the value of $\overline{p} \in \{1/2, 1\}$, we obtain that unnormalized spectral bisection achieves exact recovery provided that for $q \in [1/n, 9/20]$ one has $p \ge p_{\mathsf{thr}}(q)$ where

$$p_{\mathsf{thr}}(q) = \frac{\sqrt{\overline{p} \log n}}{\sqrt{n}} + q \tag{14}$$

is obtained by rearranging the precondition of Theorem 1, ignoring the constants, and disregarding the fact that $\alpha$ should be $O(1)$. The solid red curve in Fig. 2 plots $p_{\mathsf{thr}}(q)$ as a function of $q$. For comparison, the information-theoretic threshold for SSBM [ABH16] demands that $p \ge p_{\mathsf{info}}(q)$ where

$$p_{\mathsf{info}}(q) = \left( \sqrt{2} \sqrt{\frac{\log n}{n}} + \sqrt{q} \right)^2. \tag{15}$$

The dashed red curve in Fig. 2 plots $p_{\mathsf{info}}(q)$ as a function of $q$.

**Empirical evidence.**    From Fig. 2, one can see that our experiments reflect the behavior predicted by Theorem 1 quite closely, although empirically we achieve 100% agreement slightly above $p_{\mathsf{thr}}(q)$ (i.e. the solid red curve). However, this is likely due to the constant factors from Theorem 1 that we ignored, and also $n = 2000$ is plausibly too small to show asymptotic behaviors. Nevertheless, we do achieve 100% agreement consistently as soon as we surpass the information-theoretic threshold $p_{\mathsf{info}}(q)$: in the regime of our experiment, it appears that the unnormalized Laplacian is robust all the way to the optimal threshold for exact recovery in the SSBM.

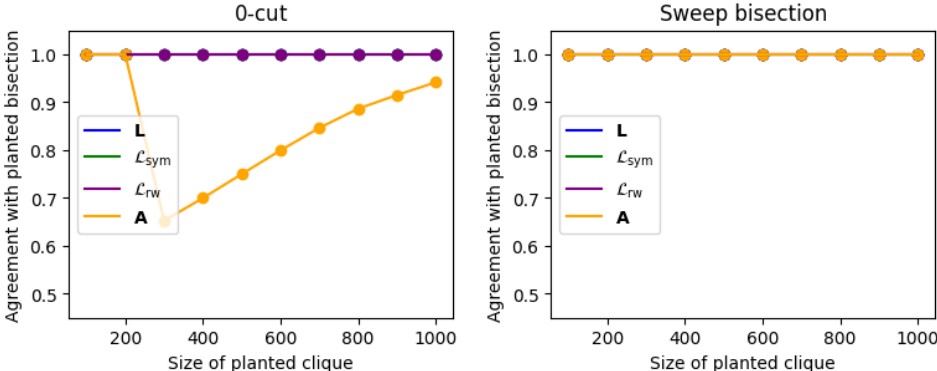

Figure 3: Agreement with the planted bisection of the bipartition obtained from several matrices associated with an input graph generated from a distribution $\mathcal{D}_q^{G_1,G_2} \in \mathsf{DCM}(n, d_{\mathsf{in}}, q)$ for fixed values of $n, q$ and varying the size of the planted clique $S$. In the left plot, the bipartition is the $0$-cut of the second eigenvector, as in Algorithm 1. In the right plot, the bipartition is the sweep cut of the first $n/2$ vertices in the second eigenvector.

## B.2    Varying the size of a planted clique in a DCM

In some sense, the experiments from Section 4 and Appendix B.1 can be thought of as experiments for the deterministic clusters model too. This is because each realization of the internal edges gives rise to a different DCM distribution (see Section 2). We complement our previous discussion by illustrating the behavior of certain families of DCM distributions that are conceptually different than those considered in Section 4.

**Benchmark distribution.**    Let $n$ be divisible by 4 and let $\{P_1, P - 2\}$ be a partitioning of $V = [n]$ into two equally-sized subsets. Fix $p \in [0, 1]$. For some set $S \subseteq P_1$ such that $S = \{1, \ldots, |S|\}$ (for simplicity), let $G_2 = (P_2, E_2) \sim \mathsf{ER}(n/2, p)$ be a graph drawn from the Erdős-Rényi distribution with sampling rate $p$, and let $G_1 = (P_1, E_1) \sim \mathsf{ERPC}(n/2, p, S)$ be also a graph drawn from the Erdős-Rényi distribution with sampling rate $p$ where we additionally plant a clique on the vertices $S$. Fixing $G_1, G_2$, for $q \in [0, 1]$ we consider the distribution $\mathcal{D}_q^{G_1,G_2}$ over graphs $G = (V, E)$ where $G[P_1] = G_1$, $G[P_2] = G_2$, and every edge $(u, v) \in P_1 \times P_2$ is sampled independently with probability $q$. One can see that $\mathcal{D}_q^{G_1,G_2}$ is in fact in the set $\mathsf{DCM}(n, d_{\mathsf{in}}, q)$ for some $d_{\mathsf{in}}$.

**Setup.**    Let us fix $n = 2000$, $p = 9/\sqrt{n}$, $q = 1/\sqrt{n}$. For varying values of $|S|$ in the range $[|P_1|/10, |P_1|]$, we sample $G_1 = (P_1, E_1) \sim \mathsf{ERPC}(n/2, p, S)$ and $G_2 = (P_2, E_2) \sim \mathsf{ER}(n/2, p)$, and then draw $t = 10$ independent samples $G$ from $\mathcal{D}_q^{G_1,G_2}$. For each sample $G$, we run spectral bisection (i.e. Algorithm 1) with matrices $\mathbf{L}, \mathcal{L}_{\mathsf{sym}}, \mathcal{L}_{\mathsf{rw}}, \mathbf{A}$. Then, we compute the agreement of the bipartition hence obtained with respect to the planted bisection, and average it out across the $t$ independent draws. The results are shown in the left plot of Fig. 3. Again, another natural way to get a bipartition of $V$ from the eigenvector is a sweep cut, and the average agreements that this results in are shown in the right plot of Fig. 3.

**Theoretical framing.**    Ignoring the constants, Theorem 2 guarantees that exact recovery is achieved by unnormalized spectral bisection as long as $d_{\mathsf{in}} \geq nq + \sqrt{n}$ and $\lambda_3(\widehat{\mathbf{L}}) - \lambda_2(\widehat{\mathbf{L}}) \geq \sqrt{n} + nq + \sqrt{nq \log n} + \log n$, where $\widehat{\mathbf{L}}$ is the expected Laplacian of $\mathcal{D}_q^{G_1,G_2}$. For each clique size that we consider, Fig. 4 shows the minimum in-cluster degree of the graphs $G_1, G_2$ that we draw (in the left plot), and the spectral gap $\lambda_3(\widehat{\mathbf{L}}) - \lambda_2(\widehat{\mathbf{L}})$. The red horizontal lines in the left and right plot respectively correspond to the value of $nq + \sqrt{n}$ and $\sqrt{n} + nq + \sqrt{nq \log n} + \log n$ on the $y$-axis, indicating the lower bound on $d_{\mathsf{in}}$ and $\lambda_3(\widehat{\mathbf{L}}) - \lambda_2(\widehat{\mathbf{L}})$ demanded by Theorem 2.

**Empirical evidence: consistency.**    From Fig. 4, one can see that all the distributions $\mathcal{D}_q^{G_1,G_2}$ that we use roughly meet the requirement of Theorem 2. Indeed, in the left plot of Fig. 3 one sees that unnormalized spectral bisection consistently achieves exact recovery for all clique sizes. On the contrary, the bipartition obtained by running spectral bisection with the adjacency matrix $\mathbf{A}$

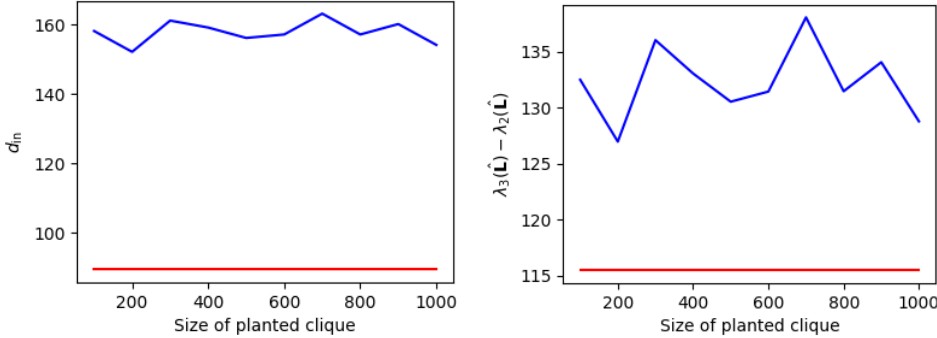

Figure 4: The minimum in-cluster degree $d_{\text{in}}$ and the spectral gap $\lambda_3(\widehat{\mathbf{L}}) - \lambda_2(\widehat{\mathbf{L}})$ of distributions $\mathcal{D}_q^{G_1,G_2} \in \mathsf{DCM}(n, d_{\text{in}}, q)$ with fixed values of $n, q$ and varying the size of the planted clique $S$. The red horizontal line on the left corresponds to the value $nq + \sqrt{n}$, the red horizontal line on the right corresponds to the value $\sqrt{n} + nq + \sqrt{nq \log n} + \log n$.

misclassifies a fraction of the vertices for certain sizes of the planted clique. Nevertheless, the sweep cut obtained from all the matrices recovers the planted bisection exactly.

**Empirical evidence: example embedding.** Let us fix the value $|S| = 800$ for the size of the planted clique, for which we see in Fig. 3 that the adjacency matrix fails to recover the planted bisection. We generate a graph from a distribution $\mathcal{D}_q^{G_1,G_2}$ with clique size $|S| = 800$, and plot how the vertices are embedded in the real line by the second eigenvector of all the matrices we consider. The result is shown in Fig. 5, where the three horizontal dashed lines, from top to bottom, respectively correspond to the value of $1/\sqrt{n}, 0, -1/\sqrt{n}$ on the $y$-axis. Graphically, one can see that the embedding in the unnormalized Laplacian is indeed the one that moves the least away from the values $\pm 1/\sqrt{n}$, and in fact the vertices $\{1, \ldots, 800\} \subseteq P_1$ where we plant the clique concentrate even more around $1/\sqrt{n}$. This is a phenomenon related to the one illustrated by Fig. 1. Finally, one can see from the embedding that splitting vertices around $0$ does result in misclassifying a fraction of the vertices for the adjacency matrix. However, taking a sweep cut that splits the vertices into two equally sized parts recovers the planted bisection for all matrices. This reflects the results shown in Fig. 3.

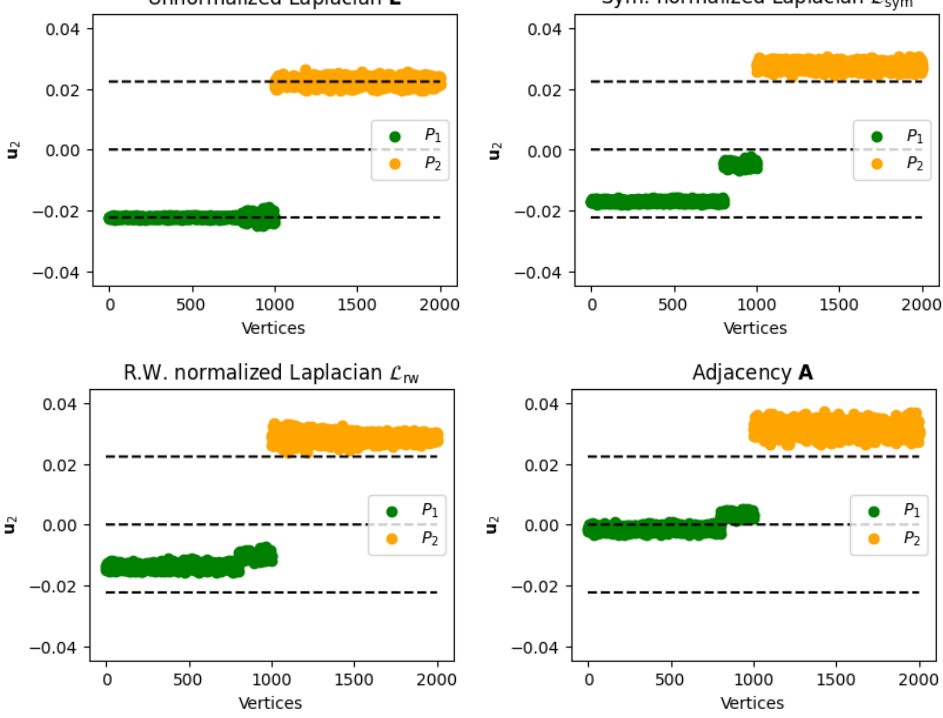

Figure 5: Embedding of the vertices given by the second eigenvector $\boldsymbol{u}_2$ of several matrices associated with a graph sampled from a distribution $\mathcal{D}_q^{G_1,G_2} \in \mathsf{DCM}(n, d_{\mathsf{in}}, q)$, with the size of the planted clique set to $|S| = 2/5 \cdot n$. Horizontal dashed lines, from top to bottom, correspond to $1/\sqrt{n}, 0, -1/\sqrt{n}$ respectively.

