# OpenReview forum: "On the Robustness of Spectral Algorithms for Semirandom Stochastic Block Models"
_NeurIPS.cc/2024/Conference — NeurIPS 2024 poster_

### Official Review · Reviewer_hmYB · 2024-07-10

**Soundness:** 3
**Presentation:** 3
**Contribution:** 3
**Rating:** 6
**Confidence:** 4

**Summary:**

The paper studies spectral algorithms for recovery in the stochastic block model (SBM).  Specifically, they consider a nonhomogeneous SBM where there are two communities $P_1, P_2$ and edges inside the communities occur (independently) with probability between $p$ and $\bar{p}$ while edges between the communities appear with probability $q$.  This is a generalization of the standard SBM where the edges inside communities occur with probability exactly $p$.  The goal is to take a graph generated from the nonhomogeneous SBM (but without the community labels) and recover the communities $P_1, P_2$.   The nonhomogeneous modification seems to only strengthen the "signal" about the communities but is actually a useful test case for the robustness of various recovery algorithms for SBM -- in fact many well-known algorithms for SBM may break in the nonhomogeneous case.  The nonhomogeneous model is a special case of the more general semi-random model where edges may be arbitrarily added within communities and edges connecting different communities may be arbitrarily removed.  The sharp characterization of when exact recovery of the communities is possible was shown in [ABH16].  Semidefinite programming algorithms achieve the sharp threshold and succeed in recovery even in the more general semi-random model [FK01].  The goal of this paper is understanding spectral algorithms for recovery, which may be more efficient than semidefinite programming algorithms.

The main results of the paper are that under a certain gap condition on $p,\bar{p},q$ that resembles the characterization in [ABH16] but is worse by some constant factor, spectral bisection based on the unnormalized Laplacian solves the recovery problem whereas using the spectrum of the normalized Laplacian provably fails.  The authors also prove a similar result in a modified setting called the deterministic cluster model.  The authors also run numerical simulations to support their results.

**Strengths:**

Spectral clustering algorithms are important for a variety of applications and understanding their robustness through the lens of semi-random models could potentially help bridge theory and practice

The paper has a neat conceptual takeaway about using the unnormalized Laplacian as opposed to the normalized Laplacian

**Weaknesses:**

The concrete theoretical results are somewhat weak.  The authors require an upper bound on the maximum edge probability that is comparable to $p,q$ -- this type of assumption isn't necessary for other algorithms such as SDPs.  They also don't achieve the sharp information-theoretic threshold of [ABH16] that other algorithms do.



The experiments are run on purely synthetic data but it would be more compelling if there were experiments on real graphs.  It is not clear that the nonhomogeneous SBM model proposed in the paper is a reasonable model for graphs that appear in practice because of this upper bound on the maximum edge probability and also the fact that no perturbations are allowed for the edges between different communities.

**Questions:**

.

**Limitations:**

Yes

---

> ### Author Rebuttal · Authors · 2024-08-05
>
> Thank you for the helpful feedback! We address each concern here.
>
> > The authors require an upper bound on the maximum edge probability that is comparable to $p, q$  -- this type of assumption isn't necessary for other algorithms such as SDPs.
>
> When the degrees are sufficiently high (i.e., $p, q$ are around $n^{-1/2}$), we do prove that even with arbitrary edge additions within clusters, unnormalized spectral clustering achieves perfect recovery (please see the DCM – Model 2, lines 202-211 on page 5 and the corresponding Theorem 2). This model actually allows more power than adding in-cluster edges to an SBM. Indeed, the adversary can be adaptive, meaning they can draw arbitrary internal edges _after seeing_ the randomized crossing edges. As far as we are aware, such an adversary was not studied even for SDPs or other (more complex) polynomial time algorithms.
>
> However, the stronger model above requires high degree (around $\sqrt{n}$) to guarantee recovery, and it is an open question if we can push this result to the case of $p, q \sim (\log n)/n$ (see the discussion in Appendix A). A promising general direction here is to develop a better understanding of entrywise eigenvector perturbations for high-rank signal matrices. To our knowledge, our analysis is the first to give such a guarantee, but it will be interesting to do this for larger perturbations.
>
> > They also don't achieve the sharp information-theoretic threshold of [ABH16] that other algorithms do.
>
> Our bounds are within constants of the information-theoretic threshold, but yes, we do not know if it is possible to match it. Please refer to the response to reviewer e2Qn.
>
> > It is not clear that the nonhomogeneous SBM model proposed in the paper is a reasonable model for graphs that appear in practice because of this upper bound on the maximum edge probability and also the fact that no perturbations are allowed for the edges between different communities.
>
> Note that even the "vanilla" SBM has been a useful model in practice, by way of developing algorithms, and our model is a strict generalization. Our results provide evidence that unnormalized spectral clustering is a useful algorithm even with a significant amount of perturbation (though our proofs come with the limitation of the upper bound on max edge probability or a high-degree requirement). At a high level, the reason to study semirandom models is to understand if a given algorithm has overfit to some statistical properties about its input that should be morally irrelevant; in this sense, we believe that even with the restrictions, our results show a good level of robustness for spectral clustering. (We also refer to the discussion in lines 65-72 of the introduction, and our response to reviewer e2Qn.)

---

> > ### Comment · Reviewer_hmYB · 2024-08-11
> >
> > Thank you for the response and addressing my concerns/questions.  I still think it would be very interesting and potentially impactful to see how the proposed algorithms perform on real-world graphs!

---

### Official Review · Reviewer_e2Qn · 2024-07-10

**Soundness:** 3
**Presentation:** 3
**Contribution:** 3
**Rating:** 6
**Confidence:** 4

**Summary:**

The authors initially investigate the two-community spectral clustering algorithms applied to the 'Nonhomogeneous Symmetric Stochastic Block Model' (NSSBM). This model is characterized as a more encompassing semi-random framework, permitting a more lenient variation in the selection of in-cluster edge probabilities for each node pair, in comparison to the conventional '(Symmetric) Stochastic Block Model' (SBM). The authors subsequently apply the results derived from the NSSBM model to a more adversarial deterministic model, incorporating necessary adjustments. Finally, numerical simulation results are presented to support the theoretical insights from their study.

**Strengths:**

The idea makes sense and the problem is of interest to the community. The authors provide the theoretical guarantee of the strong consistency of the spectral method for NSSBM which has a potential high rank(\omega(n)) adjacent matrix whereas the counterpart is rank(2) for the traditional SBM model in the population level. Lemma $B.14$ ensures $\mathbold u^\star_2$ is always the second eigenvector of the Laplacian matrix even for the rank(\omega(n)) case(NSSBM).

**Weaknesses:**

(1) The NSSBM model that the authors proposed is not that realistic as variants or perturbations may happen not only in in-cluster edges but also out-cluster edges in practice. Adding the edge only inside the clusters or promoting the in-cluster edge probability will only increase the signal-noise ratio (SNR) for the SBM model(Abbe 2018).

(2) A comparison of the conditions of strong consistency for NSSBM with the existing model should help understand the paper better.

**Questions:**

Is it possible to extend the analysis to the model with variant choices of out-cluster probability for each pair of nodes such that $q_{v, w}\in [\bar{q},q]$. It seems trivial at least at the population level but I am not sure how the new change will affect the perturbation analysis.

---

> ### Author Rebuttal · Authors · 2024-08-05
>
> Thank you for the helpful feedback! We address each concern here.
>
> > Adding the edge only inside the clusters or promoting the in-cluster edge probability will only increase the signal-noise ratio (SNR) for the SBM model(Abbe 2018).
>
> This is typically the case for semi-random corruptions-- the SNR only improves compared to the uncorrupted (or purely stochastic) model. This means that the recovery problem is intuitively "easier", and it indeed is easier in an information theoretical sense. But as it turns out, algorithms for the stochastic case can fail. A classic example here is the semi-random planted clique problem, where a simple spectral algorithm can find cliques of size $\sim \sqrt{n}$ in the purely stochastic case, but it is known that the adjacency matrix-based spectral method fails under a monotone adversary and it is not known whether other spectral methods succeed to this threshold under the monotone adversary (see, e.g., pages 2-3 of https://timroughgarden.org/w17/l/l11.pdf).
>
> Thus in our setting, it is not clear at all if the simple spectral algorithm (which has a reputation for being "brittle") should achieve perfect recovery when we add edges within clusters (in fact, by our Theorem 3, the _normalized_ spectral clustering algorithm provably fails in such a model). So while it would be nicer to be able to show recovery under more general corruptions, even the simpler setting we considered requires several new ideas.
>
> >A comparison of the conditions of strong consistency for NSSBM with the existing model should help understand the paper better.
>
> A comparison of these conditions can be found in lines 191-195 of the submission and in equations (14) and (15) in Appendix F (where we show additional empirical results): when $p,q = \Theta(\log n / n)$, the threshold we obtain is within a constant of that of the SSBM. We can emphasize this more in the final version.
>
> > Is it possible to extend the analysis to the model with variant choices of out-cluster probability for each pair of nodes such that $q_{v,w} \in [\overline{q},q]$? It seems trivial at least at the population level but I am not sure how the new change will affect the perturbation analysis.
>
> This is a very interesting question, and it is posed as an open problem in Appendix A. In our current analysis, we crucially use the fact that the ideal second eigenvector can be easily characterized --- i.e., it is $u_2^{\star} = [-1_{n/2} \oplus 1_{n/2}]/\sqrt{n}$. Further, this means that the Laplacian corresponding the internal edge additions remains orthogonal to $u_2^{\star}$. But if we allow changing the out-cluster probabilities, then even at a population level, $u_2^{\star}$ need not have a simple structural form, which causes the current techniques to fail. We will add more of this discussion to the final version.

---

> > ### Comment · Reviewer_e2Qn · 2024-08-13
> >
> > Thanks for your reply. I happily maintain my current evaluation.

---

### Official Review · Reviewer_q18A · 2024-07-13

**Soundness:** 4
**Presentation:** 4
**Contribution:** 3
**Rating:** 7
**Confidence:** 3

**Summary:**

This paper considers several semirandom variants of the SBM, and investigates whether spectral algorithms achieve exact recovery. The authors give guarantees for the performance of spectral clustering from the unnormalized Laplacian in both a nonhomogenous model, and a model in which the adversary has control over intra-community edges. Suprisingly, the normalized Laplacian is shown to perform worse than the unnormalized Laplacian in the inhomogeneous model.

**Strengths:**

As far as I know, this is the first work that provides guarantees for spectral algorithms in semirandom settings.  While the analysis is an adaptation of the work of Abbe et al on entrywise eigenvector analysis, it requires some new ideas. In particular, the models studied are not low-rank. It is very interesting that the unnormalized Laplacian does better than the normalized Laplacian!

**Weaknesses:**

Sharp constants are not given.

**Questions:**

Could you please give an indication of which parts of your analysis could be tightened to yield sharp constants?

**Limitations:**

Yes

---

> ### Author Rebuttal · Authors · 2024-08-05
>
> Thank you for the supportive review!
> > Could you please give an indication of which parts of your analysis could be tightened to yield sharp constants?
>
> In the current analysis, the Lemmas B.5 and B.6 have constants that we have not optimized, for the sake of clarity in exposition. While it is possible to improve constants slightly by a more tedious analysis, it is an open problem whether they can be improved all the way to the information-theoretic threshold. This question is also conceptually interesting: for certain semirandom models and recovery regimes, the recovery threshold can shift [MPW16], whereas in other cases it does not. We will add this discussion to the final version of the paper.

---

> > ### Comment · Reviewer_q18A · 2024-08-08
> > **Acknowledgement of rebuttal**
> >
> > Thank you for the clarifications!

---

### Official Review · Reviewer_eQb7 · 2024-07-25

**Soundness:** 4
**Presentation:** 4
**Contribution:** 3
**Rating:** 6
**Confidence:** 5

**Summary:**

Spectral clustering has been a popular unsupervised method among mathematical statisticians and theoretical ML researchers. The first analysis of spectral clustering using perturbation analysis (Ng et al., 2001) appeared more than 20 years ago, which, of course, has a lot of limitations: Sparsity is not accounted for, and the number of clusters does not grow as the number of vertices increases.  The two major results that take care of these aspects are (Rohe et al., 2011) and (Lei and Rinaldo, 2015), which establish weak consistency of spectral clustering. On the other hand, there is a lot of literature on the strong consistency and in this context, authors introduces the problem very well. This paper is aimed at studying robustness of spectral algorithms a particular types of model mispecifications, using semirandom adverseries.  This paper provides several results on under what kind of semirandom models spectral clustering exactly recover groundtruth bi-clusters.

**Strengths:**

The problem is introduced very well, and even though there are a lot of technical details, one can easily go through all the results. While the present problem setting and results are based on the previous results in the literature, the contributions are respectable and provide more insights into the robustness of spectral clustering. Particularly, Theorem 3 is very interesting, which along with Theorem 1 and 2, shows that the unnormalized case is more robust to monotone adversarial changes.

**Weaknesses:**

(1) One of the biggest problem with keeping the number of clusters constant and then providing results for "n" large is that in most practical situations number of clusters grow with n. In that sense weak consistency, results that are available in the literature are much more appealing.

(2) Limited to bi-partitioning.

**Questions:**

(1) This paper doesnot deal with weak consistency results. Can you just comment on performing this kind of analysis to weak consistency case where one allow number of clusters to grow.

(2) One of the most important factor in studying spectral clustering under blockmodels models is sparsity. I could not find any comment about this in the paper.

(3) Why is the bi-partitioning constraint? Is it because, when we consider top K eigenvectors, one need to perform K-means which can introduce further problems?

(4) The Cheeger's inquality based description of hard instances for normalized case is nice but not difficult to perceive. Considering that we have higher-order cheeger inequalities, how do we extend this analysis to multi-partition case? Just looking for a comment if authors happens to know about these results.

**Limitations:**

Yes.

---

> ### Author Rebuttal · Authors · 2024-08-06
>
> Thank you for the helpful feedback and questions! We address each concern here.
>
> > One of the biggest problems with keeping the number of clusters constant and then providing results for "n" large is that in most practical situations, the number of clusters grows with n. In that sense of weak consistency, results that are available in the literature are much more appealing.
>
> As you point out, this may constitute a gap between practical scenarios and the theoretical guarantees that are proven within the SBM community. To our knowledge, most of the literature that gives provable guarantees for this problem consists of results where $k$ is independent of $n$ [Abbe17, page 114].
>
> > This paper does not deal with weak consistency results. Can you just comment on performing this kind of analysis to the weak consistency case where one allows the number of clusters to grow.
>
> As highlighted by the survey of [Abbe17], spectral methods actually do not succeed up to the KS threshold, and such thresholds in fact cannot be achieved under a monotone adversary [MPW16]. While there are spectral partitioning algorithms based on higher-order Cheeger inequalities (e.g. [LOT11]), these work with the _normalized_ Laplacian and apply to a different generative model from the one we study. By virtue of our Theorem 3 (and our answer to your last question), these algorithms seem unlikely to be robust to the kinds of perturbations we study in this work. It would be interesting to see whether spectral methods can achieve the threshold obtained by [MPW16] under a monotone adversary.
>
> > One of the most important factors in studying spectral clustering under blockmodels models is sparsity. I could not find any comment about this in the paper.
>
> The literature on block models often works in one of two regimes, the "sparse" regime where $p, q$ are $C/n$ for some constant $C$, and the "high degree" regime, where $p \gg (\log n)/n$. Our work falls into the latter setting, similar to many of the prior strong-consistency results.
>
> > Why is the bi-partitioning constraint? Is it because, when we consider top K eigenvectors, one needs to perform K-means which can introduce further problems?
>
> Yes, if we consider more eigenvectors, simply considering the sign does not suffice for obtaining a partition. For example, the works on higher-order Cheeger inequalities need more careful partitioning algorithms to obtain guarantees [LOT11]. Also, to our knowledge, it is not known whether spectral methods achieve the information-theoretic bound for exact recovery in the case of $k > 2$, even for standard generalizations of the SBM to $k > 2$ communities.
>
> Secondly, analyzing the k=2 setting is already quite non-trivial, e.g., proving that spectral algorithms achieve strong consistency for two communities was a longstanding open problem until relatively recently [AFWZ19]. While it is true that many papers in property-testing [CKKMP18] do consider the general setting of $k$, in the setting of SBMs, designing _any_ algorithm for recovery for $k>2$ often requires much more sophisticated algorithmic machinery: like the recent result of [MRW24], STOC’24 (for general $k$), generalizing the result of [DDNS21], FOCS’21 (for $k = 2$) for the robust community recovery regime. Such results are less appealing in practice than those concerning spectral algorithms, which are commonly used.
>
> > The Cheeger's inequality based description of hard instances for the normalized case is nice but not difficult to perceive. Considering that we have higher-order cheeger inequalities, how do we extend this analysis to the multi-partition case? Just looking for a comment if authors happen to know about these results.
>
> The Cheeger-based intuition of the hard instance for $k=2$ carries over to larger $k$: if the edges added by the adversary create a new, different, $k$-way sparsest cut, the embedding in the bottom $k$ eigenvectors of the normalized Laplacian should reflect (up to $poly(k)$ and square root losses) this new $k$-way sparsest cut as opposed to the planted one.
>
> [LOT11] Multi-way spectral partitioning and higher-order Cheeger inequalities (https://arxiv.org/abs/1111.1055)
>
> [MPW16] How Robust are Reconstruction Thresholds for Community Detection? (https://arxiv.org/abs/1511.01473)
>
> [Abbe17] Community Detection and Stochastic Block Models (https://arxiv.org/abs/1703.10146)
>
> [CKKMP18] Testing Graph Clusterability: Algorithms and Lower Bounds (https://arxiv.org/abs/1808.04807)
>
> [AFWZ19] Entrywise Eigenvector Analysis of Random Matrices with Low Expected Rank (https://arxiv.org/abs/1709.09565)
>
> [DDNS21] Robust recovery for stochastic block models (https://arxiv.org/abs/2111.08568)
>
> [MRW24] Robust recovery for stochastic block models, simplified and generalized (https://arxiv.org/abs/2402.13921)

---

> > ### Comment · Reviewer_eQb7 · 2024-08-13
> > **Thanks**
> >
> > Thanks for your replies. Regarding allowing the number of clusters to grow: Check
> > "Spectral clustering and the high-dimensional stochastic blockmodel" by  Karl Rohe, Sourav Chatterjee, Bin Yu

---

> > > ### Author Response · Authors · 2024-08-13
> > >
> > > Thanks for pointing this out, we will add a pointer to this classic reference in the camera-ready version. However, note that the work only guarantees weak recovery and also assumes that $k$-means can be solved optimally, so the regime is different from ours.

---

### Author Rebuttal · Authors · 2024-08-05

We would like to thank all the reviewers for their positive comments and their support of our paper. We are excited to hear the feedback and have responded to questions/provided some clarifications in comments directly responding to each review.

---

### Decision · Program_Chairs · 2024-09-25

**Decision:**

Accept (poster)

**Comment:**

The reviewers generally agreed that the manuscript gives important and interesting insights into the performance of spectral algorithms for exact recovery in semi-random stochastic block models. They also gave a number of suggestions to improve the manuscript. Please address the reviewers' comments/questions carefully when you prepare the camera-ready version.